# Deciding What to Model: Value-Equivalent Sampling for Reinforcement Learning

**Dilip Arumugam**
Department of Computer Science
Stanford University
dilip@cs.stanford.edu

**Benjamin Van Roy**
Department of Electrical Engineering
Department of Management Science & Engineering
Stanford University
bvr@stanford.edu

## Abstract

The quintessential model-based reinforcement-learning agent iteratively refines its estimates or prior beliefs about the true underlying model of the environment. Recent empirical successes in model-based reinforcement learning with function approximation, however, eschew the true model in favor of a surrogate that, while ignoring various facets of the environment, still facilitates effective planning over behaviors. Recently formalized as the value equivalence principle, this algorithmic technique is perhaps unavoidable as real-world reinforcement learning demands consideration of a simple, computationally-bounded agent interacting with an overwhelmingly complex environment, whose underlying dynamics likely exceed the agent's capacity for representation. In this work, we consider the scenario where agent limitations may entirely preclude identifying an exactly value-equivalent model, immediately giving rise to a trade-off between identifying a model that is simple enough to learn while only incurring bounded sub-optimality. To address this problem, we introduce an algorithm that, using rate-distortion theory, iteratively computes an approximately-value-equivalent, lossy compression of the environment which an agent may feasibly target in lieu of the true model. We prove an information-theoretic, Bayesian regret bound for our algorithm that holds for any finite-horizon, episodic sequential decision-making problem. Crucially, our regret bound can be expressed in one of two possible forms, providing a performance guarantee for finding either the simplest model that achieves a desired sub-optimality gap or, alternatively, the best model given a limit on agent capacity.

## 1 Introduction

A central challenge of the reinforcement-learning problem [154, 87] is exploration, where a sequential decision-making agent must judiciously balance exploitation of knowledge accumulated thus far against the need to further acquire information for optimal long-term performance. Historically, provably-efficient reinforcement-learning algorithms [91, 35, 90, 20, 23, 151, 81, 121, 45, 120, 22, 46, 10, 85, 168, 56, 105] have often relied upon one of two possible mechanisms for addressing the exploration challenge in a principled manner: optimism in the face of uncertainty or posterior sampling. Briefly, methods in the former category begin with optimistically-biased value estimates for all state-action pairs; an agent acting greedily with respect to these estimates will be incentivized to visit all state-action pairs a sufficient number of times until this bias dissipates and the agent is left with an accurate estimate of the value function for deriving optimal behavior. In contrast, posterior-sampling methods primarily operate based on Thompson sampling [156, 141] whereby the agent begins with a prior belief over the Markov Decision Process (MDP) with which it is interacting and acts optimally with respect to a single sample drawn from these beliefs. The resulting experience sampled from the true environment allows the agent to derive a corresponding posterior distribution

36th Conference on Neural Information Processing Systems (NeurIPS 2022).

and this Posterior Sampling for Reinforcement Learning (PSRL) [152] algorithm proceeds iteratively in this manner, eventually arriving at a posterior sharply concentrated around the true environment MDP. While both paradigms have laid down solid theoretical foundations for provably-efficient reinforcement learning, a line of work has demonstrated how posterior-sampling methods can be more favorable both in theory and in practice [121–123, 120, 114, 125, 63].

While existing analyses of reinforcement-learning algorithms have largely focused on providing guarantees for learning optimal solutions, real-world reinforcement learning demands consideration for a computationally-bounded agent interacting with an overwhelmingly complex environment [105]. A simplified view of this notion can be succinctly depicted in the multi-armed bandit setting [97, 38, 100]; as the number of arms increases, a Thompson sampling agent's relentless pursuit of the optimal arm will lead to large regret [138, 139]. On the other hand, one might simply settle for the first $\varepsilon$-optimal arm found, for some $\varepsilon > 0$, which may be identified in far fewer time periods. The goal of this work is to augment PSRL so as to accommodate these satisficing solutions in addition to optimal ones, paralleling existing work for satisficing in multi-armed bandit problems [140, 139, 15, 16]. To help elucidate the utility of satisficing solutions in the reinforcement-learning setting, we offer the following illustrative example:

**Example 1** (A Multi-Resolution MDP). *For a large but finite $N \in \mathbb{N}$, consider a sequence of MDPs, $\{\mathcal{M}_n\}_{n \in [N]}$, which all share a common action space $\mathcal{A}$ but vary in state space $\mathcal{S}_n$, reward function, and transition function. Moreover, for each $n \in [N]$, the rewards of the $n$th MDP are bounded in the interval $[0, \frac{1}{n}]$. An agent is confronted with the resulting product MDP, $\mathcal{M}$, defined on the state space $\mathcal{S}_1 \times \ldots \times \mathcal{S}_N$ with action space $\mathcal{A}$ and rewards summed across the $N$ constituent reward functions. The transition function is defined such that each action $a \in \mathcal{A}$ is executed across all $N$ MDPs simultaneously and the resulting individual transitions are combined into a transition of $\mathcal{M}$.*

Example 1 presents a simple scenario where, as $N \uparrow \infty$, a complex environment retains a wealth of information and yet, due to the scale of $N$ and the boundedness of rewards for each constituent MDP $\mathcal{M}_n$, only a subset of that information is within the agent's reach or even necessary for producing reasonably competent behavior. Despite this fact, PSRL will persistently act to fully identify the transition and reward structure of all $\{\mathcal{M}_n\}_{n \in [N]}$, for any value of $N$. Without knowing which MDPs are more important *a priori* and even as data accumulates during learning, PSRL is unable to forego learning granular components of $\mathcal{M}$, eventually accumulating optimal reward at the cost of more time. Intuitively, however, one might anticipate that there exists a value $M \ll N$ such that learning the subsequence of MDPs $\{\mathcal{M}_n\}_{n \in [M]}$ in fewer time periods is sufficient for achieving a desired degree of sub-optimality, since the rewards of the remaining MDPs $\{\mathcal{M}_n\}_{n > M}$ make suitably negligible contributions to the overall rewards of $\mathcal{M}$. Alternatively, for a computationally-bounded decision maker, the agent's resource limitations ought to translate into a value $C \ll N$ such that $\{\mathcal{M}_n\}_{n \in [C]}$ is feasible and learning this subsequence is the best possible outcome under the agent capacity constraints. In this work, we introduce an algorithm that, in a purely data-driven and automated fashion, implicitly identifies such a value $M$ or $C$ to facilitate tractable, near-optimal learning in what may otherwise be an intractable problem. Following Arumugam and Van Roy [15], a key tool for defining a notion of satisficing in reinforcement learning will be rate-distortion theory [146, 25].

The paper proceeds as follows: we introduce our problem formulation in Section 3, present our generalization of PSRL in Section 4, and provide a complementary regret analysis in Section 5. Due to space constraints, technical proofs, an overview of related work, and discussion of our results in a broader context are relegated to the appendix.

## 2 Preliminaries

In this section, we provide brief background on information theory and details on our notation. All random variables are defined on a probability space $(\Omega, \mathcal{F}, \mathbb{P})$. For any random variable $X : \Omega \to \mathcal{X}$ taking values on the measurable space $(\mathcal{X}, \mathbb{X})$, we use $\sigma(X) \triangleq \{X^{-1}(A) \mid A \in \mathbb{X}\} \subseteq \mathcal{F}$ to denote the $\sigma$-algebra generated by $X$. For any natural number $N \in \mathbb{N}$, we denote the index set as $[N] \triangleq \{1, 2, \ldots, N\}$. For any arbitrary set $\mathcal{X}$, $\Delta(\mathcal{X})$ denotes the set of all probability distributions with support on $\mathcal{X}$. For any two arbitrary sets $\mathcal{X}$ and $\mathcal{Y}$, we denote the class of all (measurable) functions mapping from $\mathcal{X}$ to $\mathcal{Y}$ as $\{\mathcal{X} \to \mathcal{Y}\} \triangleq \{f \mid f : \mathcal{X} \to \mathcal{Y}\}$. While our exposition throughout the paper will consistently refer to bits of information, it will be useful for the purposes of analysis that all logarithms be in base $e$.

## 2.1 Information Theory

Here we introduce various concepts in probability theory and information theory used throughout this paper. We encourage readers to consult [41, 71, 129, 59] for more background.

We define the mutual information between any two random variables $X, Y$ through the Kullback-Leibler (KL) divergence:

$$\mathbb{I}(X;Y) = D_{\mathrm{KL}}(\mathbb{P}((X,Y) \in \cdot) \,||\, \mathbb{P}(X \in \cdot) \times \mathbb{P}(Y \in \cdot)) \qquad D_{\mathrm{KL}}(P \,||\, Q) = \begin{cases} \int \log\left(\frac{dP}{dQ}\right) dP & P \ll Q \\ +\infty & P \not\ll Q \end{cases},$$

where $P$ and $Q$ are both probability measures on the same measurable space and $\frac{dP}{dQ}$ denotes the Radon-Nikodym derivative of $P$ with respect to $Q$. An analogous definition of conditional mutual information holds through the expected KL-divergence for any three random variables $X, Y, Z$:

$$\mathbb{I}(X;Y \mid Z) = \mathbb{E}\left[D_{\mathrm{KL}}(\mathbb{P}((X,Y) \in \cdot \mid Z) \,||\, \mathbb{P}(X \in \cdot \mid Z) \times \mathbb{P}(Y \in \cdot \mid Z))\right].$$

With these definitions in hand, we may define the entropy and conditional entropy for any two random variables $X, Y$ as

$$\mathbb{H}(X) = \mathbb{I}(X;X) \qquad \mathbb{H}(Y \mid X) = \mathbb{H}(Y) - \mathbb{I}(X;Y).$$

This yields the following identities for mutual information and conditional mutual information for any three arbitrary random variables $X$, $Y$, and $Z$:

$$\mathbb{I}(X;Y) = \mathbb{H}(X) - \mathbb{H}(X \mid Y) = \mathbb{H}(Y) - \mathbb{H}(Y|X), \qquad \mathbb{I}(X;Y|Z) = \mathbb{H}(X|Z) - \mathbb{H}(X \mid Y, Z) = \mathbb{H}(Y|Z) - \mathbb{H}(Y|X,Z).$$

Through the chain rule of the KL-divergence and the fact that $D_{\mathrm{KL}}(P \,||\, P) = 0$ for any probability measure $P$, we obtain another equivalent definition of mutual information,

$$\mathbb{I}(X;Y) = \mathbb{E}\left[D_{\mathrm{KL}}(\mathbb{P}(Y \in \cdot \mid X) \,||\, \mathbb{P}(Y \in \cdot))\right],$$

as well as the chain rule of mutual information: $\mathbb{I}(X;Y_1, \ldots, Y_n) = \sum_{i=1}^{n} \mathbb{I}(X;Y_i \mid Y_1, \ldots, Y_{i-1})$.

Finally, for any three random variables $X$, $Y$, and $Z$ which form the Markov chain $X \to Y \to Z$, we have the following data-processing inequality: $\mathbb{I}(X;Z) \leq \mathbb{I}(X;Y)$.

## 3 Problem Formulation

We formulate a sequential decision-making problem as a finite-horizon, episodic Markov Decision Process (MDP) [24, 130] defined by $\mathcal{M} = \langle \mathcal{S}, \mathcal{A}, \mathcal{R}, \mathcal{T}, \beta, H \rangle$. Here $\mathcal{S}$ denotes a set of states, $\mathcal{A}$ is a set of actions, $\mathcal{R} : \mathcal{S} \times \mathcal{A} \to [0,1]$ is a deterministic reward function providing evaluative feedback signals (in the unit interval) to the agent, $\mathcal{T} : \mathcal{S} \times \mathcal{A} \to \Delta(\mathcal{S})$ is a transition function prescribing distributions over next states, $\beta \in \Delta(\mathcal{S})$ is an initial state distribution, and $H \in \mathbb{N}$ is the maximum episode length or horizon.

As is standard in Bayesian reinforcement learning [70], neither the transition function nor the reward function are known to the agent and, consequently, both are treated as random variables. Since all other components of the MDP are thought of as known a priori, the randomness in the model $(\mathcal{R}, \mathcal{T})$ fully accounts for the randomness in $\mathcal{M}$, which is also a random variable. We denote by $\mathcal{M}^\star$ the true MDP with model $(\mathcal{R}^\star, \mathcal{T}^\star)$ that the agent interacts with and attempts to solve over the course of $K$ episodes. Within each episode, the agent acts for exactly $H$ steps beginning with an initial state $s_1 \sim \beta$. For each $h \in [H]$, the agent observes the current state $s_h \in \mathcal{S}$, selects action $a_h \sim \pi_h(\cdot \mid s_h) \in \mathcal{A}$, enjoys a reward $r_h = \mathcal{R}(s_h, a_h) \in [0,1]$, and transitions to the next state $s_{h+1} \sim \mathcal{T}(\cdot \mid s_h, a_h) \in \mathcal{S}$.

A stationary, stochastic policy for timestep $h \in [H]$, $\pi_h : \mathcal{S} \to \Delta(\mathcal{A})$, encodes a pattern of behavior mapping individual states to distributions over possible actions. Letting $\{\mathcal{S} \to \Delta(\mathcal{A})\}$ denote the class of all stationary, stochastic policies, a non-stationary policy $\pi = (\pi_1, \ldots, \pi_H) \in \{\mathcal{S} \to \Delta(\mathcal{A})\}^H$ is a collection of exactly $H$ stationary, stochastic policies whose overall performance in any MDP $\mathcal{M}$ at timestep $h \in [H]$ when starting at state $s \in \mathcal{S}$ and taking action $a \in \mathcal{A}$ is assessed by its associated action-value function $Q_{\mathcal{M},h}^\pi(s,a) = \mathbb{E}\left[\sum_{h'=h}^{H} \mathcal{R}(s_{h'}, a_{h'}) \mid s_h = s, a_h = a\right]$, where the

expectation integrates over randomness in the action selections and transition dynamics. Taking the corresponding value function as $V_{\mathcal{M},h}^\pi(s) = \mathbb{E}_{a \sim \pi_h(\cdot|s)}\left[Q_{\mathcal{M},h}^\pi(s,a)\right]$, we define the optimal policy $\pi^\star = (\pi_1^\star, \pi_2^\star, \ldots, \pi_H^\star)$ as achieving supremal value $V_{\mathcal{M},h}^\star(s) = \sup_{\pi \in \{\mathcal{S} \to \Delta(\mathcal{A})\}^H} V_{\mathcal{M},h}^\pi(s)$ for all $s \in \mathcal{S}$, $h \in [H]$. For brevity, we will write any value function $V \in \{S \to \mathbb{R}\}$ without its argument to implicitly integrate over randomness in the initial state: $V = \mathbb{E}_{s_1 \sim \beta(\cdot)}[V(s_1)]$. We let $\tau_k = (s_1^{(k)}, a_1^{(k)}, r_1^{(k)}, \ldots, s_H^{(k)}, a_H^{(k)}, r_H^{(k)}, s_{H+1}^{(k)})$ be the random variable denoting the trajectory experienced by the agent in the $k$th episode. Meanwhile, $H_k = \{\tau_1, \tau_2, \ldots, \tau_{k-1}\} \in \mathcal{H}_k$ is the random variable representing the entire history of the agent's interaction within the environment at the start of the $k$th episode; the sequence of history random variables $\{H_k\}_{k \in [K]}$ induce and, by definition, are adapted to the filtration $\{\sigma(H_k)\}_{k \in [K]}$ of $(\Omega, \mathcal{F})$. We call attention to the fact that we have yet to make any further restrictions on the state-action space $\mathcal{S} \times \mathcal{A}$, such as finiteness; notably, the main results of this paper are not limited to tabular MDPs. As mentioned by Lattimore and Szepesvári [100] (also as Proposition 7.28 of Bertsekas and Shreve [28]), the Ionescu-Tulcea Theorem [80] ensures the existence of a probability space upon which $\tau_k$ and $H_k$ are well-defined random variables for all episodes $k \in [K]$.

Abstractly, a reinforcement-learning algorithm is a sequence of non-stationary policies $(\pi^{(1)}, \ldots, \pi^{(K)})$ where for each episode $k \in [K]$, $\pi^{(k)} : \mathcal{H}_k \to \{\mathcal{S} \to \Delta(\mathcal{A})\}^H$ is a function of the current history $H_k$. We define the regret of a reinforcement-learning algorithm over $K$ episodes as

$$\text{REGRET}(K, \pi^{(1)}, \ldots, \pi^{(K)}, \mathcal{M}^\star) = \sum_{k=1}^{K} \Delta_k \qquad \Delta_k \triangleq V_{\mathcal{M}^\star,1}^\star - V_{\mathcal{M}^\star,1}^{\pi^{(k)}},$$

where $\Delta_k$ denotes the episodic regret or regret incurred during the $k$th episode with respect to the true MDP $\mathcal{M}^\star$. An agent's initial uncertainty in the (unknown) true MDP $\mathcal{M}^\star$ is reflected by an arbitrary prior distribution $\mathbb{P}(\mathcal{M}^\star \in \cdot \mid H_1)$. Since the regret is a random variable due to our uncertainty in $\mathcal{M}^\star$, we integrate over this randomness to arrive at the Bayesian regret:

$$\text{BAYESREGRET}(K, \pi^{(1)}, \ldots, \pi^{(K)}) = \mathbb{E}\left[\text{REGRET}(K, \pi^{(1)}, \ldots, \pi^{(K)}, \mathcal{M}^\star)\right].$$

Broadly speaking, our goal is to design a provably-efficient reinforcement-learning algorithm that incurs bounded Bayesian regret.

Throughout the paper, we will denote the entropy and conditional entropy conditioned upon a specific realization of an agent's history $H_k$, for some episode $k \in [K]$, as $\mathbb{H}_k(X) \triangleq \mathbb{H}(X \mid H_k = H_k)$ and $\mathbb{H}_k(X \mid Y) \triangleq \mathbb{H}_k(X \mid Y, H_k = H_k)$, for two arbitrary random variables $X$ and $Y$. This notation will also apply analogously to the mutual information $\mathbb{I}_k(X;Y) \triangleq \mathbb{I}(X;Y \mid H_k = H_k) = \mathbb{H}_k(X) - \mathbb{H}_k(X \mid Y) = \mathbb{H}_k(Y) - \mathbb{H}_k(Y \mid X)$, as well as the conditional mutual information $\mathbb{I}_k(X;Y \mid Z) \triangleq \mathbb{I}(X;Y \mid H_k = H_k, Z)$, given an arbitrary third random variable, $Z$. Note that their dependence on the realization of random history $H_k$ makes both $\mathbb{I}_k(X;Y)$ and $\mathbb{I}_k(X;Y \mid Z)$ random variables themselves. The traditional notion of conditional mutual information given the random variable $H_k$ arises by integrating over this randomness:

$$\mathbb{E}\left[\mathbb{I}_k(X;Y)\right] = \mathbb{I}(X;Y \mid H_k) \qquad \mathbb{E}\left[\mathbb{I}_k(X;Y \mid Z)\right] = \mathbb{I}(X;Y \mid H_k, Z).$$

Additionally, we will also adopt a similar notation to express a conditional expectation given the random history $H_k$: $\mathbb{E}_k[X] \triangleq \mathbb{E}[X|H_k]$.

## 4  Satisficing Through Posterior Sampling

### 4.1  Rate-Distortion Theory

We begin with a brief, high-level overview of rate-distortion theory [146, 25] and encourage readers to consult [41] for more details and [26] for a survey of advances in rate-distortion theory towards solving the lossy source coding problem in information theory. A lossy compression problem consumes as input a fixed information source $\mathbb{P}(X \in \cdot)$ and a measurable distortion function $d : \mathcal{X} \times \mathcal{Z} \to \mathbb{R}_{\geq 0}$ which quantifies the loss of fidelity by using $Z$ in place of $X$. Then, for any

$D \in \mathbb{R}_{\geq 0}$, the rate-distortion function quantifies the fundamental limit of lossy compression as

$$\mathcal{R}(D) = \inf_{Z \in \Lambda(D)} \mathbb{I}(X; Z) \qquad \Lambda(D) \triangleq \{Z : \Omega \to \mathcal{Z} \mid \mathbb{E}\left[d(X, Z)\right] \leq D\},$$

where the infimum is taken over all random variables $Z$ that incur bounded expected distortion, $\mathbb{E}\left[d(X, Z)\right] \leq D$. Naturally, $\mathcal{R}(D)$ represents the minimum number of bits of information that must be retained from $X$ in order to achieve this bounded expected loss of fidelity[1]. Throughout the paper, various facts of the rate-distortion function will be referenced as needed. For now, we simply note that, in keeping with the problem formulation of the previous section which does not automatically assume discrete random variables, the rate-distortion function is well-defined for abstract information source and channel output random variables [43].

Just as in past work that studies satisficing in multi-armed bandit problems [138, 139, 15], we will use rate-distortion theory to formalize and identify the best simplified MDP $\widetilde{\mathcal{M}}_k$ that the agent will attempt to learn over the course of each episode $k \in [K]$. The dependence on the particular episode comes from the fact that this lossy compression mechanism or channel will treat the agent's current beliefs over the true MDP $\mathbb{P}(\mathcal{M}^\star \in \cdot \mid H_k)$ as the information source to be compressed.

## 4.2 The Value Equivalence Principle

As outlined in the previous section, the second input for a well-specified lossy-compression problem is a distortion function prescribing non-negative real values to realizations of the information source and channel output random variables $(\mathcal{M}^\star, \widetilde{\mathcal{M}})$ that quantify the loss of fidelity incurred by using $\widetilde{\mathcal{M}}$ in lieu of $\mathcal{M}^\star$. To define this function, we will leverage an approximate notion of value equivalence [72, 73]. For any arbitrary MDP $\mathcal{M}$ with model $(\mathcal{R}, \mathcal{T})$ and any stationary, stochastic policy $\pi : \mathcal{S} \to \Delta(\mathcal{A})$, define the Bellman operator $\mathcal{B}^\pi_\mathcal{M} : \{\mathcal{S} \to \mathbb{R}\} \to \{\mathcal{S} \to \mathbb{R}\}$ as follows:

$$\mathcal{B}^\pi_\mathcal{M} V(s) \triangleq \mathbb{E}_{a \sim \pi(\cdot|s)}\left[\mathcal{R}(s, a) + \mathbb{E}_{s' \sim \mathcal{T}(\cdot|s,a)}\left[V(s')\right]\right], \qquad \forall s \in \mathcal{S}.$$

The Bellman operator is a foundational tool in dynamic-programming approaches to reinforcement learning [29] and gives rise to the classic Bellman equation: for any MDP $\mathcal{M} = \langle \mathcal{S}, \mathcal{A}, \mathcal{R}, \mathcal{T}, \beta, H \rangle$ and any non-stationary policy $\pi = (\pi_1, \ldots, \pi_H)$, the value functions induced by $\pi$ satisfy $V^\pi_{\mathcal{M},h}(s) = \mathcal{B}^{\pi_h}_\mathcal{M} V^\pi_{\mathcal{M},h+1}(s)$, for all $h \in [H]$ and with $V^\pi_{\mathcal{M},H+1}(s) = 0, \forall s \in \mathcal{S}$. For any two MDPs $\mathcal{M} = \langle \mathcal{S}, \mathcal{A}, \mathcal{R}, \mathcal{T}, \beta, H \rangle$ and $\widehat{\mathcal{M}} = \langle \mathcal{S}, \mathcal{A}, \widehat{\mathcal{R}}, \widehat{\mathcal{T}}, \beta, H \rangle$, Grimm et al. [72] define a notion of equivalence between them despite their differing models. For any policy class $\Pi \subseteq \{\mathcal{S} \to \Delta(\mathcal{A})\}$ and value function class $\mathcal{V} \subseteq \{\mathcal{S} \to \mathbb{R}\}$, $\mathcal{M}$ and $\widehat{\mathcal{M}}$ are value equivalent with respect to $\Pi$ and $\mathcal{V}$ if and only if $\mathcal{B}^\pi_\mathcal{M} V = \mathcal{B}^\pi_{\widehat{\mathcal{M}}} V, \forall \pi \in \Pi, V \in \mathcal{V}$. In words, two different models are deemed value equivalent if they induce identical Bellman updates under any pair of policy and value function from $\Pi \times \mathcal{V}$. Grimm et al. [72] prove that when $\Pi = \{\mathcal{S} \to \Delta(\mathcal{A})\}$ and $\mathcal{V} = \{\mathcal{S} \to \mathbb{R}\}$, the set of all exactly value-equivalent models is a singleton set containing only the true model of the environment. The key insight behind value equivalence, however, is that practical model-based reinforcement-learning algorithms need not be concerned with modeling every granular detail of the underlying environment and may, in fact, stand to benefit by optimizing an alternative criterion besides the traditional maximum-likelihood objective [147, 66, 116, 17, 65, 57, 1, 44, 21, 144, 110, 113, 161]. Indeed, by restricting focus to decreasing subsets of policies $\Pi \subset \{\mathcal{S} \to \Delta(\mathcal{A})\}$ and value functions $\mathcal{V} \subset \{\mathcal{S} \to \mathbb{R}\}$, the space of exactly value-equivalent models is monotonically increasing.

For brevity, let $\mathfrak{R} \triangleq \{\mathcal{S} \times \mathcal{A} \to [0, 1]\}$ and $\mathfrak{T} \triangleq \{\mathcal{S} \times \mathcal{A} \to \Delta(\mathcal{S})\}$ denote the classes of all reward functions and transition functions, respectively. Recall that, with $\langle \mathcal{S}, \mathcal{A}, \beta, H \rangle$ all known, the uncertainty in a random MDP $\mathcal{M}$ is entirely driven by its model $(\mathcal{R}, \mathcal{T})$ such that we may think of the support of $\mathcal{M}^\star$ as $\text{supp}(\mathcal{M}^\star) = \mathfrak{M} \triangleq \mathfrak{R} \times \mathfrak{T}$. We define a distortion function on pairs of MDPs $d : \mathfrak{M} \times \mathfrak{M} \to \mathbb{R}_{\geq 0}$ for any $\Pi \subseteq \{\mathcal{S} \to \Delta(\mathcal{A})\}, \mathcal{V} \subseteq \{\mathcal{S} \to \mathbb{R}\}$ as

$$d_{\Pi,\mathcal{V}}(\mathcal{M}, \widehat{\mathcal{M}}) = \sup_{\substack{\pi \in \Pi \\ V \in \mathcal{V}}} ||\mathcal{B}^\pi_\mathcal{M} V - \mathcal{B}^\pi_{\widehat{\mathcal{M}}} V||^2_\infty = \sup_{\substack{\pi \in \Pi \\ V \in \mathcal{V}}} \left(\sup_{s \in \mathcal{S}} |\mathcal{B}^\pi_\mathcal{M} V(s) - \mathcal{B}^\pi_{\widehat{\mathcal{M}}} V(s)|\right)^2.$$

In words, $d_{\Pi,\mathcal{V}}$ is the supremal squared Bellman error between MDPs $\mathcal{M}$ and $\widehat{\mathcal{M}}$ across all states $s \in \mathcal{S}$ with respect to the policy class $\Pi$ and value function class $\mathcal{V}$.

---

[1] With a slight abuse of notation, we overload $\mathcal{R}$.

## 4.3 Value-Equivalent Sampling for Reinforcement Learning

By virtue of the previous two sections, we are now in a position to define the lossy compression problem that characterizes a MDP $\widetilde{\mathcal{M}}$ that the agent will aspire to learn in each episode $k \in [K]$ instead of the true MDP $\mathcal{M}^\star$. For any $\Pi \subseteq \{\mathcal{S} \to \Delta(\mathcal{A})\}$; $\mathcal{V} \subseteq \{\mathcal{S} \to \mathbb{R}\}$; $k \in [K]$; and $D \geq 0$, we define the rate-distortion function

$$\mathcal{R}_k^{\Pi,\mathcal{V}}(D) = \inf_{\widetilde{\mathcal{M}} \in \Lambda_k(D)} \mathbb{I}_k(\mathcal{M}^\star; \widetilde{\mathcal{M}}), \ \Lambda_k(D) \triangleq \{\widetilde{\mathcal{M}} : \Omega \to \mathfrak{M} \mid \mathbb{E}_k[d_{\Pi,\mathcal{V}}(\mathcal{M}^\star, \widetilde{\mathcal{M}})] \leq D\}. \quad (1)$$

This rate-distortion function characterizes the fundamental limit of MDP compression under our chosen distortion measure resulting in a channel that retains the minimum amount of information from the true MDP $\mathcal{M}^\star$ while yielding an approximately value-equivalent MDP in expectation. Observe that this distortion constraint is a notion of approximate value equivalence which collapses to the exact value equivalence of Grimm et al. [72] as $D \to 0$. Meanwhile, as $D \to \infty$, we accommodate a more aggressive compression of the true MDP $\mathcal{M}^\star$ resulting in less faithful Bellman updates.

---

**Algorithm 1** Posterior Sampling for Reinforcement Learning (PSRL) [152]

**Input:** Prior $\mathbb{P}(\mathcal{M}^\star \in \cdot \mid H_1)$
**for** $k \in [K]$ **do**
    Sample $M_k \sim \mathbb{P}(\mathcal{M}^\star \in \cdot \mid H_k)$
    Get optimal policy $\pi^{(k)} = \pi^\star_{M_k}$
    Execute $\pi^{(k)}$ and get trajectory $\tau_k$
    Update history $H_{k+1} = H_k \cup \tau_k$
    Induce posterior $\mathbb{P}(\mathcal{M}^\star \in \cdot \mid H_{k+1})$
**end for**

**Algorithm 2** Value-equivalent Sampling for Reinforcement Learning (VSRL)

**Input:** Prior $\mathbb{P}(\mathcal{M}^\star \in \cdot \mid H_1)$, Threshold $D \in \mathbb{R}_{\geq 0}$, Distortion function $d_{\Pi,\mathcal{V}} : \mathfrak{M} \times \mathfrak{M} \to \mathbb{R}_{\geq 0}$
**for** $k \in [K]$ **do**
    Compute $\widetilde{\mathcal{M}}_k$ achieving $\mathcal{R}_k^{\Pi,\mathcal{V}}(D)$ limit (Equation 1)
    Sample MDP $M^\star \sim \mathbb{P}(\mathcal{M}^\star \in \cdot \mid H_k)$
    Sample compression $M_k \sim \mathbb{P}(\widetilde{\mathcal{M}}_k \in \cdot \mid \mathcal{M}^\star = M^\star)$
    Compute optimal policy $\pi^{(k)} = \pi^\star_{M_k}$
    Execute $\pi^{(k)}$ and observe trajectory $\tau_k$
    Update history $H_{k+1} = H_k \cup \tau_k$
    Induce posterior $\mathbb{P}(\mathcal{M}^\star \in \cdot \mid H_{k+1})$
**end for**

---

A standard algorithm for our problem setting is widely known as Posterior Sampling for Reinforcement Learning (PSRL) [152, 120], which we present as Algorithm 1, while our Value-equivalent Sampling for Reinforcement Learning (VSRL) is given as Algorithm 2. The key distinction between them is that, at each episode $k \in [K]$, the latter takes the posterior sample $M^\star \sim \mathbb{P}(\mathcal{M}^\star \in \cdot \mid H_k)$ and passes it through the channel that achieves the rate-distortion limit (Equation 1) at this episode to get the $M_k$ whose optimal policy is executed in the environment.

The core impetus for this work is to recognize that, for complex environments, pursuit of the exact MDP $\mathcal{M}^\star$ (as in PSRL) may be an entirely infeasible goal. Consider a MDP that represents control of a real-world, physical system; learning a transition function of the associated environment, at some level, demands that the agent internalize the laws of physics and motion with near-perfect accuracy. More formally, identifying $\mathcal{M}^\star$ demands the agent obtain exactly $\mathbb{H}_1(\mathcal{M}^\star)$ bits of information from the environment which, under an uninformative prior, may either be prohibitively large by far exceeding the agent's capacity constraints or be simply impractical under time and resource constraints.

As a remedy for this problem, we embrace the idea of being "sufficiently satisfying" or *satisficing* [148, 140, 138, 139, 15, 16]; as succinctly stated by Herbert A. Simon during his 1978 Nobel Memorial Lecture, "decision makers can satisfice either by finding optimum solutions for a simplified world, or by finding satisfactory solutions for a more realistic world." Rather than spend an inordinate amount of time trying to recover an optimum solution to the true environment, we will instead design an algorithm that pursues optimum solutions for a sequence of simplified environments. In the next section, our analysis demonstrates that finding such optimum solutions for simplified worlds ultimately acts as a mechanism for achieving a satisfactory solution for the realistic, complex world. Naturally, the loss of fidelity between the simplified and true environments translates into a fixed amount of regret that an agent designer consciously and willingly accepts for two reasons: (1) they expect a reduction in the amount of time, data, and bits of information needed to identify the simplified environment and (2) in tasks where the environment encodes irrelevant information and

exact knowledge is not needed to achieve optimal behavior [66, 72, 73, 161], this worst-case error term may be negligible anyways while still maintaining greater efficiency than traditional PSRL.

Recalling Example 1 that revolves around a particular sequence of MDPs, $\{\mathcal{M}_n\}_{n \in [N]}$, we note that as the distortion threshold $D$ increases, the significance of MDPs in the sequence indexed by larger values of $n \in [N]$ rapidly diminishes. As $D \uparrow \infty$, the lossy compression $\widetilde{\mathcal{M}}_k$ needn't convey information about any of the MDPs in $\{\mathcal{M}_n\}_{n \in [N]}$. Conversely, at $D = 0$, a VSRL agent must necessarily obtain enough information about the entire sequence so as to facilitate planning over $\Pi$ and $\mathcal{V}$. In between, however, the agent need only concern itself with a particular subsequence of $\{\mathcal{M}_n\}_{n \in [N]}$ while the remaining MDPs can be ignored due to their negligible contribution to overall value and, therefore, expected distortion under $d_{\Pi,\mathcal{V}}$.

## 5  Regret Analysis

In this section, we offer an information-theoretic analysis of VSRL (Algorithm 2) before refining our regret bounds to the tabular setting. We conclude by highlighting how our performance guarantees can be expressed via a notion of agent capacity that is considerate of real-world reinforcement learning.

### 5.1  An Information-Theoretic Bayesian Regret Bound

To establish a Bayesian regret bound for VSRL we first require a regret decomposition that acknowledges the agent's new objective of identifying an approximately value-equivalent MDP in each episode, $\widetilde{\mathcal{M}}_k$, rather than the true MDP $\mathcal{M}^\star$. Crucially, this regret decomposition leverages the precise form of our distortion function $d_{\Pi,\mathcal{V}}(\mathcal{M}^\star, \widetilde{\mathcal{M}}_k)$.

**Theorem 1.** *Take any $\Pi \supseteq \{\mathcal{S} \rightarrow \mathcal{A}\}$, any $\mathcal{V} \supseteq \{V^\pi \mid \pi \in \Pi^H\}$, and fix any $D \geq 0$. For each episode $k \in [K]$, let $\widetilde{\mathcal{M}}_k$ be any MDP that achieves the rate-distortion limit of $\mathcal{R}_k^{\Pi,\mathcal{V}}(D)$ with information source $\mathbb{P}(\mathcal{M}^\star \in \cdot \mid H_k)$ and distortion function $d_{\Pi,\mathcal{V}}$. Then,*

$$\text{BAYESREGRET}(K, \pi^{(1)}, \dots, \pi^{(K)}) \leq \mathbb{E}\left[ \sum_{k=1}^{K} \mathbb{E}_k \left[ V^\star_{\widetilde{\mathcal{M}}_k, 1} - V^{\pi^{(k)}}_{\widetilde{\mathcal{M}}_k, 1} \right] \right] + 2KH\sqrt{D}.$$

Theorem 1 shows how the Bayesian regret incurred by VSRL can be separated into an error term the agent must pay for learning a simplified MDP $\widetilde{\mathcal{M}}_k$, rather than $\mathcal{M}^\star$, and the Bayesian regret incurred while trying to learn $\widetilde{\mathcal{M}}_k$. This first term mirrors the satisficing regret of Russo and Van Roy [138, 139] for multi-armed bandits where the performance of the agent in the $k$th episode is being measured with respect to a compressed MDP $\widetilde{\mathcal{M}}_k$, rather than the true MDP $\mathcal{M}^\star$. While further discussion on the choices of $\Pi$ and $\mathcal{V}$ is provided later in this section, we simply note that the conditions placed upon them in Theorem 1 are an artifact of VSRL only executing optimal policies in each time period $h \in [H]$ which, under the assumptions of our problem formulation, are deterministic.

The remainder of this section is devoted to an analysis for establishing an information-theoretic bound on the satisficing regret term of Theorem 1. A central tool of our analysis will be the information ratio [136, 137] at the $k$th episode:

$$\Gamma_k \triangleq \frac{\mathbb{E}_k \left[ V^\star_{\widetilde{\mathcal{M}}_k, 1} - V^{\pi^{(k)}}_{\widetilde{\mathcal{M}}_k, 1} \right]^2}{\mathbb{I}_k(\widetilde{\mathcal{M}}_k; \tau_k, M_k)} \qquad \forall k \in [K].$$

In words, the information ratio is the ratio between squared expected regret in the $k$th episode with respect to $\widetilde{\mathcal{M}}_k$ and the information gained about $\widetilde{\mathcal{M}}_k$ in the $k$th episode by sampling MDP $M_k$ and observing trajectory $\tau_k$, given the current history $H_k$. Numerous prior works have leveraged similar or generalized types of information ratios for analyzing multi-armed bandit problems [135–139, 54, 99, 169, 36, 15, 98] as well as reinforcement-learning problems [104]; in comparison to the latter, we simply note that our analysis bears stronger resemblance to those in multi-armed bandits by not constructing confidence sets over MDPs [121, 120, 104], avoiding a restricted focus to tabular problems. That said, our results are contingent upon the existence of a uniform upper bound to the information ratios across all episodes, a non-trivial result [78] that we leave to future work.

Through our information-ratio analysis, we obtain the following information-theoretic bound on satisficing Bayesian regret:

**Theorem 2.** *If $\Gamma_k \le \overline{\Gamma}$, for all $k \in [K]$, then $\mathbb{E}\left[\sum_{k=1}^{K} \mathbb{E}_k\left[V_{\widetilde{\mathcal{M}}_k,1}^{\star} - V_{\widetilde{\mathcal{M}}_k,1}^{\pi^{(k)}}\right]\right] \le \sqrt{\overline{\Gamma} K \mathcal{R}_1^{\Pi,\mathcal{V}}(D)}$.*

An immediate consequence of the preceding theorems is the following corollary which establishes our main result, an information-theoretic Bayesian regret bound for VSRL. We omit the proof as it follows directly from applying Theorems 1 and 2 in sequence.

**Corollary 1.** *Take any $\Pi \supseteq \{\mathcal{S} \to \mathcal{A}\}$, any $\mathcal{V} \supseteq \{V^{\pi} \mid \pi \in \Pi^H\}$, and fix any $D > 0$. For any prior distribution $\mathbb{P}(\mathcal{M}^{\star} \in \cdot \mid H_1)$, if $\Gamma_k \le \overline{\Gamma}$ for all $k \in [K]$, then VSRL (Algorithm 2) has*

$$\text{BAYESREGRET}(K, \pi^{(1)}, \ldots, \pi^{(K)}) \le \sqrt{\overline{\Gamma} K \mathcal{R}_1^{\Pi,\mathcal{V}}(D)} + 2KH\sqrt{D}.$$

Once again we recall that, since the rate-distortion function is well-defined for arbitrary source and channel output random variables defined on abstract alphabets [43], the Bayesian regret bound of Corollary 1 holds for any finite-horizon, episodic MDP, extending beyond past analyses of PSRL constrained only to tabular MDPs. We defer a discussion of practical considerations for implementing VSRL to the appendix.

At this point, we call attention to the parameterization of our lossy compression problem by a particular policy class $\Pi$ and value function class $\mathcal{V}$, whose dependence we inherit from the value equivalence principle [72]. The next result clarifies how the performance of VSRL is affected by fluctuations in these classes via a dominance relationship [150] between the induced distortion functions.

**Lemma 1.** *For any two $\Pi, \Pi'$ and any $\mathcal{V}, \mathcal{V}'$ such that $\Pi' \subseteq \Pi \subseteq \{\mathcal{S} \to \Delta(\mathcal{A})\}$ and $\mathcal{V}' \subseteq \mathcal{V} \subseteq \{\mathcal{S} \to \mathbb{R}\}$, we have $\mathcal{R}_k^{\Pi,\mathcal{V}}(D) \ge \mathcal{R}_k^{\Pi',\mathcal{V}'}(D)$, $\forall k \in [K], D > 0$.*

Property 3 of Grimm et al. [72] highlights how the set of value-equivalent MDPs grows as the policy and value function classes shrink. Lemma 1 provides an intuitive, information-theoretic counterpart to their result where, as the sets of policies and value functions over which models will be assessed diminish, an agent may naturally compress more aggressively and throw away larger quantities of bits from each source distribution over the true MDP $\mathcal{M}^{\star}$.

Since a compressed MDP $\widetilde{\mathcal{M}}_k$ that achieves the rate-distortion limit has *expected* distortion bounded by $D$, one may wonder how the probability of not recovering an approximately-value-equivalent MDP scales as $D \uparrow \infty$. To that end, we conclude this section with a final result that brings clarity to this via a generalization [60] of Fano's inequality [64]. We leave investigation of other generalizations of Fano's inequality that might yield similarly interesting results to future work [160, 8].

**Lemma 2.** *Take any $\Pi \subseteq \{\mathcal{S} \to \Delta(\mathcal{A})\}$ and $\mathcal{V} \subseteq \{\mathcal{S} \to \mathbb{R}\}$. For any $D \ge 0$ and any $k \in [K]$, define $\delta = \sup_{\widehat{M} \in \mathfrak{M}} \mathbb{P}(d_{\Pi,\mathcal{V}}(\mathcal{M}^{\star}, \widehat{M}) \le D \mid H_k)$. Then,*

$$\sup_{\widetilde{\mathcal{M}} \in \Lambda_k(D)} \mathbb{P}(d_{\Pi,\mathcal{V}}(\mathcal{M}^{\star}, \widetilde{\mathcal{M}}) > D \mid H_k) \ge 1 - \frac{\mathcal{R}_k^{\Pi,\mathcal{V}}(D) + \log(2)}{\log\left(\frac{1}{\delta}\right)}.$$

For any episode $k \in [K]$, the left-hand side of the inequality in Lemma 2 denotes the worst-case error probability of sampling a compressed MDP $\widetilde{\mathcal{M}}$ that is not approximately-value-equivalent to $\mathcal{M}^{\star}$. The right-hand side conveys that, in order to avoid such an error with reasonable probability, one requires a setting of $D < \infty$ such that $\mathcal{R}_k^{\Pi,\mathcal{V}}(D) \approx \log\left(\frac{1}{\delta}\right)$.

## 5.2 Specializing to Tabular MDPs

While the preceding subsection constitutes the main contribution of this paper, the presence of information-theoretic terms makes it difficult to compare our guarantees to those obtained in prior work, which typically focuses on the tabular setting. To help remedy this, we offer the following theorem which restricts focus to the case where the agent pursues an exactly value-equivalent model of the tabular environment. Notably, the results of this section still retain a dependence on a uniform upper bound to the information ratio whose exact form is a result left to future work.

**Theorem 3.** *Take any $\Pi \supseteq \{\mathcal{S} \to \mathcal{A}\}$, any $\mathcal{V} \supseteq \{V^{\pi} \mid \pi \in \Pi^H\}$, and let $D = 0$. For any prior distribution $\mathbb{P}(\mathcal{M}^{\star} \in \cdot \mid H_1)$ over tabular MDPs, if $\Gamma_k \le \overline{\Gamma}$ for all $k \in [K]$, then VSRL (Algorithm 2) has $\text{BAYESREGRET}(K, \pi^{(1)}, \ldots, \pi^{(K)}) \le \mathcal{O}\left(|\mathcal{S}|\sqrt{\overline{\Gamma}|\mathcal{A}|K}\right)$.*

An immediate observation is that the Bayesian regret bound of Theorem 3 matches the dependence on the number of states, $|\mathcal{S}|$, obtained in the first (weaker) guarantee established for PSRL by Osband et al. [121]; we suspect that this guarantee for VSRL is unimprovable without further distributional assumptions [120, 119]. As an alternative, we contemplate how a change in the distortion measure used by VSRL might incur an improved regret bound when specialized to the tabular setting.

Specifically, notice that the only piece of the VSRL analysis tethered to the particular form of the distortion function $d_{\Pi,\mathcal{V}}(\mathcal{M}, \widehat{\mathcal{M}})$ is Theorem 1, while all other components remain agnostic to the precise criterion for assessing the loss of fidelity between original and compressed MDPs. Consequently, there is potential for a modified distortion function to offer an improved regret analysis relative to Theorem 3. Rather than concerning ourselves with planning over multiple behaviors, we consider a distortion function based solely on the optimal action-value functions:

$$d_{Q^\star}(\mathcal{M}, \widehat{\mathcal{M}}) = \sup_{h \in [H]} ||Q^\star_{\mathcal{M},h} - Q^\star_{\widehat{\mathcal{M}},h}||^2_\infty = \sup_{h \in [H]} \sup_{(s,a) \in \mathcal{S} \times \mathcal{A}} |Q^\star_{\mathcal{M},h}(s,a) - Q^\star_{\widehat{\mathcal{M}},h}(s,a)|^2.$$

We use $\mathcal{R}^{Q^\star}_k(D)$ to denote the rate-distortion function under this new measure of distortion, $d_{Q^\star}(\mathcal{M}, \widehat{\mathcal{M}})$. In order for this new distortion function to be compatible with VSRL, we require an analogue to the regret decomposition of Theorem 1.

**Theorem 4.** *Fix any $D \geq 0$ and, for each episode $k \in [K]$, let $\widetilde{\mathcal{M}}_k$ be any MDP that achieves the rate-distortion limit of $\mathcal{R}^{Q^\star}_k(D)$ with information source $\mathbb{P}(\mathcal{M}^\star \in \cdot \mid H_k)$ and distortion function $d_{Q^\star}$.*

*Then,* $\text{BAYESREGRET}(K, \pi^{(1)}, \ldots, \pi^{(K)}) \leq \mathbb{E}\left[ \sum_{k=1}^K \mathbb{E}_k \left[ V^\star_{\widetilde{\mathcal{M}}_k,1} - V^{\pi^{(k)}}_{\widetilde{\mathcal{M}}_k,1} \right] \right] + 2K(H+1)\sqrt{D}.$

With this regret decomposition in hand, we may recover the analogue to Corollary 1, whose proof is immediate and, therefore, omitted.

**Corollary 2.** *Fix any $D > 0$. For any prior distribution $\mathbb{P}(\mathcal{M}^\star \in \cdot \mid H_1)$, if $\Gamma_k \leq \overline{\Gamma}$ for all $k \in [K]$, then VSRL (Algorithm 2) with distortion function $d_{Q^\star}$ has $\text{BAYESREGRET}(K, \pi^{(1)}, \ldots, \pi^{(K)}) \leq$*
$\sqrt{\overline{\Gamma} K \mathcal{R}^{Q^\star}_1(D)} + 2K(H+1)\sqrt{D}.$

As illustrated by the following lemma, the significance of this change in distortion measure from $d_{\Pi,\mathcal{V}}$ to $d_{Q^\star}$ is that the optimal action-value functions may now act as an information bottleneck [158] between the original MDP $\mathcal{M}^\star$ and compressed MDP $\widetilde{\mathcal{M}}_k$.

**Lemma 3.** *For each episode $k \in [K]$ and for $D = 0$, let $\widetilde{\mathcal{M}}_k$ be any MDP that achieves the rate-distortion limit of $\mathcal{R}^{Q^\star}_k(D)$ with information source $\mathbb{P}(\mathcal{M}^\star \in \cdot \mid H_k)$ and distortion function $d_{Q^\star}$. Then, we have the Markov chain $\mathcal{M}^\star \to Q^\star_{\mathcal{M}^\star} \to \widetilde{\mathcal{M}}_k$, where $Q^\star_{\mathcal{M}^\star} = \{Q^\star_{\mathcal{M}^\star,h}\}_{h \in [H]}$ is the collection of random variables denoting the optimal action-value functions of $\mathcal{M}^\star$.*

Lemma 3, through the data-processing inequality, immediately leads us to an analogue of Theorem 3 that matches the dependence on $|\mathcal{S}|$ in the best known Bayesian regret bound for PSRL [120].

**Theorem 5.** *For $D = 0$ and any prior distribution $\mathbb{P}(\mathcal{M}^\star \in \cdot \mid H_1)$ over tabular MDPs, if $\Gamma_k \leq \overline{\Gamma}$ for all $k \in [K]$, then VSRL with distortion function $d_{Q^\star}$ has $\text{BAYESREGRET}(K, \pi^{(1)}, \ldots, \pi^{(K)}) \leq$*
$\widetilde{\mathcal{O}}\left( \sqrt{\overline{\Gamma} |\mathcal{S}||\mathcal{A}| K H} \right).$

Ultimately, Theorem 5 confirms that while there is great flexibility in the original definition of value equivalence to support planning across multiple policies and value functions, focusing on optimal value functions gives rise to more efficient learning. Moreover, comparing the result with the PSRL regret bound of Osband and Van Roy [120] for tabular MDPs, this suggests an achievable uniform upper bound to the information ratio as $\overline{\Gamma} \lesssim H^2$, where the $\lesssim$ accounts for numerical constants and logarithmic factors.

## 5.3 Capacity-Sensitive Performance Guarantees

We recognize that the information-theoretic regret bounds of the previous two sections, like many other guarantees for provably-efficient reinforcement learning before them, implicitly and unrealistically

assume that an agent is of unbounded capacity and may pursue any approximately-value-equivalent model under a given distortion threshold $D$. In the context of real-world reinforcement learning [62, 105], however, fundamental limits on computational resources and time leave an agent designer with a bounded agent to be deployed within an overwhelmingly complex environment. As such, this designer may seldom be in a position to dictate an ideal or desired sub-optimality threshold $D$, but rather must make do with a known constraint on agent capacity; guarantees on sample-efficient reinforcement learning cognizant of such a fundamental constraint are nascent.

While there are numerous possibilities for how one might choose to formally characterize agent capacity, we here adopt a fundamental perspective that learning is the process of acquiring information and so take this capacity to imply the existence of a non-negative real value $R \in \mathbb{R}_{>0}$ such that the agent may only acquire and retain exactly $R$ bits of information. To help contextualize this notion of agent capacity, we introduce the distortion-rate function [146, 25, 41] which quantifies the fundamental limit of expected distortion under an information constraint:

$$\mathcal{D}_k^{Q^\star}(R) = \inf_{\widetilde{\mathcal{M}} \in \Upsilon_k(R)} \mathbb{E}_k \left[ d_{Q^\star}(\mathcal{M}^\star, \widetilde{\mathcal{M}}) \right] \qquad \mathcal{D}_k^{Q^\star}(R) = \inf_{\widetilde{\mathcal{M}} \in \Upsilon_k(R)} \mathbb{E}_k \left[ d_{Q^\star}(\mathcal{M}^\star, \widetilde{\mathcal{M}}) \right], \quad (2)$$

where the infimum is taken over all channels with bounded rate, $\Upsilon_k(R) \triangleq \{\widetilde{\mathcal{M}} : \Omega \to \mathfrak{M} \mid \mathbb{I}_k(\mathcal{M}^\star; \widetilde{\mathcal{M}}) \le R\}$. In words, given the agent's current beliefs over the true MDP $\mathbb{P}(\mathcal{M}^\star \in \cdot \mid H_k)$, the infimum of the distortion-rate function is taken over all potential lossy compressions of the environment that fall within the agent's capacity constraint of $R$ bits and identifies the one that preserves the most useful information, as measured by the distortion function. Conveniently, the rate-distortion function and distortion-rate function are inverses of one another [41] $(\mathcal{R}(\mathcal{D}(R)) = R)$ such that we recover the following two capacity-sensitive regret bounds directly from Corollaries 1 and 2 by simply taking the input distortion threshold of VSRL equal to the associated distortion-rate function in the first episode ($D = \mathcal{D}_1^{\Pi,\mathcal{V}}(R)$ and $D = \mathcal{D}_1^{Q^\star}(R)$, respectively).

**Corollary 3.** *Take any* $\Pi \supseteq \{\mathcal{S} \to \mathcal{A}\}$*, any* $\mathcal{V} \supseteq \{V^\pi \mid \pi \in \Pi^H\}$*, and let* $R > 0$ *be the agent capacity. For any prior distribution* $\mathbb{P}(\mathcal{M}^\star \in \cdot \mid H_1)$*, if* $\Gamma_k \le \overline{\Gamma}$ *for all* $k \in [K]$*, then VSRL (Algorithm 2) with distortion function* $d_{\Pi,\mathcal{V}}$ *has* BAYESREGRET$(K, \pi^{(1)}, \dots, \pi^{(K)}) \le \sqrt{\overline{\Gamma}KR} + 2KH\sqrt{\mathcal{D}_1^{\Pi,\mathcal{V}}(R)}$.

**Corollary 4.** *Let* $R > 0$ *be the agent capacity. For any prior distribution* $\mathbb{P}(\mathcal{M}^\star \in \cdot \mid H_1)$*, if* $\Gamma_k \le \overline{\Gamma}$ *for all* $k \in [K]$*, then VSRL (Algorithm 2) with distortion function* $d_{Q^\star}$ *has* BAYESREGRET$(K, \pi^{(1)}, \dots, \pi^{(K)}) \le \sqrt{\overline{\Gamma}KR} + 2K(H+1)\sqrt{\mathcal{D}_1^{Q^\star}(R)}$.

Turning back to Example 1, note how an agent with significantly limited capacity cannot possibly hope to capture all the granularity contained in the entire MDP sequence $\{\mathcal{M}_n\}_{n \in [N]}$, for large values of $N$. For a capacity of exactly $R$ bits, Corollaries 3 and 4 immediately translate this fundamental limit into a corresponding performance guarantee, allowing the agent to identify a subsequence $\{\mathcal{M}_n\}_{n \in [C]}$ for some $C \ll N$ which only requires gathering $R$ bits of information from the environment.

# 6 Conclusion

In this paper, we began with a finite-horizon, episodic MDP and considered the ramifications of a real-world reinforcement-learning scenario wherein the relative complexity of the environment is so immense that an agent may find itself incapable of perfectly recovering optimal behavior. An immediate consequence of this reality is the need to strike an appropriate balance between what is performant and what is achievable. We introduced the VSRL algorithm for incrementally synthesizing *simple* and *useful* approximations of the environment from which an agent might still recover near-optimal behaviors. Recognizing the information-theoretic nature of this lossy MDP compression, we provided an analysis of VSRL whose performance guarantees, by virtue of rate-distortion theory, are twofold. The first set of guarantees ensure VSRL recovers the simplest compression of the environment which still incurs bounded sub-optimality, as specified by the agent designer. Alternatively, the second set of guarantees maintain that VSRL finds the best compression of the environment subject to constraints on agent capacity. Through our general problem formulation and information-theoretic analysis, both regret bounds hold for any finite-horizon, episodic MDP, regardless of whether or not the state-action space is finite. That said, the question of how to practically instantiate VSRL for high-dimensional settings is an open problem left to future work.

## Acknowledgements

The authors gratefully acknowledge Christopher Grimm for initial discussions that provided an impetus for this work. The authors also thank Adithya Devraj, Shi Dong, John Duchi, Hong Jun Jeon, Saurabh Kumar, and Xiuyuan (Lucy) Lu for insightful comments on various components of the paper. Financial support from Army Research Office (ARO) grant W911NF2010055 is gratefully acknowledged.

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
