# A Related Work

This paper follows suit with a long line of work on provably-efficient reinforcement learning [91, 35, 90, 20, 23, 151, 81, 121, 45, 120, 22, 46, 10, 85, 168, 56, 105]. As previously discussed, these methods can be categorized based on their use of optimism in the face of uncertainty or posterior sampling to address the exploration challenge. Notably, methods in the latter category are Bayesian reinforcement-learning algorithms [70] that, through their use of Thompson sampling [156, 141], are exclusively concerned with identifying optimal solutions. The notable exception to this statement is the method of Lu et al. [105], which is based on information-directed sampling [135, 137]; while their analysis does accommodate other learning targets besides the optimal policy, an agent designer is responsible for supplying this target to the agent a priori whereas we adaptively compute an information-theoretically sound target grounded in rate-distortion theory.

In contrast to this class of approaches, optimism-based methods tend to obey PAC-MDP guarantees [90, 151] which, given a fixed parameter $\varepsilon > 0$, offer a high-probability bound on the total number of timesteps for which the agent's behavior is worse than $\varepsilon$-sub-optimal. Through this tolerance parameter $\varepsilon$, an agent designer can express a preference for efficiently identifying a deliberately sub-optimal solution; our work can be seen as providing an analogous knob for Bayesian reinforcement-learning methods that deliberately pursue a satisficing solution while also remaining competitive with regret guarantees for optimism-based methods [45, 46, 85, 168]. In this way, our theoretical guarantees are more general than those for PSRL [121, 118, 2, 120, 10]. Importantly, the nature of our contribution is not to be confused with the PAC-BAMDP framework of Kolter and Ng [94] which characterizes algorithms that adhere to a high-probability bound on the total number of sub-optimal timesteps relative to the Bayes-optimal policy [18, 149]. We refer readers to the work of Ghavamzadeh et al. [70] for a broader survey of Bayesian reinforcement-learning methods, including those which do not employ posterior sampling [152], but instead entertain other approximations [51, 52, 163, 39, 12, 74–76] to tractably solve the resulting Bayes-Adaptive Markov Decision Process (BAMDP) [61], typically while foregoing rigorous theoretical guarantees.

A perhaps third distinct class of provably-efficient reinforcement-learning algorithms [96, 84, 47, 58, 153] proceeds by iteratively selecting an element of a function class (typically denoting a collection of regressors for either a value function or transition model), inducing a policy from the chosen function, and then carefully eliminating all hypotheses of the function class that are inconsistent with the observed data resulting from policy rollouts in the environment. To the extent that one might be willing to characterize this high-level algorithmic template as an iterative, manual compression and refinement of the initial function class, our algorithm can be seen as bringing the appropriate tool of rate-distortion theory to bear on the inherent lossy compression problem and developing the complementary information-theoretic analysis.

The concept of designing algorithms to learn such near-optimal or satisficing solutions has been well-studied in the multi-armed bandit setting [38, 100]. Indeed, the need to forego optimizing for an optimal arm arises naturally in various contexts [37, 92, 134, 142, 53, 27, 164, 32]. A general study of such satisficing solutions through the lens of information theory was first proposed by Russo et al. [140], Russo and Van Roy [138, 139] and later extended to develop practical algorithms by Arumugam and Van Roy [15, 16]. Our work provides the natural, theoretical extension of these ideas to the full reinforcement-learning setting, leaving investigation of practical instantiations to future work (see Section B). The algorithm and regret bound we provide bears some resemblance to the compressed Thompson sampling algorithm of Dong and Van Roy [54] for bandit problems. Crucially, while the compressive statistic of the environment utilized by their algorithm is computed once a priori, our algorithm recomputes its learning target in each episode, refining it as the agent's knowledge of the true environment accumulates. Similar to these prior works, we leverage rate-distortion theory [146] as a principled tool for a mathematically-precise characterization of satisficing solutions. We simply note that our use of rate-distortion theory for reinforcement learning in this work stands in stark contrast to that of prior work which examines state abstraction in reinforcement learning [6] or attempts to control the entropy of the resulting policy [157, 133, 145].

We also recognize the connection between this work and prior work at the intersection of information theory and control theory [166, 108, 109, 33, 155, 95]. These works parallel our setting in their consideration for an agent that must stabilize a system with limited *observational* capacity, augmenting the standard control objective subject to a constraint on the rate of the channel that processes raw observations; this problem formulation more closely aligns with a partially-observable Markov

Decision Process [19, 88] or an agent learning with a state abstraction [101, 4, 159]. In contrast, our work is concerned with an overall limit on the total amount of information an agent may acquire from the environment and, in turn, how that translates into its selection of a feasible learning target. That said, we suspect there could be a strong, subtle synergy between these prior works and the capacity-sensitive performance guarantees for our algorithm (see Section 5.3).

## B   Discussion

In this section, we outline connections between VSRL and follow-up work to the value equivalence principle [73], explore opportunities for even further compression through state abstraction [101, 4], and contemplate potential avenues for how our theory might inform practice.

### B.1   Proper Value Equivalence

While the value equivalence principle examines a single application of each Bellman operator, in follow-up work Grimm et al. [73] introduce the notion of proper value equivalence, which considers the limit of infinitely many applications or, stated more concisely, the fixed points of the associated operators. A model $\widetilde{\mathcal{M}}$ is proper value equivalent if $V^\pi_{\mathcal{M}^\star,h} = V^\pi_{\widetilde{\mathcal{M}},h}$, $\forall \pi \in \Pi$, $h \in [H]$. This notion allows for a simpler parameterization through the policy class $\Pi$ alone, without the need for a complementary value function class $\mathcal{V}$. Conveniently, through Proposition 2 of Grimm et al. [73], it follows that to obtain the set of proper value equivalent models with respect to $\Pi$, one need only find the set of models that are value equivalent for each $\pi \in \Pi$ and its induced value function, $V^\pi$. In our context, we can establish an approximate version of this by using the distortion function $d_{\Pi,\mathcal{V}}$ where $\mathcal{V} = \{V^\pi \mid \pi \in \Pi^H\}$ (recall that previous results obeyed the less stringent condition that $\mathcal{V} \supseteq \{V^\pi \mid \pi \in \Pi^H\}$).

Grimm et al. [73] go on to study proper value equivalence for the set of all deterministic policies, $\Pi = \{\mathcal{S} \to \mathcal{A}\}$ and, through their Corollary 1, show that an optimal policy for any model which is proper value equivalent to $\Pi$ is also optimal in the original MDP $\mathcal{M}^\star$. Again, we recall that our prior guarantees were made under the less restrictive assumption that $\Pi \supseteq \{\mathcal{S} \to \mathcal{A}\}$. Coupling these insights on proper value equivalence together, we see that when VSRL is run with $\Pi = \{\mathcal{S} \to \mathcal{A}\}$ and $\mathcal{V} = \{V^\pi \mid \pi \in \Pi^H\}$, the agent aims to recover an approximately proper value-equivalent model of the true environment and, when $D = 0$, the optimal policy associated with this compressed MDP will be optimal for $\mathcal{M}^\star$. Finally, through their Proposition 5, Grimm et al. [73] identify the set of all proper value equivalent models with respect to $\{\mathcal{S} \to \mathcal{A}\}$ as the largest possible value equivalence class that is guaranteed to yield optimal performance in the true environment. Meanwhile, our Lemma 1 again establishes the information-theoretic analogue of this claim; namely, that VSRL configured to learn a model from this largest value equivalence class requires the fewest bits of information from the true environment. The importance of proper value equivalence culminates with experiments that highlight how MuZero [144] succeeds by optimizing a proper value-equivalent loss function. We leave to future work the question of how VSRL might pave the way towards more principled exploration strategies for practical algorithms like MuZero.

### B.2   Greater Compression via State Abstraction

A core disconnect between VSRL and contemporary deep model-based reinforcement learning approaches is that our lossy compression problem forces VSRL to identify a model defined with respect to the original state space whereas methods in the latter category learn a model with respect to a state abstraction. Indeed, algorithms like MuZero and its predecessors [147, 116, 144] never approximate reward functions and transition models with respect to the raw image observations generated by the environment, but instead incrementally learn some latent representation of state upon which a corresponding model is approximated for planning. This philosophy is born out of several years of work that elucidate the important of state abstraction as a key tool for avoiding the irrelevant information encoded in environment states and addressing the challenge of generalization for sample-efficient reinforcement learning large-scale environments [165, 30, 50, 67, 86, 101, 159, 68, 83, 4–6, 55, 58, 14, 107, 9, 7, 3, 56]. In this section, we briefly introduce a small extension of VSRL that builds on these insights to accommodate lossy MDP compressions defined on a simpler, abstract state space (also referred to as aleatoric or situational state by Lu et al. [105], Dong et al. [56]).

Let $\Phi \subseteq \{\mathcal{S} \rightarrow [Z]\}$ denote a class of state abstractions or quantizers which map environment states to some discrete, finite abstract state space containing a known, fixed number of abstract states $Z \in \mathbb{N}$. For any abstract action-value function $Q_\phi \in \{[Z] \times \mathcal{A} \rightarrow \mathbb{R}\}$ and any state abstraction $\phi \in \Phi$, we denote by $Q_\phi \circ \phi \in \{\mathcal{S} \times \mathcal{A} \rightarrow \mathbb{R}\}$ the composition of the state abstraction and abstract value function such that $Q_\phi \circ \phi$ is a value function for the original MDP. We adopt a similar convention for any policy $\pi_\phi \in \{[Z] \rightarrow \Delta(\mathcal{A})\}$ such that $\pi_\phi \circ \phi \in \{\mathcal{S} \rightarrow \Delta(\mathcal{A})\}$. We now consider carrying out the rate-distortion optimization of VSRL in each episode over abstract MDPs such that $\widetilde{\mathcal{M}}_k \in \mathfrak{M}_\phi \triangleq \{[Z] \times \mathcal{A} \rightarrow [0,1]\} \times \{[Z] \times \mathcal{A} \rightarrow \Delta([Z])\}$. Just as before, we take the input information source to our lossy compression problem in each episode $k \in [K]$ as the agent's current beliefs over the true MDP, $\mathbb{P}(\mathcal{M}^\star \in \cdot \mid H_k)$. Unlike the preceding sections, our distortion function $d : \mathfrak{M} \times \mathfrak{M}_\phi \rightarrow \mathbb{R}_{\geq 0}$ must now quantify the loss of fidelity incurred by using a compressed abstract MDP in lieu of the true environment MDP. Consequently, we define a new distortion function

$$d_\Phi(\mathcal{M}, \widehat{\mathcal{M}}) = \sup_{\phi \in \Phi} \sup_{h \in [H]} ||Q^\star_{\mathcal{M},h} - Q^\star_{\widehat{\mathcal{M}},h} \circ \phi||^2_\infty = \sup_{\phi \in \Phi} \sup_{h \in [H]} \max_{(s,a) \in \mathcal{S} \times \mathcal{A}} |Q^\star_{\mathcal{M},h}(s,a) - Q^\star_{\widehat{\mathcal{M}},h}(\phi(s),a)|^2,$$

whose corresponding rate-distortion function is given by

$$\mathcal{R}^\Phi_k(D) = \inf_{\widetilde{\mathcal{M}} \in \Lambda_k(D)} \mathbb{I}_k(\mathcal{M}^\star; \widetilde{\mathcal{M}}) \qquad \Lambda_k(D) \triangleq \{\widetilde{\mathcal{M}} : \Omega \rightarrow \mathfrak{M} \mid \mathbb{E}_k\left[d_\Phi(\mathcal{M}^\star, \widetilde{\mathcal{M}})\right] \leq D\}.$$

Unlike when performing a lossy compression where $\widetilde{\mathcal{M}}_k \in \mathfrak{M}$, the channel that represents the identity mapping is no longer a viable option as we must now generate an abstract MDP that resides in $\mathfrak{M}_\phi$. Consequently, we require the following assumption on $\Phi$ to ensure that the set of channels over which we compute the infimum of $\mathcal{R}^\Phi_k(D)$ is non-empty.

**Assumption 1.** *For each $k \in [K]$, we have that $\Lambda_k(D) \neq \emptyset$.*

---

**Algorithm 3** Compressed Value-equivalent Sampling for Reinforcement Learning (Compressed-VSRL)

---

**Input:** Prior distribution $\mathbb{P}(\mathcal{M}^\star \in \cdot \mid H_1)$, Distortion threshold $D \in \mathbb{R}_{\geq 0}$, State abstraction class $\Phi$, Distortion function $d_\Phi : \mathfrak{M} \times \mathfrak{M}_\phi \rightarrow \mathbb{R}_{\geq 0}$,
**for** $k \in [K]$ **do**
    Compute channel $\mathbb{P}(\widetilde{\mathcal{M}}_k \in \cdot | \mathcal{M}^\star)$ achieving $\mathcal{R}^\Phi_k(D)$ limit
    Sample MDP $M^\star \sim \mathbb{P}(\mathcal{M}^\star \in \cdot \mid H_k)$
    Sample compressed MDP $M_k \sim \mathbb{P}(\widetilde{\mathcal{M}}_k \in \cdot \mid \mathcal{M}^\star = M^\star)$
    Set state abstraction $\phi_k$ to achieve the infimum: $\inf_{\phi \in \Phi} \sup_{h \in [H]} ||Q^\star_{M^\star,h} - Q^\star_{M_k,h} \circ \phi||^2_\infty$
    Compute optimal policy $\pi^\star_{M_k}$ and set $\pi^{(k)} = \pi^\star_{M_k} \circ \phi_k$
    Execute $\pi^{(k)}$ and observe trajectory $\tau_k$
    Update history $H_{k+1} = H_k \cup \tau_k$
    Induce posterior $\mathbb{P}(\mathcal{M}^\star \in \cdot \mid H_{k+1})$
**end for**

---

We present our Compressed-VSRL extension as Algorithm 3 which incorporates an additional step beyond VSRL to govern the choice of state abstraction utilized in conjunction with the sampled compressed MDP in each episode.

We strongly suspect that an analysis paralleling that of Corollaries 1 and 2, with an appropriately defined information ratio, can be carried out for Compressed-VSRL as well. However, for the sake of brevity and since the result is neither immediate nor trivial, we leave the information-theoretic regret bound stated as a conjecture.

**Conjecture 1.** *Fix any $D > 0$. For any prior distribution $\mathbb{P}(\mathcal{M}^\star \in \cdot \mid H_1)$, if $\Gamma_k \leq \overline{\Gamma}$ for all $k \in [K]$, then CVSRL (Algorithm 3) with distortion function $d_\Phi$ has*

$$\text{BAYESREGRET}(K, \pi^{(1)}, \ldots, \pi^{(K)}) \leq \sqrt{\overline{\Gamma} K \mathcal{R}^\Phi_1(D)} + 2K(H+1)\sqrt{D}.$$

The significance of Conjecture 1 for allowing a simple, bounded agent to contend with a complex environment manifests when considering analogues to Theorems 3 and 5. Specifically, for any finite-horizon, episodic MDP with a finite action space ($|\mathcal{A}| < \infty$), one may upper bound the rate-distortion function via the entropy in the abstract model $\mathbb{H}_1(\mathcal{R}_\phi, \mathcal{T}_\phi)$. Using the same proof technique as in the preceding results, this facilitates an upper bound $\mathcal{R}_1^\Phi(D) \leq \widetilde{\mathcal{O}}\left(Z^2|\mathcal{A}|\right)$ which lacks any dependence on the complexity of the (potentially infinite) environment state space, $\mathcal{S}$.

## B.3  From Theory to Practice

While the performance guarantees of VSRL hold for any finite-horizon, episodic MDP, it is important to reconcile that generality with the practicality of the instantiating the algorithm. The three key barriers to practical, scalable implementations of VSRL applied to complex tasks of interest are the representation of epistemic uncertainty, the computation of the rate-distortion function, and the synthesis of optimal policies for sampled MDPs. The first point is a fundamental obstacle to Bayesian reinforcement-learning algorithms and recent work in deep reinforcement learning has found success with simple, albeit computationally-inefficient, ensembles of networks [122, 103, 124] or even hypermodels [63]. As progress is made towards more computationally-efficient models for representing and resolving epistemic uncertainty through Bayesian deep learning [126, 127], there will be greater potential for a practical implementation of VSRL.

For addressing the second issue, a classic option for computing the channel that achieves the rate-distortion limit is the Blahut-Arimoto algorithm [31, 13] which, in theory, is a well-defined procedure even for random variables defined on abstract alphabets [43, 42]. In practice, however, computing such a channel for continuous outputs remains an open challenge [48]; still, several analyses and refinements have been made to the algorithm so far [34, 132, 143, 106, 40, 112, 162, 111, 167], and the reinforcement-learning community stands to greatly benefit from further improvements. Continuous information sources, however, are less problematic as one may draw a sufficiently large number of i.i.d. samples and substitute this empirical distribution for the source, leading to the so-called plug-in estimator of the rate-distortion function for which consistency and sample-complexity guarantees are known [79, 128]. Moreover, empirical successes for such estimators have already been demonstrated in the multi-armed bandit setting [15, 16].

The last issue touches upon the fact that while tabular problems admit several planning algorithms for recovering the optimal policy associated with the sampled MDP in each episode, the same cannot be said for arbitrary state-action spaces. At best, one might hope for simply recovering an approximation to this policy through some high-dimensional model-based planning algorithm. We leave the questions of how to practically implement such a procedure and understand its impact on our theory to future work.

Of course, all of the aforementioned issues arise when trying to directly implement VSRL roughly as described by Algorithm 2. An alternative, however, is to ask how one might take existing practical algorithms already operating at scale (such as MuZero [144]) and bring those methods closer to the spirit of VSRL? Since these practical model-based reinforcement-learning algorithms are already engaging with some form of state abstraction [101, 4, 159], this might entail further consideration for information-theoretic approaches to guiding representation learning [6, 145] as a proxy to engaging with a rate-distortion trade-off. Additionally, one of the core insights developed in this work is the formalization of model simplifications arising out of the value equivalence principle as a form of lossy compression. Curiously, recent meta reinforcement-learning approaches applied to multi-task or contextual MDPs [77] arrive at a similar need of obtaining compressed representations of the underlying MDP model during a meta-exploration phase in order to facilitate few-shot learning [131, 170, 102]. Given such approaches for learning latent/contextual variational encoders, we strongly suspect that the known connections between rate-distortion theory and information bottlenecks [158, 11] represent a viable path to bridging the ideas between our theory and value equivalence in practice. Notably, while these methods adopt a probabilistic inference perspective and do articulate the encoders as posterior distributions over the underlying task, they lack any representation of epistemic uncertainty [115], similar to MuZero; consequently, this still leaves open the earlier obstacle of how best to represent and maintain notions of epistemic uncertainty in large-scale agents.

A third competing perspective is to recognize that recent empirical successes in Bayesian reinforcement learning often avoid representing uncertainty over the model of the environment in favor of the underlying optimal action-value functions, $\{Q^\star_{\mathcal{M}^\star, h}\}_{h \in [H]}$. Such approaches apply an algorithm

known as Randomized Value Functions (RVF), rather than PSRL [122–124, 114, 125]. Naturally, the optimal action-value functions of the true MDP $\mathcal{M}^\star$ and its model are related and, in fact, there is an equivalence between RVF and PSRL [117]. By maintaining epistemic uncertainty over the action-value functions $\mathbb{P}(\{Q^\star_{\mathcal{M}^\star, h}\}_{h \in [H]} \in \cdot \mid H_k)$ rather than the underlying model, RVF methods are amenable to practical instantiation with deep neural networks [122, 124, 114, 126]. Moreover, extensions of this line of work have gone on to consider other scalable avenues for leveraging such principled, practical solutions to the exploration challenge through successor features [49, 82] and temporal-difference errors [69]. Overall, we strongly suspect that VSRL can also play an analogous role to PSRL in this sense, providing a sound theoretical foundation that gives rise to subsequent practical algorithms of a slightly different flavor.

## C  Proof of Theorem 1

Before we can prove Theorem 1, we require the following lemma whose proof we adapt from Osband et al. [121]:

**Lemma 4.** *Let $\mathcal{M}, \widehat{\mathcal{M}}$ be two arbitrary finite-horizon, episodic MDPs with models $(\mathcal{R}, \mathcal{T})$ and $(\widehat{\mathcal{R}}, \widehat{\mathcal{T}})$, respectively. Then, for any non-stationary policy $\pi = (\pi_1, \ldots, \pi_H) \in \{\mathcal{S} \to \Delta(\mathcal{A})\}^H$,*

$$V^\pi_{\mathcal{M},1} - V^\pi_{\widehat{\mathcal{M}},1} = \sum_{h=1}^H \mathbb{E}\left[\mathcal{B}^{\pi_h}_{\mathcal{M}} V^\pi_{\mathcal{M},h+1}(s_h) - \mathcal{B}^{\pi_h}_{\widehat{\mathcal{M}}} V^\pi_{\mathcal{M},h+1}(s_h)\right] = \sum_{h=1}^H \mathbb{E}\left[\mathcal{B}^{\pi_h}_{\mathcal{M}} V^\pi_{\widehat{\mathcal{M}},h+1}(s_h) - \mathcal{B}^{\pi_h}_{\widehat{\mathcal{M}}} V^\pi_{\widehat{\mathcal{M}},h+1}(s_h)\right].$$

*Proof.* By simply applying the Bellman equations, we have

$$
\begin{aligned}
V^\pi_{\mathcal{M},1} - V^\pi_{\widehat{\mathcal{M}},1} &= \mathbb{E}\left[V^\pi_{\mathcal{M},1}(s_1) - V^\pi_{\widehat{\mathcal{M}},1}(s_1)\right] \\
&= \mathbb{E}\left[\mathcal{B}^{\pi_1}_{\mathcal{M}} V^\pi_{\mathcal{M},2}(s_1) - \mathcal{B}^{\pi_1}_{\widehat{\mathcal{M}}} V^\pi_{\widehat{\mathcal{M}},2}(s_1)\right] \\
&= \mathbb{E}\left[\mathcal{B}^{\pi_1}_{\mathcal{M}} V^\pi_{\mathcal{M},2}(s_1) - \mathcal{B}^{\pi_1}_{\widehat{\mathcal{M}}} V^\pi_{\mathcal{M},2}(s_1) + \mathcal{B}^{\pi_1}_{\widehat{\mathcal{M}}} V^\pi_{\mathcal{M},2}(s_1) - \mathcal{B}^{\pi_1}_{\widehat{\mathcal{M}}} V^\pi_{\widehat{\mathcal{M}},2}(s_1)\right] \\
&= \mathbb{E}\left[\mathcal{B}^{\pi_1}_{\mathcal{M}} V^\pi_{\mathcal{M},2}(s_1) - \mathcal{B}^{\pi_1}_{\widehat{\mathcal{M}}} V^\pi_{\mathcal{M},2}(s_1) + \mathbb{E}_{s_2 \sim \widehat{\mathcal{T}}(\cdot|s_1,a_1)}\left[V^\pi_{\mathcal{M},2}(s_2) - V^\pi_{\widehat{\mathcal{M}},2}(s_2)\right]\right] \\
&= \sum_{h=1}^2 \mathbb{E}\left[\mathcal{B}^{\pi_h}_{\mathcal{M}} V^\pi_{\mathcal{M},h+1}(s_h) - \mathcal{B}^{\pi_h}_{\widehat{\mathcal{M}}} V^\pi_{\mathcal{M},h+1}(s_h)\right] + \mathbb{E}\left[V^\pi_{\mathcal{M},3}(s_3) - V^\pi_{\widehat{\mathcal{M}},3}(s_3)\right] \\
&= \ldots \\
&= \sum_{h=1}^H \mathbb{E}\left[\mathcal{B}^{\pi_h}_{\mathcal{M}} V^\pi_{\mathcal{M},h+1}(s_h) - \mathcal{B}^{\pi_h}_{\widehat{\mathcal{M}}} V^\pi_{\mathcal{M},h+1}(s_h)\right] + \underbrace{\mathbb{E}\left[V^\pi_{\mathcal{M},H+1}(s_{H+1}) - V^\pi_{\widehat{\mathcal{M}},H+1}(s_{H+1})\right]}_{=0} \\
&= \sum_{h=1}^H \mathbb{E}\left[\mathcal{B}^{\pi_h}_{\mathcal{M}} V^\pi_{\mathcal{M},h+1}(s_h) - \mathcal{B}^{\pi_h}_{\widehat{\mathcal{M}}} V^\pi_{\mathcal{M},h+1}(s_h)\right].
\end{aligned}
$$

For the second identity, we have nearly identical steps:

$$
\begin{aligned}
V^{\pi}_{\mathcal{M},1} - V^{\pi}_{\widehat{\mathcal{M}},1} &= \mathbb{E}\left[ V^{\pi}_{\mathcal{M},1}(s_1) - V^{\pi}_{\widehat{\mathcal{M}},1}(s_1) \right] \\
&= \mathbb{E}\left[ \mathcal{B}^{\pi_1}_{\mathcal{M}} V^{\pi}_{\mathcal{M},2}(s_1) - \mathcal{B}^{\pi_1}_{\widehat{\mathcal{M}}} V^{\pi}_{\widehat{\mathcal{M}},2}(s_1) \right] \\
&= \mathbb{E}\left[ \mathcal{B}^{\pi_1}_{\mathcal{M}} V^{\pi}_{\mathcal{M},2}(s_1) - \mathcal{B}^{\pi_1}_{\mathcal{M}} V^{\pi}_{\widehat{\mathcal{M}},2}(s_1) + \mathcal{B}^{\pi_1}_{\mathcal{M}} V^{\pi}_{\widehat{\mathcal{M}},2}(s_1) - \mathcal{B}^{\pi_1}_{\widehat{\mathcal{M}}} V^{\pi}_{\widehat{\mathcal{M}},2}(s_1) \right] \\
&= \mathbb{E}\left[ \mathcal{B}^{\pi_1}_{\mathcal{M}} V^{\pi}_{\widehat{\mathcal{M}},2}(s_1) - \mathcal{B}^{\pi_1}_{\widehat{\mathcal{M}}} V^{\pi}_{\widehat{\mathcal{M}},2}(s_1) + \mathbb{E}_{s_2 \sim \mathcal{T}(\cdot | s_1, a_1)}\left[ V^{\pi}_{\mathcal{M},2}(s_2) - V^{\pi}_{\widehat{\mathcal{M}},2}(s_2) \right] \right] \\
&= \sum_{h=1}^{2} \mathbb{E}\left[ \mathcal{B}^{\pi_h}_{\mathcal{M}} V^{\pi}_{\widehat{\mathcal{M}},h+1}(s_h) - \mathcal{B}^{\pi_h}_{\widehat{\mathcal{M}}} V^{\pi}_{\widehat{\mathcal{M}},h+1}(s_h) \right] + \mathbb{E}\left[ V^{\pi}_{\mathcal{M},3}(s_3) - V^{\pi}_{\widehat{\mathcal{M}},3}(s_3) \right] \\
&= \ldots \\
&= \sum_{h=1}^{H} \mathbb{E}\left[ \mathcal{B}^{\pi_h}_{\mathcal{M}} V^{\pi}_{\widehat{\mathcal{M}},h+1}(s_h) - \mathcal{B}^{\pi_h}_{\widehat{\mathcal{M}}} V^{\pi}_{\widehat{\mathcal{M}},h+1}(s_h) \right] + \underbrace{\mathbb{E}\left[ V^{\pi}_{\mathcal{M},H+1}(s_{H+1}) - V^{\pi}_{\widehat{\mathcal{M}},H+1}(s_{H+1}) \right]}_{=0} \\
&= \sum_{h=1}^{H} \mathbb{E}\left[ \mathcal{B}^{\pi_h}_{\mathcal{M}} V^{\pi}_{\widehat{\mathcal{M}},h+1}(s_h) - \mathcal{B}^{\pi_h}_{\widehat{\mathcal{M}}} V^{\pi}_{\widehat{\mathcal{M}},h+1}(s_h) \right].
\end{aligned}
$$

$\square$

**Theorem 6.** *Take any $\Pi \supseteq \{\mathcal{S} \to \mathcal{A}\}$, any $\mathcal{V} \supseteq \{V^{\pi} \mid \pi \in \Pi^H\}$, and fix any $D \geq 0$. For each episode $k \in [K]$, let $\widetilde{\mathcal{M}}_k$ be any MDP that achieves the rate-distortion limit of $\mathcal{R}^{\Pi,\mathcal{V}}_k(D)$ with information source $\mathbb{P}(\mathcal{M}^{\star} \in \cdot \mid H_k)$ and distortion function $d_{\Pi,\mathcal{V}}$. Then,*

$$
\textsc{BayesRegret}(K, \pi^{(1)}, \ldots, \pi^{(K)}) \leq \mathbb{E}\left[ \sum_{k=1}^{K} \mathbb{E}_k\left[ V^{\star}_{\widetilde{\mathcal{M}}_k,1} - V^{\pi^{(k)}}_{\widetilde{\mathcal{M}}_k,1} \right] \right] + 2KH\sqrt{D}.
$$

*Proof.* By applying definitions from Section 3 and applying the tower property of expectation, we have that

$$
\textsc{BayesRegret}(K, \pi^{(1)}, \ldots, \pi^{(K)}) = \mathbb{E}\left[ \sum_{k=1}^{K} \mathbb{E}_k\left[ \Delta_k \right] \right].
$$

Examining the $k$th episode in isolation and applying the definition of episodic regret, we have

$$
\begin{aligned}
\mathbb{E}_k[\Delta_k] &= \mathbb{E}_k\left[ V^{\star}_{\mathcal{M}^{\star},1} - V^{\pi^{(k)}}_{\mathcal{M}^{\star},1} \right] \\
&= \mathbb{E}_k\left[ V^{\star}_{\mathcal{M}^{\star},1} - V^{\star}_{\widetilde{\mathcal{M}}_k,1} + V^{\star}_{\widetilde{\mathcal{M}}_k,1} - V^{\pi^{(k)}}_{\widetilde{\mathcal{M}}_k,1} + V^{\pi^{(k)}}_{\widetilde{\mathcal{M}}_k,1} - V^{\pi^{(k)}}_{\mathcal{M}^{\star},1} \right] \\
&= \mathbb{E}_k\left[ V^{\star}_{\mathcal{M}^{\star},1} - V^{\pi^{\star}_{\mathcal{M}^{\star}}}_{\widetilde{\mathcal{M}}_k,1} + \underbrace{V^{\pi^{\star}_{\mathcal{M}^{\star}}}_{\widetilde{\mathcal{M}}_k,1} - V^{\star}_{\widetilde{\mathcal{M}}_k,1}}_{\leq 0} + V^{\star}_{\widetilde{\mathcal{M}}_k,1} - V^{\pi^{(k)}}_{\widetilde{\mathcal{M}}_k,1} + V^{\pi^{(k)}}_{\widetilde{\mathcal{M}}_k,1} - V^{\pi^{(k)}}_{\mathcal{M}^{\star},1} \right] \\
&\leq \mathbb{E}_k\left[ V^{\star}_{\mathcal{M}^{\star},1} - V^{\pi^{\star}_{\mathcal{M}^{\star}}}_{\widetilde{\mathcal{M}}_k,1} + V^{\star}_{\widetilde{\mathcal{M}}_k,1} - V^{\pi^{(k)}}_{\widetilde{\mathcal{M}}_k,1} + V^{\pi^{(k)}}_{\widetilde{\mathcal{M}}_k,1} - V^{\pi^{(k)}}_{\mathcal{M}^{\star},1} \right]
\end{aligned}
$$

For brevity, we let $\pi_h^\star \triangleq \pi_{\mathcal{M}^\star,h}^\star$ and observe that an application of Lemma [4] yields

$$
\begin{aligned}
\mathbb{E}_k \left[ V_{\mathcal{M}^\star,1}^\star - V_{\widetilde{\mathcal{M}}_k,1}^{\pi_{\mathcal{M}^\star}^\star} \right] &= \sum_{h=1}^{H} \mathbb{E}_k \left[ \mathcal{B}_{\mathcal{M}^\star}^{\pi_h^\star} V_{\mathcal{M}^\star,h+1}^\star(s_h) - \mathcal{B}_{\widetilde{\mathcal{M}}_k}^{\pi_h^\star} V_{\mathcal{M}^\star,h+1}^\star(s_h) \right] \\
&\leq \sum_{h=1}^{H} \mathbb{E}_k \left[ \left| \mathcal{B}_{\mathcal{M}^\star}^{\pi_h^\star} V_{\mathcal{M}^\star,h+1}^\star(s_h) - \mathcal{B}_{\widetilde{\mathcal{M}}_k}^{\pi_h^\star} V_{\mathcal{M}^\star,h+1}^\star(s_h) \right| \right] \\
&= \sum_{h=1}^{H} \mathbb{E}_k \left[ \sqrt{\left( \mathcal{B}_{\mathcal{M}^\star}^{\pi_h^\star} V_{\mathcal{M}^\star,h+1}^\star(s_h) - \mathcal{B}_{\widetilde{\mathcal{M}}_k}^{\pi_h^\star} V_{\mathcal{M}^\star,h+1}^\star(s_h) \right)^2} \right] \\
&\leq \sum_{h=1}^{H} \mathbb{E}_k \left[ \sqrt{ \| \mathcal{B}_{\mathcal{M}^\star}^{\pi_h^\star} V_{\mathcal{M}^\star,h+1}^\star - \mathcal{B}_{\widetilde{\mathcal{M}}_k}^{\pi_h^\star} V_{\mathcal{M}^\star,h+1}^\star \|_\infty^2 } \right] \\
&\leq \sum_{h=1}^{H} \sqrt{ \mathbb{E}_k \left[ \| \mathcal{B}_{\mathcal{M}^\star}^{\pi_h^\star} V_{\mathcal{M}^\star,h+1}^\star - \mathcal{B}_{\widetilde{\mathcal{M}}_k}^{\pi_h^\star} V_{\mathcal{M}^\star,h+1}^\star \|_\infty^2 \right] } \\
&\leq \sum_{h=1}^{H} \sqrt{ \mathbb{E}_k \left[ \sup_{\substack{\pi \in \Pi \\ V \in \mathcal{V}}} \| \mathcal{B}_{\mathcal{M}^\star}^{\pi} V - \mathcal{B}_{\widetilde{\mathcal{M}}_k}^{\pi} V \|_\infty^2 \right] } \\
&= \sum_{h=1}^{H} \sqrt{ \mathbb{E}_k \left[ d_{\Pi,\mathcal{V}}(\mathcal{M}^\star, \widetilde{\mathcal{M}}_k) \right] } \\
&\leq H\sqrt{D},
\end{aligned}
$$

where the third inequality invokes Jensen's inequality, the fourth inequality holds as $\Pi \supseteq \{\mathcal{S} \to \mathcal{A}\}$ and $\mathcal{V} \supseteq \{V^\pi \mid \pi \in \Pi^H\}$ ensures that $V_{\mathcal{M}^\star,h}^\star \in \mathcal{V}, \forall h \in [H]$, and the final inequality holds since $\widetilde{\mathcal{M}}_k$ achieves the rate-distortion limit in the $k$th episode, by assumption.

We follow the same sequence of steps to obtain

$$
\begin{aligned}
\mathbb{E}_k \left[ V_{\widetilde{\mathcal{M}}_k,1}^{\pi^{(k)}} - V_{\mathcal{M}^\star,1}^{\pi^{(k)}} \right] &= \sum_{h=1}^{H} \mathbb{E}_k \left[ \mathcal{B}_{\mathcal{M}^\star}^{\pi_h^{(k)}} V_{\mathcal{M}^\star,h+1}^{\pi^{(k)}}(s_h) - \mathcal{B}_{\widetilde{\mathcal{M}}_k}^{\pi_h^{(k)}} V_{\mathcal{M}^\star,h+1}^{\pi^{(k)}}(s_h) \right] \\
&\leq \sum_{h=1}^{H} \mathbb{E}_k \left[ \left| \mathcal{B}_{\mathcal{M}^\star}^{\pi_h^{(k)}} V_{\mathcal{M}^\star,h+1}^{\pi^{(k)}}(s_h) - \mathcal{B}_{\widetilde{\mathcal{M}}_k}^{\pi_h^{(k)}} V_{\mathcal{M}^\star,h+1}^{\pi^{(k)}}(s_h) \right| \right] \\
&= \sum_{h=1}^{H} \mathbb{E}_k \left[ \sqrt{ \left( \mathcal{B}_{\mathcal{M}^\star}^{\pi_h^{(k)}} V_{\mathcal{M}^\star,h+1}^{\pi^{(k)}}(s_h) - \mathcal{B}_{\widetilde{\mathcal{M}}_k}^{\pi_h^{(k)}} V_{\mathcal{M}^\star,h+1}^{\pi^{(k)}}(s_h) \right)^2 } \right] \\
&\leq \sum_{h=1}^{H} \mathbb{E}_k \left[ \sqrt{ \| \mathcal{B}_{\mathcal{M}^\star}^{\pi_h^{(k)}} V_{\mathcal{M}^\star,h+1}^{\pi^{(k)}} - \mathcal{B}_{\widetilde{\mathcal{M}}_k}^{\pi_h^{(k)}} V_{\mathcal{M}^\star,h+1}^{\pi^{(k)}} \|_\infty^2 } \right] \\
&\leq \sum_{h=1}^{H} \sqrt{ \mathbb{E}_k \left[ \| \mathcal{B}_{\mathcal{M}^\star}^{\pi_h^{(k)}} V_{\mathcal{M}^\star,h+1}^{\pi^{(k)}} - \mathcal{B}_{\widetilde{\mathcal{M}}_k}^{\pi_h^{(k)}} V_{\mathcal{M}^\star,h+1}^{\pi^{(k)}} \|_\infty^2 \right] } \\
&\leq \sum_{h=1}^{H} \sqrt{ \mathbb{E}_k \left[ \sup_{\substack{\pi \in \Pi \\ V \in \mathcal{V}}} \| \mathcal{B}_{\mathcal{M}^\star}^{\pi} V - \mathcal{B}_{\widetilde{\mathcal{M}}_k}^{\pi} V \|_\infty^2 \right] } \\
&= \sum_{h=1}^{H} \sqrt{ \mathbb{E}_k \left[ d_{\Pi,\mathcal{V}}(\mathcal{M}^\star, \widetilde{\mathcal{M}}_k) \right] } \\
&\leq H\sqrt{D}.
\end{aligned}
$$

Substituting back into our original expression, we have

$$\mathbb{E}_k\left[\Delta_k\right] = \mathbb{E}_k\left[V^\star_{\mathcal{M}^\star,1} - V^{\pi^{(k)}}_{\mathcal{M}^\star,1}\right]$$

$$\leq \mathbb{E}_k\left[V^\star_{\mathcal{M}^\star,1} - V^{\pi^\star_{\mathcal{M}^\star}}_{\widetilde{\mathcal{M}}_k,1} + V^\star_{\widetilde{\mathcal{M}}_k,1} - V^{\pi^{(k)}}_{\widetilde{\mathcal{M}}_k,1} + V^{\pi^{(k)}}_{\widetilde{\mathcal{M}}_k,1} - V^{\pi^{(k)}}_{\mathcal{M}^\star,1}\right]$$

$$\leq \mathbb{E}_k\left[V^\star_{\widetilde{\mathcal{M}}_k,1} - V^{\pi^{(k)}}_{\widetilde{\mathcal{M}}_k,1}\right] + 2H\sqrt{D}.$$

Applying this upper bound on episodic regret in each episode yields

$$\textsc{BayesRegret}(K, \pi^{(1)}, \ldots, \pi^{(K)}) = \mathbb{E}\left[\sum_{k=1}^K \mathbb{E}_k\left[\Delta_k\right]\right]$$

$$\leq \mathbb{E}\left[\sum_{k=1}^K \mathbb{E}_k\left[V^\star_{\widetilde{\mathcal{M}}_k,1} - V^{\pi^{(k)}}_{\widetilde{\mathcal{M}}_k,1}\right]\right] + 2KH\sqrt{D},$$

as desired. $\qquad\square$

# D  Proof of Lemma 6

In this section, we develop counterparts to the results of Arumugam and Van Roy [15] for the reinforcement-learning setting which relate each rate-distortion function $\mathcal{R}_k^{\Pi,\mathcal{V}}(D)$ to the information accumulated by the agent over the course of learning. Recall that $\tau_k = (s_1^{(k)}, a_1^{(k)}, r_1^{(k)}, \ldots, s_H^{(k)}, a_H^{(k)}, r_H^{(k)}, s_{H+1}^{(k)})$ is a random variable denoting the trajectory experienced by the agent in the $k$th episode given the history $H_k$. Let MDP $M_k$ be the MDP sampled in the $k$th episode.

**Lemma 5.** *For all $k \in [K]$,*

$$\mathbb{E}_k\left[\mathcal{R}_{k+1}^{\Pi,\mathcal{V}}(D)\right] \leq \mathcal{R}_k^{\Pi,\mathcal{V}}(D) - \mathbb{I}_k(\widetilde{\mathcal{M}}_k; \tau_k \mid M_k).$$

*Proof.* Recall that, by definition $H_{k+1} = (H_k, \tau_k)$. For all $k \in [K]$, observe that, conditioned on the true MDP $\mathcal{M}^\star$ and sampled MDP $M_k$ which generated the history $H_{k+1}$, we have that for any compressed MDP $\widetilde{\mathcal{M}}$, $\mathbb{P}(H_{k+1}, \widetilde{\mathcal{M}} \mid \mathcal{M}^\star, M_k) = \mathbb{P}(H_{k+1} \mid \mathcal{M}^\star, M_k)\mathbb{P}(\widetilde{\mathcal{M}} \mid \mathcal{M}^\star, M_k)$. Using this independence $H_{k+1} \perp \widetilde{\mathcal{M}} \mid \mathcal{M}^\star, M_k \ \forall k \in [K]$, we have that

$$0 = \mathbb{I}_k(H_{k+1}; \widetilde{\mathcal{M}} \mid \mathcal{M}^\star, M_k) = \mathbb{I}_k(H_k, \tau_k; \widetilde{\mathcal{M}} \mid \mathcal{M}^\star, M_k) = \mathbb{I}_k(\tau_k; \widetilde{\mathcal{M}} \mid \mathcal{M}^\star, M_k).$$

Moreover, we know that the sampled MDP $M_k$ does not affect our uncertainty in the true MDP $\mathcal{M}^\star$ such that

$$\mathbb{I}_k(\mathcal{M}^\star; \widetilde{\mathcal{M}}) = \mathbb{I}_k(\mathcal{M}^\star; \widetilde{\mathcal{M}} \mid M_k).$$

By the chain rule of mutual information,

$$\mathbb{I}_k(\mathcal{M}^\star; \widetilde{\mathcal{M}}) = \mathbb{I}_k(\mathcal{M}^\star; \widetilde{\mathcal{M}} \mid M_k) = \mathbb{I}_k(\mathcal{M}^\star; \widetilde{\mathcal{M}} \mid M_k) + \mathbb{I}_k(\tau_k; \widetilde{\mathcal{M}} \mid \mathcal{M}^\star, M_k) = \mathbb{I}_k(\mathcal{M}^\star, \tau_k; \widetilde{\mathcal{M}} \mid M_k).$$

Applying the chain rule a second time yields

$$\mathbb{I}_k(\mathcal{M}^\star; \widetilde{\mathcal{M}}) = \mathbb{I}_k(\mathcal{M}^\star, \tau_k; \widetilde{\mathcal{M}} \mid M_k) = \mathbb{I}_k(\widetilde{\mathcal{M}}; \tau_k \mid M_k) + \mathbb{I}_k(\mathcal{M}^\star; \widetilde{\mathcal{M}} \mid \tau_k, M_k).$$

By definition of the rate-distortion function, we have

$$\mathbb{E}_k\left[\mathcal{R}_{k+1}^{\Pi,\mathcal{V}}(D)\right] = \mathbb{E}_k\left[\inf_{\widetilde{\mathcal{M}} \in \Lambda_{k+1}(D)} \mathbb{I}_{k+1}(\mathcal{M}^\star; \widetilde{\mathcal{M}})\right], \qquad \Lambda_{k+1}(D) = \{\widetilde{\mathcal{M}} : \Omega \to \mathfrak{M} \mid \mathbb{E}_{k+1}[d_{\Pi,\mathcal{V}}(\mathcal{M}^\star, \widetilde{\mathcal{M}})] \leq D\}.$$

Recall that, by definition, $\widetilde{\mathcal{M}}_k$ achieves the rate-distortion limit of $\mathcal{R}_k^{\Pi,\mathcal{V}}(D)$, implying that $\mathbb{E}_k[d_{\Pi,\mathcal{V}}(\mathcal{M}^\star,\widetilde{\mathcal{M}}_k)] \leq D$. By the tower property of expectation, we recover that

$$\mathbb{E}_k\left[\mathbb{E}_{k+1}[d_{\Pi,\mathcal{V}}(\mathcal{M}^\star,\widetilde{\mathcal{M}}_k)]\right] = \mathbb{E}_k[d_{\Pi,\mathcal{V}}(\mathcal{M}^\star,\widetilde{\mathcal{M}}_k)] \leq D,$$

and so, in expectation given the current history $H_k$, $\widetilde{\mathcal{M}}_k \in \Lambda_{k+1}(D)$. Thus, we have that

$$\mathbb{E}_k\left[\mathcal{R}_{k+1}^{\Pi,\mathcal{V}}(D)\right] = \mathbb{E}_k\left[\inf_{\widetilde{\mathcal{M}}\in\Lambda_{k+1}(D)} \mathbb{I}_{k+1}(\mathcal{M}^\star;\widetilde{\mathcal{M}})\right] \leq \mathbb{E}_k\left[\mathbb{I}_{k+1}(\mathcal{M}^\star;\widetilde{\mathcal{M}}_k)\right].$$

Re-arranging terms from our previous chain rule expansions, we may expand the integrand as

$$\begin{aligned}
\mathbb{E}_k\left[\mathbb{I}_{k+1}(\mathcal{M}^\star;\widetilde{\mathcal{M}}_k)\right] &= \mathbb{E}_k\left[\mathbb{I}_k(\mathcal{M}^\star;\widetilde{\mathcal{M}}_k \mid \tau_k, M_k)\right] \\
&= \mathbb{E}_k\left[\mathbb{I}_k(\mathcal{M}^\star;\widetilde{\mathcal{M}}_k) - \mathbb{I}_k(\widetilde{\mathcal{M}}_k,\tau_k \mid M_k)\right] \\
&= \mathbb{I}_k(\mathcal{M}^\star;\widetilde{\mathcal{M}}_k) - \mathbb{I}_k(\widetilde{\mathcal{M}}_k,\tau_k \mid M_k) \\
&= \mathcal{R}_k^{\Pi,\mathcal{V}}(D) - \mathbb{I}_k(\widetilde{\mathcal{M}}_k;\tau_k \mid M_k),
\end{aligned}$$

where the penultimate line follows since both mutual information terms are $\sigma(H_k)$-measurable and the final line follows by definition of $\widetilde{\mathcal{M}}_k$. $\qquad\square$

At the beginning of each episode, our generalization of PSRL will identify a compressed MDP $\widetilde{\mathcal{M}}_k$ that achieves the rate-distortion limit based on the current history $H_k$. As data accumulates and the agent's knowledge of the true MDP is refined, this satisficing MDP $\widetilde{\mathcal{M}}_k$ will be recomputed to reflect that updated knowledge. The previous lemma shows that the expected number of bits the agent must identify to learn this new target MDP decreases as this adaptation occurs, highlighting two possible sources of improvement: (1) shifting from a compressed MDP $\widetilde{\mathcal{M}}_k$ to $\widetilde{\mathcal{M}}_{k+1}$ and (2) a decrease of $\mathbb{I}_k(\widetilde{\mathcal{M}};\tau_k \mid M_k)$ that occurs from observing the trajectory $\tau_k$. The former reflects the agent's improved ability in synthesizing an approximately value-equivalent MDP to pursue instead of $\mathcal{M}^\star$ while the latter captures information gained about the previous target $\widetilde{\mathcal{M}}_k$ from the experienced trajectory $\tau_k$.

**Fact 1** ([41]). *For any $\Pi, \mathcal{V}$ and all $k \in [K]$, $\mathcal{R}_k^{\Pi,\mathcal{V}}(D)$ is a non-negative, convex, and monotonically-decreasing function in $D$.*

Let $\widetilde{\mathcal{M}}$ be a compressed MDP that is exactly value-equivalent to $\mathcal{M}^\star$ which, by definition, implies a distortion of exactly zero. Further recall that $\mathcal{M}^\star$ is itself a MDP that achieves zero distortion, albeit one that has no guarantee of achieving the rate-distortion limit. Fact 1 yields the following chain of inequalities that hold for all $k \in [K]$ and $D \geq 0$:

$$\mathcal{R}_k^{\Pi,\mathcal{V}}(D) \leq \mathbb{I}_k(\mathcal{M}^\star;\widetilde{\mathcal{M}}) \leq \mathbb{I}_k(\mathcal{M}^\star;\mathcal{M}^\star) = \mathbb{H}_k(\mathcal{M}^\star).$$

This chain of inequalities confirms an important goal of satisficing in PSRL; namely, that the compressed MDP an agent attempts to solve in each episode $k \in [K]$, $\widetilde{\mathcal{M}}_k$, requires fewer bits of information than what is needed to fully identify the true MDP $\mathcal{M}^\star$. This gives rise to the following corollary:

**Corollary 5.** *For any $k \in [K]$,*

$$\mathbb{E}_k\left[\sum_{k'=k}^{K} \mathbb{I}_{k'}(\widetilde{\mathcal{M}}_{k'};\tau_{k'} \mid M_{k'})\right] \leq \mathbb{H}_k(\mathcal{M}^\star).$$

Instead of proving this corollary, we prove the following lemma which yields the corollary through Fact 1:

**Lemma 6.** *For any $k \in [K]$,*

$$\mathbb{E}_k\left[\sum_{k'=k}^{K} \mathbb{I}_{k'}(\widetilde{\mathcal{M}}_{k'};\tau_{k'} \mid M_{k'})\right] \leq \mathcal{R}_k^{\Pi,\mathcal{V}}(D).$$

*Proof.* Observe that by Lemma 5, for all $k \in [K]$,

$$\mathbb{I}_k(\widetilde{\mathcal{M}}_k; \tau_k \mid M_k) \leq \mathcal{R}_k^{\Pi,\mathcal{V}}(D) - \mathbb{E}_k \left[ \mathcal{R}_{k+1}^{\Pi,\mathcal{V}}(D) \right].$$

Directly substituting in, we have

$$\mathbb{E}_k \left[ \sum_{k'=k}^{K} \mathbb{I}_{k'}(\widetilde{\mathcal{M}}_{k'}; \tau_{k'} \mid M_{k'}) \right] \leq \mathbb{E}_k \left[ \sum_{k'=k}^{K} \left( \mathcal{R}_{k'}^{\Pi,\mathcal{V}}(D) - \mathbb{E}_{k'} \left[ \mathcal{R}_{k'+1}^{\Pi,\mathcal{V}}(D) \right] \right) \right].$$

Applying linearity of expectation and breaking apart the sum yields

$$\mathbb{E}_k \left[ \sum_{k'=k}^{K} \mathbb{I}_{k'}(\widetilde{\mathcal{M}}_{k'}; \tau_{k'} \mid M_{k'}) \right] \leq \sum_{k'=k}^{K} \mathbb{E}_k \left[ \mathcal{R}_{k'}^{\Pi,\mathcal{V}}(D) \right] - \sum_{k'=k}^{K} \mathbb{E}_k \left[ \mathbb{E}_{k'} \left[ \mathcal{R}_{k'+1}^{\Pi,\mathcal{V}}(D) \right] \right].$$

Note that the first term can simply be separated into

$$\sum_{k'=k}^{K} \mathbb{E}_k \left[ \mathcal{R}_{k'}^{\Pi,\mathcal{V}}(D) \right] = \mathbb{E}_k \left[ \mathcal{R}_k^{\Pi,\mathcal{V}}(D) \right] + \sum_{k'=k+1}^{K} \mathbb{E}_k \left[ \mathcal{R}_{k'}^{\Pi,\mathcal{V}}(D) \right] = \mathcal{R}_k^{\Pi,\mathcal{V}}(D) + \sum_{k'=k+1}^{K} \mathbb{E}_k \left[ \mathcal{R}_{k'}^{\Pi,\mathcal{V}}(D) \right].$$

Meanwhile, since $\sigma(H_k) \subseteq \sigma(H_{k'})$, the tower property of expectation yields

$$\sum_{k'=k}^{K} \mathbb{E}_k \left[ \mathbb{E}_{k'} \left[ \mathcal{R}_{k'+1}^{\Pi,\mathcal{V}}(D) \right] \right] = \sum_{k'=k}^{K} \mathbb{E}_k \left[ \mathcal{R}_{k'+1}^{\Pi,\mathcal{V}}(D) \right] = \sum_{k'=k+1}^{K} \mathbb{E}_k \left[ \mathcal{R}_{k'}^{\Pi,\mathcal{V}}(D) \right].$$

Combining the expansions results in

$$\begin{aligned}
\mathbb{E}_k \left[ \sum_{k'=k}^{K} \mathbb{I}_{k'}(\widetilde{\mathcal{M}}_{k'}; \tau_{k'} \mid M_{k'}) \right] &\leq \sum_{k'=k}^{K} \mathbb{E}_k \left[ \mathcal{R}_{k'}^{\Pi,\mathcal{V}}(D) \right] - \sum_{k'=k}^{K} \mathbb{E}_k \left[ \mathbb{E}_{k'} \left[ \mathcal{R}_{k'+1}^{\Pi,\mathcal{V}}(D) \right] \right] \\
&= \mathcal{R}_k^{\Pi,\mathcal{V}}(D) + \sum_{k'=k+1}^{K} \mathbb{E}_k \left[ \mathcal{R}_{k'}^{\Pi,\mathcal{V}}(D) \right] - \sum_{k'=k+1}^{K} \mathbb{E}_k \left[ \mathcal{R}_{k'}^{\Pi,\mathcal{V}}(D) \right] \\
&= \mathcal{R}_k^{\Pi,\mathcal{V}}(D).
\end{aligned}$$

$\square$

# E    Proof of Theorem 2

In this section, we prove a general, information-theoretic satisficing Bayesian regret bound. Central to our analysis is the information ratio in the $k$th episode:

$$\Gamma_k \triangleq \frac{\mathbb{E}_k \left[ V_{\widetilde{\mathcal{M}}_k,1}^{\star} - V_{\widetilde{\mathcal{M}}_k,1}^{\pi^{(k)}} \right]^2}{\mathbb{I}_k(\widetilde{\mathcal{M}}_k; \tau_k, M_k)}, \qquad \forall k \in [K].$$

**Theorem 7** (Information-Theoretic Satisficing Regret Bound). *If $\Gamma_k \leq \overline{\Gamma}$, for all $k \in [K]$, then*

$$\mathbb{E}_k \left[ \sum_{k=1}^{K} \mathbb{E} \left[ V_{\widetilde{\mathcal{M}}_k,1}^{\star} - V_{\widetilde{\mathcal{M}}_k,1}^{\pi^{(k)}} \right] \right] \leq \sqrt{\overline{\Gamma} K \mathcal{R}_1^{\Pi,\mathcal{V}}(D)}.$$

*Proof.* The definition of the information ratio $\Gamma_k$ for each term in the sum followed by the fact that $\Gamma_k \leq \overline{\Gamma}, \forall k \in [K]$ yields

$$\mathbb{E} \left[ \sum_{k=1}^{K} \mathbb{E}_k \left[ V_{\widetilde{\mathcal{M}}_k,1}^{\star} - V_{\widetilde{\mathcal{M}}_k,1}^{\pi^{(k)}} \right] \right] = \mathbb{E} \left[ \sum_{k=1}^{K} \sqrt{\Gamma_k \mathbb{I}_k(\widetilde{\mathcal{M}}_k; \tau_k, M_k)} \right] \leq \sqrt{\overline{\Gamma}} \mathbb{E} \left[ \sum_{k=1}^{K} \sqrt{\mathbb{I}_k(\widetilde{\mathcal{M}}_k; \tau_k, M_k)} \right].$$

Applying the tower property of expectation and Jensen's inequality in sequence yields

$$\sqrt{\overline{\Gamma}} \mathbb{E} \left[ \sum_{k=1}^{K} \sqrt{\mathbb{I}_k(\widetilde{\mathcal{M}}_k; \tau_k, M_k)} \right] \leq \sqrt{\overline{\Gamma}} \mathbb{E} \left[ \sum_{k=1}^{K} \sqrt{\mathbb{E}_k \left[ \mathbb{I}_k(\widetilde{\mathcal{M}}_k; \tau_k, M_k) \right]} \right].$$

By the Cauchy-Schwarz inequality, we have that

$$\sqrt{\overline{\Gamma}}\mathbb{E}\left[\sum_{k=1}^{K}\sqrt{\mathbb{E}_k\left[\mathbb{I}_k(\widetilde{\mathcal{M}}_k;\tau_k,M_k)\right]}\right] \leq \sqrt{\overline{\Gamma}}\mathbb{E}\left[\sqrt{K\sum_{k=1}^{K}\mathbb{E}_k\left[\mathbb{I}_k(\widetilde{\mathcal{M}}_k;\tau_k,M_k)\right]}\right].$$

Recall that the sampled $M_k$ by itself offers no information about $\widetilde{\mathcal{M}}_k$. Consequently, by the chain rule of mutual information, we have

$$\mathbb{I}_k(\widetilde{\mathcal{M}}_k;\tau_k,M_k) = \mathbb{I}_k(\widetilde{\mathcal{M}}_k;M_k) + \mathbb{I}_k(\widetilde{\mathcal{M}}_k;\tau_k \mid M_k) = \mathbb{I}_k(\widetilde{\mathcal{M}}_k;\tau_k \mid M_k).$$

Therefore,

$$\sqrt{\overline{\Gamma}}\mathbb{E}\left[\sqrt{K\sum_{k=1}^{K}\mathbb{E}_k\left[\mathbb{I}_k(\widetilde{\mathcal{M}}_k;\tau_k,M_k)\right]}\right] = \sqrt{\overline{\Gamma}}\mathbb{E}\left[\sqrt{K\sum_{k=1}^{K}\mathbb{E}_k\left[\mathbb{I}_k(\widetilde{\mathcal{M}}_k;\tau_k \mid M_k)\right]}\right].$$

Directly applying Lemma 6 followed by Jensen's inequality yields

$$\sqrt{\overline{\Gamma}}\mathbb{E}\left[\sqrt{K\sum_{k=1}^{K}\mathbb{E}_k\left[\mathbb{I}_k(\widetilde{\mathcal{M}}_k;\tau_k \mid M_k)\right]}\right] \leq \sqrt{\overline{\Gamma}}\mathbb{E}\left[\sqrt{K\mathcal{R}_1^{\Pi,\mathcal{V}}(D)}\right] \leq \sqrt{\overline{\Gamma}K\mathbb{E}\left[\mathcal{R}_1^{\Pi,\mathcal{V}}(D)\right]}.$$

Since the expectation is with respect to the prior $\mathbb{P}(\mathcal{M}^\star \in \cdot \mid H_1)$ and $\mathcal{R}_1^{\Pi,\mathcal{V}}(D)$ is $\sigma(H_1)$-measurable, we have

$$\sqrt{\overline{\Gamma}K\mathbb{E}\left[\mathcal{R}_1^{\Pi,\mathcal{V}}(D)\right]} = \sqrt{\overline{\Gamma}K\mathcal{R}_1^{\Pi,\mathcal{V}}(D)},$$

as desired. $\qquad\qquad\square$

# F    Proof of Lemma 1

In this section, we clarify how the shrinkage or growth of the policy class $\Pi$ and value function class $\mathcal{V}$ affect the rate-distortion function at the $k$th episode, $\mathcal{R}_k^{\Pi,\mathcal{V}}(D)$.

**Lemma 7** (Dominance with Approximate Value Equivalence). *For any two $\Pi, \Pi'$ and any $\mathcal{V}, \mathcal{V}'$ such that $\Pi' \subseteq \Pi \subseteq \{\mathcal{S} \to \Delta(\mathcal{A})\}$ and $\mathcal{V}' \subseteq \mathcal{V} \subseteq \{\mathcal{S} \to \mathbb{R}\}$, we have*

$$\mathcal{R}_k^{\Pi,\mathcal{V}}(D) \geq \mathcal{R}_k^{\Pi',\mathcal{V}'}(D), \qquad \forall k \in [K], D > 0.$$

*Proof.* Recall that the distortion function $d : \mathfrak{M} \times \mathfrak{M} \to \mathbb{R}_{\geq 0}$ with respect to policy class $\Pi$ and value function class $\mathcal{V}$ is given by

$$d_{\Pi,\mathcal{V}}(\mathcal{M},\widehat{\mathcal{M}}) = \sup_{\substack{\pi\in\Pi\\V\in\mathcal{V}}} ||\mathcal{B}_{\mathcal{M}}^\pi V - \mathcal{B}_{\widehat{\mathcal{M}}}^\pi V||_\infty^2 = \sup_{\substack{\pi\in\Pi\\V\in\mathcal{V}}}\left(\max_{s\in\mathcal{S}}|\mathcal{B}_{\mathcal{M}}^\pi V(s) - \mathcal{B}_{\widehat{\mathcal{M}}}^\pi V(s)|\right)^2,$$

with an analogous definition holding for the distortion function $d_{\Pi',\mathcal{V}'}$ under $\Pi'$ and $\mathcal{V}'$. In the parlance of Stjernvall [150], we have that $d_{\Pi,\mathcal{V}}$ dominates $d_{\Pi',\mathcal{V}'}$ if for all source distributions $\mathbb{P}(\mathcal{M}^\star \in \cdot \mid H_k)$ and all distortion thresholds $D > 0$,

$$\mathcal{R}_k^{\Pi,\mathcal{V}}(D) \geq \mathcal{R}_k^{\Pi',\mathcal{V}'}(D).$$

In words, a distortion function $d_1$ that dominates another distortion function $d_2$ requires more bits of information in order to achieve the rate-distortion limit for all information sources and at all distortion thresholds. From this definition, it is clear that statement of the theorem holds if we can establish a dominance relationship between $d_{\Pi,\mathcal{V}}$ and $d_{\Pi',\mathcal{V}'}$.

Recognizing the significant amount of calculation needed to exhaustively verify a dominance relationship by hand, Stjernvall [150] prescribes six sufficient conditions for establishing dominance (with varying degrees of strength) between distortion functions; we will leverage the second of these characterizations (C2).

Fix an arbitrary source distribution $\mathbb{P}(\mathcal{M}^\star \in \cdot \mid H_k)$ and distortion threshold $D > 0$. We denote by $\widetilde{\mathcal{M}}_k$ the MDP that achieves the rate-distortion limit $\mathcal{R}_k^{\Pi,\mathcal{V}}(D)$ under our chosen source, distortion threshold, and distortion function $d_{\Pi,\mathcal{V}}$. By definition of the supremum, we have that for any two MDPs $\mathcal{M}, \widehat{\mathcal{M}}$

$$d_{\Pi',\mathcal{V}'}(\mathcal{M}, \widehat{\mathcal{M}}) = \sup_{\substack{\pi \in \Pi' \\ V \in \mathcal{V}'}} ||\mathcal{B}_{\mathcal{M}}^\pi V - \mathcal{B}_{\widehat{\mathcal{M}}}^\pi V||_\infty^2 \leq \sup_{\substack{\pi \in \Pi \\ V \in \mathcal{V}}} ||\mathcal{B}_{\mathcal{M}}^\pi V - \mathcal{B}_{\widehat{\mathcal{M}}}^\pi V||_\infty^2 = d_{\Pi,\mathcal{V}}(\mathcal{M}, \widehat{\mathcal{M}}).$$

Consequently, since $\widetilde{\mathcal{M}}_k$ achieves the rate-distortion limit, we have

$$\mathbb{E}_k\left[ d_{\Pi',\mathcal{V}'}(\mathcal{M}^\star, \widetilde{\mathcal{M}}_k) \right] \leq \mathbb{E}_k\left[ d_{\Pi,\mathcal{V}}(\mathcal{M}^\star, \widetilde{\mathcal{M}}_k) \right] \leq D.$$

Observe that, since our information source and distortion threshold were arbitrary, we have that for all sources $\mathbb{P}(\mathcal{M}^\star \in \cdot \mid H_k)$ and all thresholds $D > 0$ with $\widetilde{\mathcal{M}}_k$ achieving the rate-distortion limit under distortion $d_{\Pi,\mathcal{V}}$, there exists a Markov chain $\mathcal{M}^\star - \widetilde{\mathcal{M}}_k - \widetilde{\mathcal{M}}_k'$ such that $\widetilde{\mathcal{M}}_k = \widetilde{\mathcal{M}}_k'$ (the mapping between them is the identity function) and $\mathbb{E}\left[ d_{\Pi',\mathcal{V}'}(\mathcal{M}^\star, \widetilde{\mathcal{M}}_k') \right] \leq D$. Thus, by Theorem 2 of Stjernvall [150] (specifically, C2 $\implies$ D4), we have that $d_{\Pi,\mathcal{V}}$ dominates $d_{\Pi',\mathcal{V}'}$ for any $\Pi' \subseteq \Pi \subseteq \{\mathcal{S} \to \Delta(\mathcal{A})\}$ and $\mathcal{V}' \subseteq \mathcal{V} \subseteq \{\mathcal{S} \to \mathbb{R}\}$. As previously discussed, the claim of the theorem follows as an immediate consequence, by definition of dominance. $\square$

# G   Proof of Lemma 2

Fano's inequality [64] is a key result in information theory that relates conditional entropy to the probability of error in a discrete, multi-way hypothesis testing problem. The traditional form of the result, however, determines an error as the inability to exactly recover the random variable being estimated. Naturally, given the lossy compression context of this work, a more useful analysis will use a lack of adherence to the distortion upper bound as the more appropriate notion of error. For this purpose, we require a more general result of the same flavor as those developed by Duchi and Wainwright [60]; in particular, we leverage an extension of their generalized Fano's inequality which is given as Question 7.1 in [59], whose proof we provide and adapt to our setting for completeness. We first require the following lemma:

**Lemma 8.** *Let $P$ and $Q$ be two arbitrary probability measures on the same measurable space such that $P \ll Q$. Then,*

$$D_{\mathrm{KL}}(P \,||\, Q) \geq \log\left( \frac{1}{Q(P > 0)} \right) = \log\left( \frac{1}{Q(supp\,(P))} \right).$$

*Proof.* The proof is immediate via a generalization of the traditional log-sum inequality [41]. Specifically, since $P \ll Q$, we have

$$D_{\mathrm{KL}}(P \,||\, Q) = \int \log\left( \frac{dP}{dQ} \right) dP = \int_{P>0} \log\left( \frac{dP}{dQ} \right) dP \geq \left( \int dP \right) \log\left( \frac{\int dP}{\int_{P>0} dQ} \right) = \log\left( \frac{1}{Q(P > 0)} \right).$$

$\square$

**Theorem 8.** *Take any $\Pi \subseteq \{\mathcal{S} \to \Delta(\mathcal{A})\}$ and $\mathcal{V} \subseteq \{\mathcal{S} \to \mathbb{R}\}$. For any $D \geq 0$ and any $k \in [K]$, define $\delta = \sup_{\widehat{M} \in \mathfrak{M}} \mathbb{P}(d_{\Pi,\mathcal{V}}(\mathcal{M}^\star, \widehat{M}) \leq D \mid H_k)$. Then,*

$$\sup_{\widetilde{\mathcal{M}} \in \Lambda_k(D)} \mathbb{P}(d_{\Pi,\mathcal{V}}(\mathcal{M}^\star, \widetilde{\mathcal{M}}) > D \mid H_k) \geq 1 - \frac{\mathcal{R}_k^{\Pi,\mathcal{V}}(D) + \log(2)}{\log\left( \frac{1}{\delta} \right)}.$$

*Proof.* For any episode $k \in [K]$, recall that the agent's beliefs over the true MDP $\mathcal{M}^\star$ are distributed according to $\mathbb{P}(\mathcal{M}^\star \in \cdot \mid H_k)$. Let $\widetilde{\mathcal{M}}$ be an arbitrary random variable denoting a compressed MDP

taking values in the set $\mathfrak{M}$ and, for a fixed distortion threshold $D$, we let $\mathcal{N} \subset \mathfrak{M} \times \mathfrak{M}$ denote the measurable subset of $\mathfrak{M} \times \mathfrak{M}$ that consists of all pairs of MDP which are approximately value equivalent; that is, $(M, \widehat{M}) \in \mathcal{N} \Longleftrightarrow d_{\Pi, \mathcal{V}}(M, \widehat{M}) \leq D$. For any MDP $\widehat{M} \in \mathfrak{M}$, we define a slice

$$\mathcal{N}_{\widehat{M}} \triangleq \{M \in \mathfrak{M} \mid (M, \widehat{M}) \in \mathcal{N}\},$$

as the collection of MDPs that are approximately value equivalent to a given $\widehat{M}$. In the context of Fano's inequality and our lossy compression problem, $\mathcal{N}_{\widehat{M}}$ is the set of original or uncompressed MDPs for which a channel output of $\widehat{M}$ should not be considered an error. Furthermore, define

$$p^{\max} \triangleq \sup_{\widehat{M} \in \mathfrak{M}} \mathbb{P}(\mathcal{M}^\star \in \mathcal{N}_{\widehat{M}} \mid H_k) \qquad p^{\min} \triangleq \inf_{\widehat{M} \in \mathfrak{M}} \mathbb{P}(\mathcal{M}^\star \in \mathcal{N}_{\widehat{M}} \mid H_k).$$

Recall that for $p \in [0, 1]$, we have the binary entropy function $h_2(p) = -p \log(p) - (1-p) \log(1-p)$.

Define the indicator random variable $E = \mathbb{1}((\mathcal{M}^\star, \widetilde{\mathcal{M}}) \notin \mathcal{N})$. Recalling that

$$\mathbb{I}(X; Y) = \mathbb{E}\left[D_{\mathrm{KL}}(\mathbb{P}(Y \in \cdot \mid X) \,||\, \mathbb{P}(Y \in \cdot))\right],$$

we have

$$\mathbb{I}_k(\mathcal{M}^\star; (\widetilde{\mathcal{M}}, E)) \mathbb{E}\left[D_{\mathrm{KL}}(\mathbb{P}_k(\mathcal{M}^\star \in \cdot \mid \widetilde{\mathcal{M}}, E) \,||\, \mathbb{P}_k(\mathcal{M}^\star \in \cdot))\right]$$

$$= \mathbb{P}_k(E = 1) \cdot \mathbb{E}\left[D_{\mathrm{KL}}(\mathbb{P}_k(\mathcal{M}^\star \in \cdot \mid \widetilde{\mathcal{M}}, E = 1) \,||\, \mathbb{P}_k(\mathcal{M}^\star \in \cdot))\right]$$

$$+ \mathbb{P}_k(E = 0) \cdot \mathbb{E}\left[D_{\mathrm{KL}}(\mathbb{P}_k(\mathcal{M}^\star \in \cdot \mid \widetilde{\mathcal{M}}, E = 0) \,||\, \mathbb{P}_k(\mathcal{M}^\star \in \cdot))\right].$$

At this point, we observe that for any $\widehat{M} \in \mathfrak{M}$,

$$\mathrm{supp}\left(\mathbb{P}_k(\mathcal{M}^\star \in \cdot \mid \widetilde{\mathcal{M}} = \widehat{M}, E = 0)\right) \subset \mathcal{N}_{\widehat{M}} \qquad \mathrm{supp}\left(\mathbb{P}_k(\mathcal{M}^\star \in \cdot \mid \widetilde{\mathcal{M}} = \widehat{M}, E = 1)\right) \subset \mathcal{N}_{\widehat{M}}^c,$$

by definition of the slice $\mathcal{N}_{\widehat{M}}$. Thus,

$$\mathbb{P}(\mathcal{M}^\star \in \mathrm{supp}\left(\mathbb{P}_k(\mathcal{M}^\star \in \cdot \mid \widetilde{\mathcal{M}} = \widehat{M}, E = 0)\right) \mid H_k) \leq \mathbb{P}(\mathcal{M}^\star \in \mathcal{N}_{\widehat{M}} \mid H_k)$$

$$\mathbb{P}(\mathcal{M}^\star \in \mathrm{supp}\left(\mathbb{P}_k(\mathcal{M}^\star \in \cdot \mid \widetilde{\mathcal{M}} = \widehat{M}, E = 1)\right) \mid H_k) \leq \mathbb{P}(\mathcal{M}^\star \in \mathcal{N}_{\widehat{M}}^c \mid H_k) = 1 - \mathbb{P}(\mathcal{M}^\star \in \mathcal{N}_{\widehat{M}} \mid H_k)$$

and, consequently, we have by Lemma 8 that

$$D_{\mathrm{KL}}(\mathbb{P}_k(\mathcal{M}^\star \in \cdot \mid \widetilde{\mathcal{M}} = \widehat{M}, E = 0) \,||\, \mathbb{P}_k(\mathcal{M}^\star \in \cdot)) \geq \log\left(\frac{1}{\mathbb{P}(\mathcal{M}^\star \in \mathcal{N}_{\widehat{M}} \mid H_k)}\right) \geq \log\left(\frac{1}{p^{\max}}\right),$$

$$D_{\mathrm{KL}}(\mathbb{P}_k(\mathcal{M}^\star \in \cdot \mid \widetilde{\mathcal{M}} = \widehat{M}, E = 1) \,||\, \mathbb{P}_k(\mathcal{M}^\star \in \cdot)) \geq \log\left(\frac{1}{1 - \mathbb{P}(\mathcal{M}^\star \in \mathcal{N}_{\widehat{M}} \mid H_k)}\right) \geq \log\left(\frac{1}{1 - p^{\min}}\right).$$

Applying these lower bounds to our original mutual information term, we see that

$$\mathbb{I}_k(\mathcal{M}^\star; (\widetilde{\mathcal{M}}, E)) \geq \mathbb{P}(E = 1 \mid H_k) \log\left(\frac{1}{1 - p^{\min}}\right) + \mathbb{P}(E = 0 \mid H_k) \log\left(\frac{1}{p^{\max}}\right)$$

$$= \mathbb{P}(E = 1 \mid H_k) \log\left(\frac{1}{1 - p^{\min}}\right) + (1 - \mathbb{P}(E = 1 \mid H_k)) \log\left(\frac{1}{p^{\max}}\right)$$

$$= \mathbb{P}(E = 1 \mid H_k) \log\left(\frac{p^{\max}}{1 - p^{\min}}\right) + \log\left(\frac{1}{p^{\max}}\right).$$

Now applying the chain rule of mutual information, the definition of mutual information, the non-negativity of entropy and the fact that conditioning reduces entropy in sequence, we obtain

$$\mathbb{I}_k(\mathcal{M}^\star; (\widetilde{\mathcal{M}}, E)) = \mathbb{I}_k(\mathcal{M}^\star; \widetilde{\mathcal{M}}) + \mathbb{I}_k(\mathcal{M}^\star; E \mid \widetilde{\mathcal{M}})$$

$$= \mathbb{I}_k(\mathcal{M}^\star; \widetilde{\mathcal{M}}) + \mathbb{H}_k(E \mid \widetilde{\mathcal{M}}) - \mathbb{H}_k(E \mid \widetilde{\mathcal{M}}, \mathcal{M}^\star)$$

$$\leq \mathbb{I}_k(\mathcal{M}^\star; \widetilde{\mathcal{M}}) + \mathbb{H}_k(E \mid \widetilde{\mathcal{M}})$$

$$\leq \mathbb{I}_k(\mathcal{M}^\star; \widetilde{\mathcal{M}}) + \mathbb{H}_k(E)$$

$$\leq \mathbb{I}_k(\mathcal{M}^\star; \widetilde{\mathcal{M}}) + \mathbb{H}(E)$$

Combining the upper and lower bounds while multiplying through by $-1$ yields

$$h_2(\mathbb{P}(E=1)) + \mathbb{P}(E=1 \mid H_k)\log\left(\frac{1-p^{\min}}{p^{\max}}\right) \geq \log\left(\frac{1}{p^{\max}}\right) - \mathbb{I}_k(\mathcal{M}^\star; \widetilde{\mathcal{M}}).$$

Recognizing that we have the following upper bounds

$$\log(2) + \mathbb{P}(E=1 \mid H_k)\log\left(\frac{1}{p^{\max}}\right) \geq h_2(\mathbb{P}(E=1)) + \mathbb{P}(E=1 \mid H_k)\log\left(\frac{1}{p^{\max}}\right)$$

$$\geq h_2(\mathbb{P}(E=1)) + \mathbb{P}(E=1 \mid H_k)\log\left(\frac{1-p^{\min}}{p^{\max}}\right),$$

and re-arranging terms yields

$$\mathbb{P}(E=1 \mid H_k) \geq \frac{\log\left(\frac{1}{p^{\max}}\right) - \mathbb{I}_k(\mathcal{M}^\star; \widetilde{\mathcal{M}}) - \log(2)}{\log\left(\frac{1}{p^{\max}}\right)} = 1 - \frac{\mathbb{I}_k(\mathcal{M}^\star; \widetilde{\mathcal{M}}) + \log(2)}{\log\left(\frac{1}{\delta}\right)},$$

where $\delta = \sup_{\widehat{M} \in \mathfrak{M}} \mathbb{P}(d_{\Pi,\mathcal{V}}(\mathcal{M}^\star, \widehat{M}) \leq D \mid H_k)$. Noting that

$$\mathbb{P}(E=1 \mid H_k) = \mathbb{P}((\mathcal{M}^\star, \widetilde{\mathcal{M}}) \notin \mathcal{N} \mid H_k) = \mathbb{P}(d_{\Pi,\mathcal{V}}(\mathcal{M}^\star, \widetilde{\mathcal{M}}) > D \mid H_k),$$

and taking the supremum on both sides, we have

$$\sup_{\widetilde{\mathcal{M}} \in \Lambda_k(D)} \mathbb{P}(d_{\Pi,\mathcal{V}}(\mathcal{M}^\star, \widetilde{\mathcal{M}}) > D \mid H_k) \geq \sup_{\widetilde{\mathcal{M}} \in \Lambda_k(D)}\left[1 - \frac{\mathbb{I}_k(\mathcal{M}^\star; \widetilde{\mathcal{M}}) + \log(2)}{\log\left(\frac{1}{\delta}\right)}\right]$$

$$= 1 - \inf_{\widetilde{\mathcal{M}} \in \Lambda_k(D)} \frac{\mathbb{I}_k(\mathcal{M}^\star; \widetilde{\mathcal{M}}) + \log(2)}{\log\left(\frac{1}{\delta}\right)}$$

$$= 1 - \frac{\mathcal{R}_k^{\Pi,\mathcal{V}}(D) + \log(2)}{\log\left(\frac{1}{\delta}\right)},$$

as desired. $\qquad\square$

## H   Proof of Theorem 3

In specializing to the tabular MDP setting, we wish to simplify our information-theoretic Bayesian regret bound (Corollary 1) into one that only depends on the standard problem-specific quantities ($|\mathcal{S}|, |\mathcal{A}|, K, H$). To do this, we will necessarily decompose mutual information into its constituent entropy terms. Inconveniently, while mutual information is well-defined for arbitrary random variables, entropy is infinite for continuous random variables (like the reward function and transition function random variables, $\mathcal{R}^\star$ and $\mathcal{T}^\star$). Rather than resorting to differential entropy, which lacks several desirable properties of Shannon entropy, we explicitly replace these random variables by their discretized analogues, obtained via a sufficiently-fine quantization of their ranges a priori such that the differential entropy of the original random variables is well-approximated by the associated metric entropy or $\varepsilon$-entropy [93], courtesy of Theorem 8.3.1 of [41].

Recall that, for any $\varepsilon > 0$, a $\varepsilon$-cover of a set $\Theta$ with respect to a (semi)-metric $\rho: \Theta \times \Theta \to \mathbb{R}_{\geq 0}$ is a set $\{\theta_1, \ldots, \theta_N\}$ with $\theta_i \in \Theta$, $\forall i \in [N]$, such that for any other point $\theta \in \Theta$, $\exists n \in [N]$ such that $\rho(\theta, \theta_n) \leq \varepsilon$. The $\varepsilon$-covering number of $\Theta$ is defined as

$$\mathcal{N}(\varepsilon, \Theta, \rho) \triangleq \inf\{N \in \mathbb{N} \mid \exists \text{ an } \varepsilon\text{-cover } \{\theta_1, \ldots, \theta_N\} \text{ of } \Theta\}.$$

Conversely, a $\varepsilon$-packing of a set $\Theta$ with respect to $\rho$ is a set $\{\theta_1, \ldots, \theta_M\}$ with $\theta_i \in \Theta$, $\forall i \in [M]$, such that for any distinct $i, j \in [N]$, we have $\rho(\theta_i, \theta_j) \geq \varepsilon$. The $\varepsilon$-packing number of a set $\Theta$ is defined as

$$\mathcal{M}(\varepsilon, \Theta, \rho) \triangleq \sup\{M \in \mathbb{N} \mid \exists \text{ an } \varepsilon\text{-packing } \{\theta_1, \ldots, \theta_M\} \text{ of } \Theta\}.$$

With slight abuse of notation, for any norm $||\cdot||$ on a set $\Theta$, we write $\mathcal{N}(\varepsilon, \Theta, ||\cdot||)$ to denote the $\varepsilon$-covering number under the metric induced by $||\cdot||$, and similarly for the $\varepsilon$-packing number $\mathcal{M}(\varepsilon, \Theta, ||\cdot||)$. Theorem IV of [93] establishes the following relationship between the $\varepsilon$-covering number and $\varepsilon$-packing number that we will use to upper bound metric entropy:

**Fact 2.** *For any metric space $(\Theta, \rho)$ and any $\varepsilon > 0$, $\mathcal{N}(\varepsilon, \Theta, \rho) \leq \mathcal{M}(\varepsilon, \Theta, \rho)$.*

This allows for a generalization of Lemma 7.6 of [59] to norm balls of arbitrary radius whose proof we include for completeness.

**Lemma 9.** *For any norm $|| \cdot ||$, let $\mathbb{B}^d = \{\theta \in \mathbb{R}^d \mid ||\theta|| \leq 1\}$ denote the unit $|| \cdot ||$-ball in $\mathbb{R}^d$. For any $r \in (0, \infty)$, we let $r\mathbb{B}^d = \{\theta \in \mathbb{R}^d \mid ||\theta|| \leq r\}$ denote the scaling of the unit ball by $r$ or, equivalently, the $|| \cdot ||$-ball of radius $r$. Then, for any $\varepsilon \in (0, r]$,*

$$\log\left(\mathcal{N}(\varepsilon, r\mathbb{B}^d, || \cdot ||)\right) \leq d \log\left(1 + \frac{2r}{\varepsilon}\right).$$

*Proof.* Let $\mathrm{Vol}\,(\cdot)$ be the function that denotes the volume of an input ball in $\mathbb{R}^d$ such that $\mathrm{Vol}\,(r\mathbb{B}^d) = r^d$. Since an $\varepsilon$-packing requires filling $r\mathbb{B}^d$ with disjoint balls of diameter $\varepsilon$, we have

$$\mathcal{M}(\varepsilon, r\mathbb{B}^d, || \cdot ||)\mathrm{Vol}\left(\frac{\varepsilon}{2}\mathbb{B}^d\right) = \sum_{i=1}^{\mathcal{M}(\varepsilon, r\mathbb{B}^d, ||\cdot||)} \mathrm{Vol}\left(\frac{\varepsilon}{2}\mathbb{B}^d\right) \leq \mathrm{Vol}\left(\left(r + \frac{\varepsilon}{2}\right)\mathbb{B}^d\right).$$

Dividing through by $\mathrm{Vol}\left(\frac{\varepsilon}{2}\mathbb{B}^d\right)$ yields

$$\mathcal{M}(\varepsilon, r\mathbb{B}^d, || \cdot ||) \leq \frac{\mathrm{Vol}\left(\left(r + \frac{\varepsilon}{2}\right)\mathbb{B}^d\right)}{\mathrm{Vol}\left(\frac{\varepsilon}{2}\mathbb{B}^d\right)} = \left(\frac{r + \frac{\varepsilon}{2}}{\frac{\varepsilon}{2}}\right)^d = \left(1 + \frac{2r}{\varepsilon}\right)^d.$$

Applying Fact 2 gives us

$$\mathcal{N}(\varepsilon, r\mathbb{B}^d, || \cdot ||) \leq \mathcal{M}(\varepsilon, r\mathbb{B}^d, || \cdot ||) \leq \left(1 + \frac{2r}{\varepsilon}\right)^d,$$

and taking logarithms on both sides renders the desired inequality. $\qquad\square$

**Theorem 9.** *Take any $\Pi \supseteq \{\mathcal{S} \to \mathcal{A}\}$, any $\mathcal{V} \supseteq \{V^\pi \mid \pi \in \Pi^H\}$, and let $D = 0$. For any prior distribution $\mathbb{P}(\mathcal{M}^\star \in \cdot \mid H_1)$ over tabular MDPs, if $\Gamma_k \leq \overline{\Gamma}$ for all $k \in [K]$, then VSRL (Algorithm 2) has*

$$\mathrm{BAYESREGRET}(K, \pi^{(1)}, \ldots, \pi^{(K)}) \leq \mathcal{O}\left(|\mathcal{S}|\sqrt{\overline{\Gamma}|\mathcal{A}|K}\right).$$

*Proof.* Using Fact 1, we have that

$$\mathcal{R}_1^{\Pi, \mathcal{V}}(D) \leq \mathbb{H}_1(\mathcal{M}^\star) = \mathbb{H}_1(\mathcal{R}^\star, \mathcal{T}^\star) = \mathbb{H}_1(\mathcal{R}^\star) + \mathbb{H}_1(\mathcal{T}^\star \mid \mathcal{R}^\star) = \mathbb{H}_1(\mathcal{R}^\star) + \mathbb{H}_1(\mathcal{T}^\star),$$

where the first equality recognizes that all randomness in the true MDP $\mathcal{M}^\star$ is driven by the model $(\mathcal{R}^\star, \mathcal{T}^\star)$, the second equality applies the chain rule of entropy, and the final equality recognizes that the reward function and transition function random variables are independent.

For some fixed $\varepsilon_\mathcal{R} > 0$, consider the $\frac{\varepsilon_\mathcal{R}}{2}$-cover of the unit interval $[0, 1]$ with respect to the $L_1$-norm $|| \cdot ||_1$ as a quantization into bins of width $\varepsilon_\mathcal{R}$. Observe that the true environment reward function $\mathcal{R}^\star : \mathcal{S} \times \mathcal{A} \to [0, 1]$ is well-approximated by mapping state-action pairs onto this $\frac{\varepsilon_\mathcal{R}}{2}$-cover, for a sufficiently small $\varepsilon_\mathcal{R} > 0$. Consequently, we treat $\mathcal{R}^\star$ as a discrete random variable where $|\mathrm{supp}(\mathcal{R}^\star)| = \mathcal{N}(\frac{\varepsilon_\mathcal{R}}{2}, [0, 1], || \cdot ||_1)^{|\mathcal{S}||\mathcal{A}|}$. Recall that, for a discrete random variable $X$ with support on $\mathcal{X}$, $\mathbb{H}(X) \leq \log(|\mathcal{X}|)$. Applying this upper bound and Lemma 9 in sequence, we have that

$$\mathbb{H}_1(\mathcal{R}^\star) \leq |\mathcal{S}||\mathcal{A}| \log\left(\mathcal{N}(\frac{\varepsilon_\mathcal{R}}{2}, [0, 1], || \cdot ||_1)\right) \leq |\mathcal{S}||\mathcal{A}| \log\left(1 + \frac{4}{\varepsilon_\mathcal{R}}\right).$$

Applying the same sequence of steps *mutatis mutandis* for the transition function $\mathcal{T}^\star$ under a $\frac{\varepsilon_\mathcal{T}}{2}$-cover, for some fixed $\varepsilon_\mathcal{T} > 0$, we also have

$$\mathbb{H}_1(\mathcal{T}^\star) \leq |\mathcal{S}|^2|\mathcal{A}| \log\left(\mathcal{N}(\frac{\varepsilon_\mathcal{T}}{2}, [0, 1], || \cdot ||_1)\right) \leq |\mathcal{S}|^2|\mathcal{A}| \log\left(1 + \frac{4}{\varepsilon_\mathcal{T}}\right).$$

Applying these bounds following the earlier rate-distortion function upper bound to the result of Corollary 1 with $D = 0$, we have

$$\mathrm{BAYESREGRET}(K, \pi^{(1)}, \ldots, \pi^{(K)}) \leq \sqrt{\overline{\Gamma}K\left(|\mathcal{S}||\mathcal{A}| \log\left(1 + \frac{4}{\varepsilon_\mathcal{R}}\right) + |\mathcal{S}|^2|\mathcal{A}| \log\left(1 + \frac{4}{\varepsilon_\mathcal{T}}\right)\right)}.$$

$\qquad\square$

# I Proof of Theorem 4

Our proof of Theorem 4 utilizes the following fact, widely known as the performance-difference lemma, adapted to the finite-horizon setting whose proof we replicate here.

**Lemma 10** (Performance-Difference Lemma [89]). *For any finite-horizon MDP $\langle \mathcal{S}, \mathcal{A}, \mathcal{R}, \mathcal{T}, \beta, H \rangle$ and any two non-stationary policies $\pi_1, \pi_2 \in \Pi^H$, let $\rho^{\pi_2}(\tau)$ denote the distribution over trajectories induced by policy $\pi_2$. Then,*

$$V_1^{\pi_1} - V_1^{\pi_2} = \mathbb{E}_{\tau \sim \rho^{\pi_2}} \left[ \sum_{h=1}^{H} \left( V_h^{\pi_1}(s_h) - Q_h^{\pi_1}(s_h, a_h) \right) \right].$$

*Proof.*

$$
\begin{aligned}
V_1^{\pi_1} - V_1^{\pi_2} &= \mathbb{E}_{s_1 \sim \beta} \left[ V_1^{\pi_1}(s_1) - V_1^{\pi_2}(s_1) \right] \\
&= \mathbb{E}_{s_1 \sim \beta} \left[ V_1^{\pi_1}(s_1) - \mathbb{E}_{\tau \sim \rho^{\pi_2}} \left[ \sum_{h=1}^{H} \mathcal{R}(s_h, a_h) \mid s_1 \right] \right] \\
&= \mathbb{E}_{\tau \sim \rho^{\pi_2}} \left[ V_1^{\pi_1}(s_1) - \sum_{h=1}^{H} \mathcal{R}(s_h, a_h) \right] \\
&= \mathbb{E}_{\tau \sim \rho^{\pi_2}} \left[ V_1^{\pi_1}(s_1) + \sum_{h=2}^{H} V_h^{\pi_1}(s_h) - \sum_{h=1}^{H} \left( \mathcal{R}(s_h, a_h) - V_{h+1}^{\pi_1}(s_{h+1}) \right) \right] \\
&= \mathbb{E}_{\tau \sim \rho^{\pi_2}} \left[ \sum_{h=1}^{H} V_h^{\pi_1}(s_h) - \left( \mathcal{R}(s_h, a_h) + V_{h+1}^{\pi_1}(s_{h+1}) \right) \right] \\
&= \mathbb{E}_{\tau \sim \rho^{\pi_2}} \left[ \sum_{h=1}^{H} \left( V_h^{\pi_1}(s_h) - \left( \mathcal{R}(s_h, a_h) + \mathbb{E} \left[ V_{h+1}^{\pi_1}(s_{h+1}) \mid s_h, a_h \right] \right) \right) \right] \\
&= \mathbb{E}_{\tau \sim \rho^{\pi_2}} \left[ \sum_{h=1}^{H} \left( V_h^{\pi_1}(s_h) - Q_h^{\pi_1}(s_h, a_h) \right) \right],
\end{aligned}
$$

where the penultimate line invokes the tower property of expectation. $\qquad \square$

**Theorem 10.** *Fix any $D \geq 0$ and, for each episode $k \in [K]$, let $\widetilde{\mathcal{M}}_k$ be any MDP that achieves the rate-distortion limit of $\mathcal{R}_k^{Q^\star}(D)$ with information source $\mathbb{P}(\mathcal{M}^\star \in \cdot \mid H_k)$ and distortion function $d_{Q^\star}$. Then,*

$$\text{BAYESREGRET}(K, \pi^{(1)}, \dots, \pi^{(K)}) \leq \mathbb{E} \left[ \sum_{k=1}^{K} \mathbb{E}_k \left[ V_{\widetilde{\mathcal{M}}_k, 1}^\star - V_{\widetilde{\mathcal{M}}_k, 1}^{\pi^{(k)}} \right] \right] + (2H + 2) K \sqrt{D}.$$

*Proof.* By applying definitions from Section 3 and applying the tower property of expectation, we have that

$$\text{BAYESREGRET}(K, \pi^{(1)}, \dots, \pi^{(K)}) = \mathbb{E} \left[ \sum_{k=1}^{K} \mathbb{E}_k \left[ \Delta_k \right] \right].$$

Examining the $k$th episode in isolation and applying the definition of episodic regret, we have

$$\mathbb{E}_k\left[\Delta_k\right] = \mathbb{E}_k\left[V^\star_{\mathcal{M}^\star,1} - V^{\pi^{(k)}}_{\mathcal{M}^\star,1}\right]$$

$$= \mathbb{E}_k\left[V^\star_{\mathcal{M}^\star,1} - V^\star_{\widetilde{\mathcal{M}}_k,1} + V^\star_{\widetilde{\mathcal{M}}_k,1} - V^{\pi^{(k)}}_{\widetilde{\mathcal{M}}_k,1} + V^{\pi^{(k)}}_{\widetilde{\mathcal{M}}_k,1} - V^{\pi^{(k)}}_{\mathcal{M}^\star,1}\right]$$

$$= \mathbb{E}_k\left[V^\star_{\mathcal{M}^\star,1} - V^\star_{\widetilde{\mathcal{M}}_k,1} + V^\star_{\widetilde{\mathcal{M}}_k,1} - V^{\pi^{(k)}}_{\widetilde{\mathcal{M}}_k,1} + \underbrace{V^{\pi^{(k)}}_{\widetilde{\mathcal{M}}_k,1} - V^\star_{\widetilde{\mathcal{M}}_k,1}}_{\leq 0} + V^\star_{\widetilde{\mathcal{M}}_k,1} - V^{\pi^{(k)}}_{\mathcal{M}^\star,1}\right]$$

$$\leq \mathbb{E}_k\left[V^\star_{\mathcal{M}^\star,1} - V^\star_{\widetilde{\mathcal{M}}_k,1} + V^\star_{\widetilde{\mathcal{M}}_k,1} - V^{\pi^{(k)}}_{\widetilde{\mathcal{M}}_k,1} + V^\star_{\widetilde{\mathcal{M}}_k,1} - V^{\pi^{(k)}}_{\mathcal{M}^\star,1}\right]$$

$$= \mathbb{E}_k\left[V^\star_{\mathcal{M}^\star,1} - V^\star_{\widetilde{\mathcal{M}}_k,1} + V^\star_{\widetilde{\mathcal{M}}_k,1} - V^{\pi^{(k)}}_{\widetilde{\mathcal{M}}_k,1} + V^\star_{\widetilde{\mathcal{M}}_k,1} - V^\star_{\mathcal{M}^\star,1} + V^\star_{\mathcal{M}^\star,1} - V^{\pi^{(k)}}_{\mathcal{M}^\star,1}\right].$$

Observe that

$$\mathbb{E}_k\left[V^\star_{\mathcal{M}^\star,1} - V^\star_{\widetilde{\mathcal{M}}_k,1}\right] \leq \mathbb{E}_k\left[\|V^\star_{\mathcal{M}^\star,1} - V^\star_{\widetilde{\mathcal{M}}_k,1}\|_\infty\right]$$

$$= \mathbb{E}_k\left[\max_{s\in\mathcal{S}}|V^\star_{\mathcal{M}^\star,1}(s) - V^\star_{\widetilde{\mathcal{M}}_k,1}(s)|\right]$$

$$= \mathbb{E}_k\left[\max_{s\in\mathcal{S}}|\max_{a\in\mathcal{A}}Q^\star_{\mathcal{M}^\star,1}(s,a) - \max_{a'\in\mathcal{A}}Q^\star_{\widetilde{\mathcal{M}}_k,1}(s,a')|\right]$$

$$\leq \mathbb{E}_k\left[\max_{s\in\mathcal{S}}\max_{a\in\mathcal{A}}|Q^\star_{\mathcal{M}^\star,1}(s,a) - Q^\star_{\widetilde{\mathcal{M}}_k,1}(s,a)|\right]$$

$$= \mathbb{E}_k\left[\|Q^\star_{\mathcal{M}^\star,1} - Q^\star_{\widetilde{\mathcal{M}}_k,1}\|_\infty\right]$$

$$= \mathbb{E}_k\left[\sqrt{\|Q^\star_{\mathcal{M}^\star,1} - Q^\star_{\widetilde{\mathcal{M}}_k,1}\|^2_\infty}\right]$$

$$\leq \sqrt{\mathbb{E}_k\left[\|Q^\star_{\mathcal{M}^\star,1} - Q^\star_{\widetilde{\mathcal{M}}_k,1}\|^2_\infty\right]}$$

$$\leq \sqrt{\mathbb{E}_k\left[\sup_{h\in H}\|Q^\star_{\mathcal{M}^\star,h} - Q^\star_{\widetilde{\mathcal{M}}_k,h}\|^2_\infty\right]}$$

$$= \sqrt{\mathbb{E}_k\left[d_{Q^\star}(\mathcal{M}^\star,\widetilde{\mathcal{M}}_k)\right]}$$

$$\leq \sqrt{D},$$

where the penultimate inequality is due to Jensen's inequality and the final inequality holds as $\widetilde{\mathcal{M}}_k$ achieves the rate-distortion limit under $d_{Q^\star}$, by assumption. Moreover, the exact argument can be repeated to see that

$$\mathbb{E}_k\left[V^\star_{\widetilde{\mathcal{M}}_k,1} - V^\star_{\mathcal{M}^\star,1}\right] \leq \mathbb{E}_k\left[\|V^\star_{\widetilde{\mathcal{M}}_k,1} - V^\star_{\mathcal{M}^\star,1}\|_\infty\right]$$

$$= \mathbb{E}_k\left[\|V^\star_{\mathcal{M}^\star,1} - V^\star_{\widetilde{\mathcal{M}}_k,1}\|_\infty\right]$$

$$\leq \sqrt{D}.$$

Combining these two inequalities yields

$$\mathbb{E}_k\left[\Delta_k\right] \leq \mathbb{E}_k\left[V^\star_{\mathcal{M}^\star,1} - V^\star_{\widetilde{\mathcal{M}}_k,1} + V^\star_{\widetilde{\mathcal{M}}_k,1} - V^{\pi^{(k)}}_{\widetilde{\mathcal{M}}_k,1} + V^\star_{\widetilde{\mathcal{M}}_k,1} - V^\star_{\mathcal{M}^\star,1} + V^\star_{\mathcal{M}^\star,1} - V^{\pi^{(k)}}_{\mathcal{M}^\star,1}\right]$$

$$\leq \mathbb{E}_k\left[V^\star_{\widetilde{\mathcal{M}}_k,1} - V^{\pi^{(k)}}_{\widetilde{\mathcal{M}}_k,1} + V^\star_{\mathcal{M}^\star,1} - V^{\pi^{(k)}}_{\mathcal{M}^\star,1}\right] + 2\sqrt{D}.$$

Observe that by virtue of posterior sampling [135, 121, 120] the compressed MDP being targeted by the agent $\widetilde{\mathcal{M}}_k$ and the sampled MDP $M_k$ are identically distributed, conditioned upon the information available within any history $H_k$, and so we have

$$\mathbb{E}_k\left[V^\star_{\mathcal{M}^\star,1} - V^{\pi^{(k)}}_{\mathcal{M}^\star,1}\right] = \mathbb{E}_k\left[V^\star_{\mathcal{M}^\star,1} - V^{\pi^\star_{M_k}}_{\mathcal{M}^\star,1}\right] = \mathbb{E}_k\left[V^\star_{\mathcal{M}^\star,1} - V^{\pi^\star_{\widetilde{\mathcal{M}}_k}}_{\mathcal{M}^\star,1}\right].$$

Now applying the performance-difference lemma (Lemma 10), we see that

$$\mathbb{E}_k \left[ V^\star_{\mathcal{M}^\star,1} - V^{\pi^\star_{\widetilde{\mathcal{M}}_k}}_{\mathcal{M}^\star,1} \right] = \mathbb{E}_k \left[ \mathbb{E}_{\rho^{\pi^\star_{\widetilde{\mathcal{M}}_k}}} \left[ \sum_{h=1}^H \left( V^\star_{\mathcal{M}^\star,h}(s_h) - Q^\star_{\mathcal{M}^\star,h}(s_h,a_h) \right) \right] \right]$$

$$= \mathbb{E}_k \left[ \mathbb{E}_{\rho^{\pi^\star_{\widetilde{\mathcal{M}}_k}}} \left[ \sum_{h=1}^H \left( \max_{a \in \mathcal{A}} Q^\star_{\mathcal{M}^\star,h}(s_h,a) - Q^\star_{\mathcal{M}^\star,h}(s_h,a_h) \right) \right] \right]$$

$$\leq \mathbb{E}_k \left[ \mathbb{E}_{\rho^{\pi^\star_{\widetilde{\mathcal{M}}_k}}} \left[ \sum_{h=1}^H \left| \max_{a \in \mathcal{A}} Q^\star_{\mathcal{M}^\star,h}(s_h,a) - Q^\star_{\mathcal{M}^\star,h}(s_h,a_h) \right| \right] \right].$$

Define $a^\star = \arg\max_{a \in \mathcal{A}} Q^\star_{\mathcal{M}^\star,h}(s_h,a)$ such that

$$\mathbb{E}_k \left[ V^\star_{\mathcal{M}^\star,1} - V^{\pi^\star_{\widetilde{\mathcal{M}}_k}}_{\mathcal{M}^\star,1} \right] = \mathbb{E}_k \left[ \mathbb{E}_{\rho^{\pi^\star_{\widetilde{\mathcal{M}}_k}}} \left[ \sum_{h=1}^H \left| \max_{a \in \mathcal{A}} Q^\star_{\mathcal{M}^\star,h}(s_h,a) - Q^\star_{\mathcal{M}^\star,h}(s_h,a_h) \right| \right] \right]$$

$$= \mathbb{E}_k \left[ \mathbb{E}_{\rho^{\pi^\star_{\widetilde{\mathcal{M}}_k}}} \left[ \sum_{h=1}^H \left| Q^\star_{\mathcal{M}^\star,h}(s_h,a^\star) - Q^\star_{\mathcal{M}^\star,h}(s_h,a_h) \right| \right] \right]$$

$$= \mathbb{E}_k \left[ \mathbb{E}_{\rho^{\pi^\star_{\widetilde{\mathcal{M}}_k}}} \left[ \sum_{h=1}^H \left| Q^\star_{\mathcal{M}^\star,h}(s_h,a^\star) - Q^\star_{\widetilde{\mathcal{M}}_k,h}(s_h,a^\star) + Q^\star_{\widetilde{\mathcal{M}}_k,h}(s_h,a^\star) - Q^\star_{\mathcal{M}^\star,h}(s_h,a_h) \right| \right] \right].$$

Applying the triangle inequality and examining each difference in isolation, we have

$$\mathbb{E}_k \left[ \mathbb{E}_{\rho^{\pi^\star_{\widetilde{\mathcal{M}}_k}}} \left[ \sum_{h=1}^H \left| Q^\star_{\mathcal{M}^\star,h}(s_h,a^\star) - Q^\star_{\widetilde{\mathcal{M}}_k,h}(s_h,a^\star) \right| \right] \right] \leq \mathbb{E}_k \left[ \mathbb{E}_{\rho^{\pi^\star_{\widetilde{\mathcal{M}}_k}}} \left[ \sum_{h=1}^H \|Q^\star_{\mathcal{M}^\star,h} - Q^\star_{\widetilde{\mathcal{M}}_k,h}\|_\infty \right] \right]$$

$$\leq H \mathbb{E}_k \left[ \sup_{h \in H} \|Q^\star_{\mathcal{M}^\star,h} - Q^\star_{\widetilde{\mathcal{M}}_k,h}\|_\infty \right]$$

$$= H \mathbb{E}_k \left[ \sup_{h \in H} \sqrt{\|Q^\star_{\mathcal{M}^\star,h} - Q^\star_{\widetilde{\mathcal{M}}_k,h}\|_\infty^2} \right]$$

$$\leq H \sqrt{\mathbb{E}_k \left[ \sup_{h \in H} \|Q^\star_{\mathcal{M}^\star,h} - Q^\star_{\widetilde{\mathcal{M}}_k,h}\|_\infty^2 \right]}$$

$$= H \sqrt{\mathbb{E}_k \left[ d_{Q^\star}(\mathcal{M}^\star, \widetilde{\mathcal{M}}_k) \right]}$$

$$\leq H \sqrt{D},$$

where the penultimate inequality follows from Jensen's inequality and the final inequality follows since $\widetilde{\mathcal{M}}_k$ achieves the rate-distortion limit.

For the remaining term, we have

$$\mathbb{E}_k \left[ V^\star_{\mathcal{M}^\star,1} - V^{\pi^\star_{\widetilde{\mathcal{M}}_k}}_{\mathcal{M}^\star,1} \right] \leq H\sqrt{D} + \mathbb{E}_k \left[ \mathbb{E}_{\rho^{\pi^\star_{\widetilde{\mathcal{M}}_k}}} \left[ \sum_{h=1}^H \left| Q^\star_{\widetilde{\mathcal{M}}_k,h}(s_h,a^\star) - Q^\star_{\mathcal{M}^\star,h}(s_h,a_h) \right| \right] \right]$$

$$= H\sqrt{D} + \mathbb{E}_k \left[ \mathbb{E}_{\rho^{\pi^\star_{\widetilde{\mathcal{M}}_k}}} \left[ \sum_{h=1}^H \left| Q^\star_{\widetilde{\mathcal{M}}_k,h}(s_h,a^\star) - Q^\star_{\widetilde{\mathcal{M}}_k,h}(s_h,a_h) + Q^\star_{\widetilde{\mathcal{M}}_k,h}(s_h,a_h) - Q^\star_{\mathcal{M}^\star,h}(s_h,a_h) \right| \right] \right]$$

$$\leq H\sqrt{D} + \mathbb{E}_k \left[ \mathbb{E}_{\rho^{\pi^\star_{\widetilde{\mathcal{M}}_k}}} \left[ \sum_{h=1}^H \left| Q^\star_{\widetilde{\mathcal{M}}_k,h}(s_h,a_h) - Q^\star_{\mathcal{M}^\star,h}(s_h,a_h) \right| \right] \right],$$

where the inequality follows since $a_h$ is drawn from the optimal policy of $\widetilde{\mathcal{M}}_k$, $\pi^\star_{\widetilde{\mathcal{M}}_k}$, and so $Q^\star_{\widetilde{\mathcal{M}}_k,h}(s_h, a_h) \geq Q^\star_{\widetilde{\mathcal{M}}_k,h}(s_h, a^\star)$. Repeating the identical argument from above yields

$$\mathbb{E}_k \left[ \mathbb{E}_{\rho^{\pi^\star_{\widetilde{\mathcal{M}}_k}}} \left[ \sum_{h=1}^{H} \left| Q^\star_{\widetilde{\mathcal{M}}_k,h}(s_h, a_h) - Q^\star_{\mathcal{M}^\star,h}(s_h, a_h) \right| \right] \right] \leq \mathbb{E}_k \left[ \mathbb{E}_{\rho^{\pi^\star_{\widetilde{\mathcal{M}}_k}}} \left[ \sum_{h=1}^{H} ||Q^\star_{\mathcal{M}^\star,h} - Q^\star_{\widetilde{\mathcal{M}}_k,h}||_\infty \right] \right]$$
$$\leq H\sqrt{D}.$$

Substituting back, we see that

$$\mathbb{E}_k \left[ V^\star_{\mathcal{M}^\star,1} - V^{\pi^{(k)}}_{\mathcal{M}^\star,1} \right] \leq \mathbb{E}_k \left[ \mathbb{E}_{\rho^{\pi^\star_{\widetilde{\mathcal{M}}_k}}} \left[ \sum_{h=1}^{H} \left| \max_{a \in \mathcal{A}} Q^\star_{\mathcal{M}^\star,h}(s_h, a) - Q^\star_{\mathcal{M}^\star,h}(s_h, a_h) \right| \right] \right] \leq 2H\sqrt{D}.$$

Thus, we may complete our bound as

$$\mathbb{E}_k [\Delta_k] \leq \mathbb{E}_k \left[ V^\star_{\mathcal{M}^\star,1} - V^\star_{\widetilde{\mathcal{M}}_k,1} + V^\star_{\widetilde{\mathcal{M}}_k,1} - V^{\pi^{(k)}}_{\widetilde{\mathcal{M}}_k,1} + V^\star_{\widetilde{\mathcal{M}}_k,1} - V^\star_{\mathcal{M}^\star,1} + V^\star_{\mathcal{M}^\star,1} - V^{\pi^{(k)}}_{\mathcal{M}^\star,1} \right]$$
$$\leq \mathbb{E}_k \left[ V^\star_{\widetilde{\mathcal{M}}_k,1} - V^{\pi^{(k)}}_{\widetilde{\mathcal{M}}_k,1} + V^\star_{\mathcal{M}^\star,1} - V^{\pi^{(k)}}_{\mathcal{M}^\star,1} \right] + 2\sqrt{D}$$
$$\leq \mathbb{E}_k \left[ V^\star_{\widetilde{\mathcal{M}}_k,1} - V^{\pi^{(k)}}_{\widetilde{\mathcal{M}}_k,1} \right] + (2H + 2)\sqrt{D}.$$

Applying this upper bound on episodic regret in each episode yields

$$\textsc{BayesRegret}(K, \pi^{(1)}, \ldots, \pi^{(K)}) = \mathbb{E}\left[ \sum_{k=1}^{K} \mathbb{E}_k [\Delta_k] \right]$$
$$\leq \mathbb{E}\left[ \sum_{k=1}^{K} \mathbb{E}_k \left[ V^\star_{\widetilde{\mathcal{M}}_k,1} - V^{\pi^{(k)}}_{\widetilde{\mathcal{M}}_k,1} \right] \right] + 2K(H+1)\sqrt{D},$$

as desired. $\qquad\square$

## J  Proof of Lemma 3

To show Lemma 3, we prove the following more general result which applies whenever a distortion function adheres to a specific functional form.

Let $V, \widehat{V}$ be two arbitrary random variables defined on the same measurable space $(\mathcal{V}, \mathbb{V})$ and define the associated rate-distortion function as

$$\mathcal{R}(D) = \inf_{\widehat{V} \in \Lambda(D)} \mathbb{I}(V; \widehat{V}) = \inf_{\widehat{V} \in \Lambda(D)} D_{\mathrm{KL}}(\mathbb{P}((V, \widehat{V}) \in \cdot) \,||\, \mathbb{P}(V \in \cdot) \times \mathbb{P}(\widehat{V} \in \cdot)),$$

where the distortion function $d : \mathcal{V} \times \mathcal{V} \to \mathbb{R}_{\geq 0}$ has the form $d(v, \widehat{v}) = \ell(f(v), f(\widehat{v}))$ for any two known, deterministic functions, $f : \mathcal{V} \to \mathcal{Z}$ and a semi-metric $\ell : \mathcal{Z} \times \mathcal{Z} \to \mathbb{R}_{\geq 0}$. Effectively, this structural constraint says that our distortion measure between the original $V$ and compressed $\widehat{V}$ only depends on the statistics $f(V)$ and $f(\widehat{V})$. Under such a constraint, we may prove the following lemma

**Lemma 11.** *If $D = 0$ and $\widehat{V}$ achieves the rate-distortion limit, then we have the Markov chain $V \to f(V) \to \widehat{V}$*

*Proof.* Assume for the sake of contradiction that there exists a random variable $\widehat{V}$ that achieves the rate-distortion limit with $D = 0$ but does not induce the Markov chain $V \to f(V) \to \widehat{V}$. Since mutual information is non-negative and $\mathbb{I}(V; \widehat{V} \mid f(V)) = 0$ implies the Markov chain $V \to f(V) \to \widehat{V}$, it must be the case that $\mathbb{I}(V; \widehat{V} \mid f(V)) > 0$. Consider an independent random variable $\widehat{V}' \sim \mathbb{P}(\widehat{V} \mid f(V))$ such that

$$\mathbb{I}(V; \widehat{V}') = \mathbb{I}(V; \widehat{V}) - \underbrace{\mathbb{I}(V; \widehat{V} \mid f(V))}_{>0} < \mathbb{I}(V; \widehat{V}) = \mathcal{R}(D).$$

Clearly, we have retained all bits of information needed to preserve $f(V)$ in $\widehat{V}'$, thereby achieving the same expected distortion constraint. However, this implies that $\widehat{V}'$ achieves a strictly lower rate, contradicting our assumption that $\widehat{V}$ achieves the rate-distortion limit. Therefore, it must be the case that when $D = 0$ and $\widehat{V}$ achieves the rate-distortion limit, we have $\mathbb{I}(V; \widehat{V} \mid f(V)) = 0$ which implies the Markov chain $V \to f(V) \to \widehat{V}$.

$\square$

**Lemma 12.** *For each episode $k \in [K]$ and for $D = 0$, let $\widetilde{\mathcal{M}}_k$ be a MDP that achieves the rate-distortion limit of $\mathcal{R}_k^{Q^\star}(D)$ with information source $\mathbb{P}(\mathcal{M}^\star \mid H_k)$ and distortion function $d_{Q^\star}$. Then, we have the Markov chain $\mathcal{M}^\star \to Q_{\mathcal{M}^\star}^\star \to \widetilde{\mathcal{M}}_k$, where $Q_{\mathcal{M}^\star}^\star = \{Q_{\mathcal{M}^\star,h}^\star\}_{h \in [H]}$ is the collection of random variables denoting the optimal action-value functions of $\mathcal{M}^\star$.*

*Proof.* Recall that our distortion function,

$$d_{Q^\star}(\mathcal{M}, \widehat{\mathcal{M}}) = \sup_{h \in [H]} ||Q_{\mathcal{M},h}^\star - Q_{\widehat{\mathcal{M}},h}^\star||_\infty^2 = \sup_{h \in [H]} \max_{(s,a) \in \mathcal{S} \times \mathcal{A}} |Q_{\mathcal{M},h}^\star(s,a) - Q_{\widehat{\mathcal{M}},h}^\star(s,a)|^2,$$

only depends on the MDPs $\mathcal{M}$ and $\widehat{\mathcal{M}}$ through their respective optimal action-value functions, $\{Q_{\mathcal{M},h}^\star\}_{h \in [H]}$ and $\{Q_{\widehat{\mathcal{M}},h}^\star\}_{h \in [H]}$. Consequently, the claim holds immediately by applying Lemma 11 where $f$ computes the optimal action-value functions of an input MDP for each timestep $h \in [H]$ and $\ell$ is the metric induced by the infinity norm on $\mathbb{R}^{|\mathcal{S}| \times |\mathcal{A}|}$. $\square$

# K    Proof of Theorem 5

Our proof of Theorem 5 proceeds by leveraging Lemma 3 (instead of Fact 1) before following the same style of argument as used in Theorem 3.

**Theorem 11.** *For $D = 0$ and any prior distribution $\mathbb{P}(\mathcal{M}^\star \in \cdot \mid H_1)$ over tabular MDPs, if $\Gamma_k \leq \overline{\Gamma}$ for all $k \in [K]$, then VSRL with distortion function $d_{Q^\star}$ has*

$$\textsc{BayesRegret}(K, \pi^{(1)}, \dots, \pi^{(K)}) \leq \widetilde{\mathcal{O}} \left( \sqrt{\overline{\Gamma} |\mathcal{S}||\mathcal{A}| K H} \right).$$

*Proof.* Starting with the information-theoretic regret bound in Corollary 2, observe that for $\mathcal{M}^\star \sim \mathbb{P}(\mathcal{M}^\star \in \cdot \mid H_1)$, we have the Markov chain $\mathcal{M}^\star \to Q_{\mathcal{M}^\star}^\star \to \widetilde{\mathcal{M}}_1$, by virtue of Lemma 3. By the data-processing inequality, we immediately recover the following chain of inequalities:

$$\mathcal{R}_1^{Q^\star}(D) \leq \mathbb{I}_1(\mathcal{M}^\star; \widetilde{\mathcal{M}}_1) \leq \mathbb{I}_1(\mathcal{M}^\star; Q_{\mathcal{M}^\star}^\star).$$

Recognizing that the optimal value functions are a deterministic function of the MDP $\mathcal{M}^\star$ itself, we have

$$\mathbb{I}_1(\mathcal{M}^\star; Q_{\mathcal{M}^\star}^\star) = \mathbb{H}_k(Q_{\mathcal{M}^\star}^\star) - \mathbb{H}_k(Q_{\mathcal{M}^\star}^\star \mid \mathcal{M}^\star) = \mathbb{H}_k(Q_{\mathcal{M}^\star}^\star) = \mathbb{H}_k(Q_{\mathcal{M}^\star,1}^\star, \dots, Q_{\mathcal{M}^\star,H}^\star) \leq \sum_{h=1}^{H} \mathbb{H}_k(Q_{\mathcal{M}^\star,h}^\star),$$

where the final inequality follows by applying the chain rule of entropy and the fact that conditioning reduces entropy, in sequence.

At this point, recalling the salient exposition in the proof of Theorem 3 concerning the use of metric entropy for such function-valued random variables, we proceed to consider the $\varepsilon_{Q^\star}$-cover of the interval $[0, H]$ with respect to the $L_1$-norm $|| \cdot ||_1$, for some fixed $0 < \varepsilon_{Q^\star} < H$. Since, for a sufficiently small choice of $\varepsilon_{Q^\star}$, $Q_{\mathcal{M}^\star,h}^\star$ is well-approximated as a discrete random variable for any $h \in [H]$, we recall that the entropy of a discrete random variable $X$ taking values on $\mathcal{X}$ is bounded as $\mathbb{H}(X) \leq \log(|\mathcal{X}|)$. Applying this upper bound and Lemma 9 in sequence, we have that

$$\sum_{h=1}^{H} \mathbb{H}_k(Q_{\mathcal{M}^\star,h}^\star) \leq |\mathcal{S}||\mathcal{A}| H \log \left( \mathcal{N}(\frac{\varepsilon_{Q^\star}}{2}, [0, H], || \cdot ||_1) \right) \leq |\mathcal{S}||\mathcal{A}| H \log \left( 1 + \frac{4H}{\varepsilon_{Q^\star}} \right).$$

Applying these upper bounds to the result of Corollary 5 and recalling that $D = 0$, we have

$$\textsc{BayesRegret}(K, \pi^{(1)}, \ldots, \pi^{(K)}) \leq \sqrt{\bar{\Gamma} K |\mathcal{S}||\mathcal{A}| H \log \left( 1 + \frac{4H}{\varepsilon_{Q^\star}} \right)}.$$

$\square$