# OpenReview forum: "Deciding What to Model: Value-Equivalent Sampling for Reinforcement Learning"
_NeurIPS.cc/2022/Conference — NeurIPS 2022 Accept_

### Official Review · Reviewer_gPNG · 2022-07-07

**Rating:** 5
**Confidence:** 2
**Soundness:** 3 good
**Presentation:** 4 excellent
**Contribution:** 2 fair

**Summary:**

The authors introduce Value-equivalent Sampling for Reinforcement Learning (VSRL), an extension to Posterior Sampling for Reinforcement Learning (PSRL).

The key idea behind VSRL is to sequentially solve simplified versions of the true environment with the ultimate goal of minimizing the bayesian regret of the policy.

An information-theoretic analysis of VSRL is provided with regret bounds for the general case, the tabular setting, and an agent with limited capacity.

**Questions:**

From what I understand, it is not trivial how to instantiate VSRL in practice. Could the authors think of a minimal example on which VSRL could be run? For instance, in the same vein as the experiments run in PSRL. Such experiments would definitely change my view on the limitations of this paper, which has already a strong theoretical foundation.

**Limitations:**

The main limitation of this paper is the absence of experimental results. The authors leave the application of VSRL to actual environments as future work, but in my opinion this work is incomplete without at least toy experiments demonstrating the usefulness of the approach.

The authors do not discuss potential negative impact of their work, but it is rather not applicable here since the paper is mostly about theory.

**Strengths And Weaknesses:**

# Strengths

- The method proposed is well motivated and intuitive.
- It is supported by a thorough theoretical analysis, although regret analysis is not my area of expertise.
- The paper is mostly well written and has a nice flow of sections and ideas.

# Weaknesses

The authors do not provide any empirical evidence supporting their method. VSRL is framed as an algorithm for real-world high-dimensional settings where current agents may fail due to computational issues, but it is not even applied to a toy environment. Therefore, the reader is left at the stage of theoretical guarantees, with no actual idea of how to implement the algorithm, and how it would perform in practice.

---

> ### Author Response · Authors · 2022-08-01
> **Response to Reviewer gPNG**
>
> We thank the reviewer for their time and effort in providing feedback on our paper, despite regret analyses not being a core area of their expertise.
>
> > in my opinion this work is incomplete without at least toy experiments demonstrating the usefulness of the approach.
>
> While we appreciate the reviewer’s appetite for an empirical demonstration of our proposed algorithm, we wish to emphasize that the lack of an empirical contribution is deliberate and our paper is, by design, focused on reinforcement learning theory. Obviously, the field is built upon the contributions of purely theoretical work and, as noted by R7znE, empirical results are by no means a prerequisite to the publication of such work.
>
> > The authors do not provide any empirical evidence supporting their method. VSRL is framed as an algorithm for real-world high-dimensional settings where current agents may fail due to computational issues, but it is not even applied to a toy environment. Therefore, the reader is left at the stage of theoretical guarantees, with no actual idea of how to implement the algorithm, and how it would perform in practice.
>
> If the reviewer has not already done so, we kindly invite them to consult Section C.3 of the appendix, which is aimed at bridging the reviewer’s stated gap between our proposed theoretical algorithm and its practical implementation. As mentioned in the section, empirical evidence for handling the rate-distortion optimization issues already exists in the multi-armed bandit setting. As for further clarifying details that might paint a clearer picture of how our proposed theory can inform practice, we invite the reviewer to please consult the response to RiP5r/R7znE, who raised related concerns.

---

> > ### Comment · Reviewer_gPNG · 2022-08-07
> > **Response to rebuttal**
> >
> > While I agree that empirical results are not a prerequisite for an RL theory paper, I find it difficult to see how the field will build upon this work. However, I appreciate the efforts made by the authors to give some research directions towards a practical instantiation of VSRL. Besides, I reiterate that the paper is well written and seems technically sound. Hence, I have updated my score to 5.

---

### Official Review · Reviewer_7znE · 2022-07-10

**Rating:** 6
**Confidence:** 3
**Soundness:** 4 excellent
**Presentation:** 4 excellent
**Contribution:** 3 good

**Summary:**

This paper studies exploration in reinforcement learning from the Bayesian perspective, specifically adopting a Thompson sampling approach to obtain bounded Bayesian regret. The distinguishing factor in this analysis is the addition of a model compression component in which the user can accept a (tunable) amount of additional regret in exchange for simplifying the sampled model that must be solved in each iteration, potentially improving computational and/or sample complexity.

**Questions:**

No questions.

**Limitations:**

Yes, I thought the paper did an excellent job of acknowledging limitations. For instance, it acknowledges a missing detail in the analysis (the form of a bound on information ratios). It is also quite forthcoming about the limited connection of the theory to practically applicable algorithms.

**Strengths And Weaknesses:**

This paper moves the theory of efficient RL in an interesting direction with a novel combination of two existing ideas (PSRL and rate-distortion theory). I found it to be well-written and compelling. It is unlikely to impact practice, but it does contribute to our understanding of sample efficiency in RL, specifically by adding a tradeoff between optimality and model complexity.

--Originality--

The algorithm is clearly built upon the existing PSRL algorithm, which is fully acknowledged within the paper. The theoretical arguments are also built upon existing analyses using rate-distortion theory to quantify the simplification of the problem in a bandit context. The paper also fully acknowledges these connections.

I do think that the combination of these approaches is novel, worthwhile, and non-trivial.

The paper also does an excellent job of situating this work in the context of other theoretical results regarding efficient reinforcement learning. This occurs both in the main text and in an extensive appendix.

--Quality--

The findings in this paper are all theoretical in nature. The proofs seem well argued to me and I found no fault with them. That said, this is my first exposure to rate-distortion theory so it is possible that I would not detect a misapplication of those ideas.

--Clarity--

Overall, I found the paper to be well-written. As is typical for a theoretical paper, this paper is technically dense and requires close attention from the reader. Nevertheless, the authors have done a good job of surfacing their motivation, the core ideas of their analysis, and the potential consequences. I also appreciate the attention given to helping the reader develop intuition with rate-distortion, which is not as widely familiar to the ICML audience as other aspects of the work.

I do have one minor clarity suggestion: the paper currently uses R for both reward and rate-distortion; it might be better to separate the notation for those two concepts.

--Significance--

I think this paper tackles an important issue. The theory of efficient RL typically implicitly assumes that the agent is capable of learning a perfect model of and then perfectly solving an arbitrary MDP. It is refreshing to see theoretical work in this area that widens attention beyond this limiting assumption. It seems plausible that the framework used here could be built upon in future work to further study efficient RL with imperfect models, for instance considering different possible limits on agent capacity.

I will also say that it is not entirely clear what lesson to take from these results. The paper acknowledges that this algorithm is not practically implementable (not uncommon for a paper of this sort). But there is a high-level principle here that could inspire more practical approaches, namely that if you are careful, you can get away with simplifying the sampled model that you get from your posterior so it's easier to solve.

In all, I don't think that this paper presents a notably surprising theoretical result and certainly won't impact practice in the near term, but it does move the ball in an important direction.

---

> ### Author Response · Authors · 2022-08-01
> **Response to Reviewer 7znE**
>
> We thank the reviewer for their detailed assessment of our paper. We appreciate the reviewer’s comments acknowledging the novelty and compelling nature of our contributions. We will certainly keep the minor clarity suggestion in mind in the next iteration of the paper. Our goal in this response is to address the reviewer’s comments regarding practicality and overall takeaways of our paper; hopefully, our response might allow the reviewer to reconsider the potential impact of our work on practice in the near term.
>
> > I will also say that it is not entirely clear what lesson to take from these results. The paper acknowledges that this algorithm is not practically implementable (not uncommon for a paper of this sort). But there is a high-level principle here that could inspire more practical approaches, namely that if you are careful, you can get away with simplifying the sampled model that you get from your posterior so it's easier to solve.
>
> > In all, I don't think that this paper presents a notably surprising theoretical result and certainly won't impact practice in the near term, but it does move the ball in an important direction.
>
> Inspired by how RiP5r posed their questions about our paper, we split the response into two discussions on (1) the practicality of VSRL/PSRL and (2) how the rate-distortion theory connection elucidated in this work might augment the design of existing algorithms that exploit value equivalence.
>
> In an updated version of the paper we intend to amend Section C.3 of the appendix to more explicitly clarify how this work might inform the development of new algorithms. Specifically, we can call attention to the fact that practical instantiations of the Bayesian reinforcement learning setting adopted in this work apply the principle of randomized value functions (RVF), which is equivalent to PSRL [1]. These methods represent epistemic uncertainty over the optimal value function $Q^\star$, rather than the underlying model, making them more amenable to implementation via deep neural networks [2-6]. Moreover, extensions of this line of work have gone on to consider other avenues for leveraging such principled, practical methods for handling the exploration challenge through successor features [7] and temporal-difference errors [8]. Overall, we believe that VSRL can serve an analogous role in this sense to PSRL, providing a sound theoretical foundation that gives rise to subsequent practical algorithms.
>
> Orthogonal to the practicality of VSRL, we believe it is also important to provide insight in the paper (either in Section C.3, elsewhere in Section C, or maybe even in Section 3.3 itself) on how our derived information-theoretic connections might inform practical use of the value equivalence principle. One of the core insights developed here is the formalization of model simplifications arising out of the value equivalence principle as a form of lossy compression. Curiously, recent meta reinforcement-learning approaches applied to multi-task or contextual MDPs arrive at a similar need of obtaining compressed representations of the underlying MDP model during a meta-exploration phase in order to facilitate few-shot learning [9-11]. Notably, while these methods do articulate these latent/contextual variational encoders as posterior distributions over the task, they lack any representation of epistemic uncertainty, similar to MuZero.  While the connection we draw here is tenuous, especially relative to the more concrete discussion around the practicality of VSRL/PSRL above, we strongly suspect that the known connections between rate-distortion theory and information bottlenecks [12, 13] represent a viable path to bridging the ideas between our theory and value equivalence in practice.

---

> > ### Author Response · Authors · 2022-08-01
> > **References**
> >
> > [1] Osband, Ian. "Deep Exploration via Randomized Value Functions." PhD dissertation, Stanford University, 2016.
> >
> > [2] Osband, Ian, Charles Blundell, Alexander Pritzel, and Benjamin Van Roy. "Deep exploration via bootstrapped DQN." Advances in neural information processing systems 29 (2016).
> >
> > [3] Osband, Ian, John Aslanides, and Albin Cassirer. "Randomized prior functions for deep reinforcement learning." Advances in Neural Information Processing Systems 31 (2018).
> >
> > [4] O’Donoghue, Brendan, Ian Osband, Remi Munos, and Volodymyr Mnih. "The uncertainty Bellman equation and exploration." In International Conference on Machine Learning, pp. 3836-3845. 2018.
> >
> > [5] Dwaracherla, Vikranth, Xiuyuan Lu, Morteza Ibrahimi, Ian Osband, Zheng Wen, and Benjamin Van Roy. "Hypermodels for Exploration." In International Conference on Learning Representations. 2019.
> >
> > [6] Osband, Ian, Zheng Wen, Mohammad Asghari, Morteza Ibrahimi, Xiyuan Lu, and Benjamin Van Roy. "Epistemic neural networks." arXiv preprint arXiv:2107.08924 (2021).
> >
> > [7] Janz, David, Jiri Hron, Przemysław Mazur, Katja Hofmann, José Miguel Hernández-Lobato, and Sebastian Tschiatschek. "Successor uncertainties: exploration and uncertainty in temporal difference learning." Advances in Neural Information Processing Systems 32 (2019).
> >
> > [8] Flennerhag, Sebastian, Jane X. Wang, Pablo Sprechmann, Francesco Visin, Alexandre Galashov, Steven Kapturowski, Diana L. Borsa, Nicolas Heess, Andre Barreto, and Razvan Pascanu. "Temporal difference uncertainties as a signal for exploration." arXiv preprint arXiv:2010.02255 (2020).
> >
> > [9] Rakelly, Kate, Aurick Zhou, Chelsea Finn, Sergey Levine, and Deirdre Quillen. "Efficient off-policy meta-reinforcement learning via probabilistic context variables." In International conference on machine learning, pp. 5331-5340. PMLR, 2019.
> >
> > [10] Zintgraf, Luisa, Kyriacos Shiarlis, Maximilian Igl, Sebastian Schulze, Yarin Gal, Katja Hofmann, and Shimon Whiteson. "VariBAD: A Very Good Method for Bayes-Adaptive Deep RL via Meta-Learning." In International Conference on Learning Representations. 2019.
> >
> > [11]  Liu, Evan Z., Aditi Raghunathan, Percy Liang, and Chelsea Finn. "Decoupling exploration and exploitation for meta-reinforcement learning without sacrifices." In International conference on machine learning, pp. 6925-6935. PMLR, 2021.
> >
> > [12] Tishby, Naftali, Fernando C. Pereira, and William Bialek. "The information bottleneck method." arXiv preprint physics/0004057 (2000).
> >
> > [13]  Alemi, Alexander, Ben Poole, Ian Fischer, Joshua Dillon, Rif A. Saurous, and Kevin Murphy. "Fixing a broken ELBO." In International Conference on Machine Learning, pp. 159-168. PMLR, 2018.

---

> > ### Comment · Reviewer_7znE · 2022-08-07
> > **Generally in agreement**
> >
> > Thanks to the authors for the response. I generally agree with the points here, and I believe my review expresses as much (my apologies if it doesn't come across). In particular:
> > - I do see the potential for this theoretical insight to inspire more practical algorithms in the future (just as PSRL has)
> > - I do agree that the novel framing of the question as an application of compression has the opportunity to open up new avenues toward scientific/theoretical understanding of the impact of model simplification.
> >
> > At the same time, I think it's the case (correct me if I am misunderstanding) that the main takeaway from this result fairly generic: you can get away with simplifying your model during planning as long as you approximately preserve Bellman errors. I do genuinely appreciate that this result shows how one can trade off aggressive simplification and control performance and that it re-formalizes the idea of value equivalence in a new way. At the same time, it doesn't offer much new insight that would, for instance, cause a practitioner to change the way they are approaching value equivalent modeling or even significantly change how they understand it to be working (roughly speaking, that the model is just complex enough to do a good job of giving value estimates). I find this result interesting and novel and I would be glad to see it in the literature, but, for those reasons, I do think its near-term impact is likely to be limited.
> >
> > Genuine question: have I missed something or mischaracterized something in the above?

---

> > > ### Author Response · Authors · 2022-08-08
> > > **Reply to Reviewer R7znE**
> > >
> > > We thank the reviewer for their reply and greatly appreciate their comments.
> > >
> > > We concur with the reviewer's assessment of the impact of our work on the near term; namely that, while our contributions do constitute useful insights and valuable new perspective, their capacity to influence the behavior of practitioners in the immediate future is still somewhat limited. A task we knowingly leave to follow-up work is the empirical demonstration and dissemination of how these theoretical ideas can concretely shape practice.
> > >
> > > Based on all the comments and productive discussion, we believe the reviewer has neither missed nor mischaracterized our contributions.

---

### Official Review · Reviewer_iP5r · 2022-07-11

**Rating:** 6
**Confidence:** 3
**Soundness:** 3 good
**Presentation:** 3 good
**Contribution:** 2 fair

**Summary:**

In this paper the authors describe a new theoretical framework for exploration in RL. Their work builds upon earlier results on Posterior Sampling in Reinforcement learning. The main idea is to obtain posterior models which are compressed versions of the true MDP using the value equivalence principle as a distortion metric. By doing so they can harness the benefits of using value equivalent models and of posterior sampling. They provide a Bayesian regret bound for their method, which is the result of the regret caused by the use of a compressed model and that incurred while learning the model.

**Questions:**

My main concern is that it is not clear how this work can be useful in developing new algorithms even in simple tabular problems. The authors do not provide a practical instantiation of their method even in small toy domains and instead discuss the challenges of developing such an algorithm. This is different from standard PSRL. At the same time I cannot see how the results presented in this work can inform the design of better value equivalent algorithms which usually operate in high dimensional problems at scale (i.e. MuZero). Can the authors provide more insight on this ?

**Limitations:**

The authors have adequately addressed the limitations and potential negative societal impact of their work.

**Strengths And Weaknesses:**

The paper provides a solid theoretical framework for value equivalent posterior sampling in reinforcement learning. The recent success and simplicity of value-equivalent approaches provide motivation for such work. On the other hand, posterior sampling provides solid theoretical guarantees to the problem of exploration in RL. Studying the combination of the two approaches can be useful to the community.

The paper is well written / structured and provides a lot of additional analysis and discussion in the appendix. My main concern is described in the questions section to the authors.

---

> ### Author Response · Authors · 2022-08-01
> **Response to Reviewer iP5r**
>
> We appreciate the reviewer’s time and effort in providing feedback on our paper. We were glad to see that the reviewer recognizes the impact and value of our theoretical contributions. While the reviewer echoes the other reviewers in their concern over the implications of our theory on practice, we separate their main question into two pieces: (1) the practicality of VSRL/PSRL and (2) how the rate-distortion theory connection elucidated in this work might augment the design of existing algorithms that exploit value equivalence.
>
> > My main concern is that it is not clear how this work can be useful in developing new algorithms even in simple tabular problems. The authors do not provide a practical instantiation of their method even in small toy domains and instead discuss the challenges of developing such an algorithm. This is different from standard PSRL.
>
> The reviewer is correct that PSRL, when restricted to tabular MDPs, is amenable to practical implementation (rewards at each state-action pair can be modeled with a Beta prior and transition distributions treated via a Dirichlet prior). For more complicated, higher-dimensional settings, however, representing epistemic uncertainty over the full model of the MDP and performing exact planning over sampled models are open challenges. Consequently, aside from the deliberate theoretical focus of this work, we felt that the paper would not be well served by toy experiments incapable of aligning with the setting that motivates our algorithm.
>
> That said, we feel that an expansion of or addendum to Section C.3 in the appendix is certainly possible to clarify how this work might inform the development of new algorithms. Specifically, we can call attention to the fact that practical instantiations of the Bayesian reinforcement learning setting adopted in this work apply the principle of randomized value functions (RVF), which is equivalent to PSRL [1]. These methods represent epistemic uncertainty over the optimal value function $Q^\star$, rather than the underlying model, making them more amenable to implementation via deep neural networks [2-6]. Moreover, extensions of this line of work have gone on to consider other avenues for leveraging such principled, practical methods for handling the exploration challenge through successor features [7] and temporal-difference errors [8]. Overall, we believe that VSRL can serve an analogous role in this sense to PSRL, providing a sound theoretical foundation that gives rise to subsequent practical algorithms.
>
> > At the same time I cannot see how the results presented in this work can inform the design of better value equivalent algorithms which usually operate in high dimensional problems at scale (i.e. MuZero). Can the authors provide more insight on this ?
>
> Orthogonal to the practicality of VSRL, we believe it is also important to provide insight in the paper (either in Section C.3, elsewhere in Section C, or maybe even in Section 3.3 itself) on how our derived information-theoretic connections might inform practical use of the value equivalence principle. One of the core insights developed here is the formalization of model simplifications arising out of the value equivalence principle as a form of lossy compression. Curiously, recent meta reinforcement-learning approaches applied to multi-task or contextual MDPs arrive at a similar need of obtaining compressed representations of the underlying MDP model during a meta-exploration phase in order to facilitate few-shot learning [9-11]. Notably, while these methods do articulate these latent/contextual variational encoders as posterior distributions over the task, they lack any representation of epistemic uncertainty, similar to MuZero.  While the connection we draw here is tenuous relative to the more concrete discussion around the practicality of VSRL/PSRL above, we strongly suspect that the known connections between rate-distortion theory and information bottlenecks [12, 13] represent a viable path to bridging the ideas between our theory and value equivalence in practice.

---

> > ### Author Response · Authors · 2022-08-01
> > **References**
> >
> > [1] Osband, Ian. "Deep Exploration via Randomized Value Functions." PhD dissertation, Stanford University, 2016.
> >
> > [2] Osband, Ian, Charles Blundell, Alexander Pritzel, and Benjamin Van Roy. "Deep exploration via bootstrapped DQN." Advances in neural information processing systems 29 (2016).
> >
> > [3] Osband, Ian, John Aslanides, and Albin Cassirer. "Randomized prior functions for deep reinforcement learning." Advances in Neural Information Processing Systems 31 (2018).
> >
> > [4] O’Donoghue, Brendan, Ian Osband, Remi Munos, and Volodymyr Mnih. "The uncertainty Bellman equation and exploration." In International Conference on Machine Learning, pp. 3836-3845. 2018.
> >
> > [5] Dwaracherla, Vikranth, Xiuyuan Lu, Morteza Ibrahimi, Ian Osband, Zheng Wen, and Benjamin Van Roy. "Hypermodels for Exploration." In International Conference on Learning Representations. 2019.
> >
> > [6] Osband, Ian, Zheng Wen, Mohammad Asghari, Morteza Ibrahimi, Xiyuan Lu, and Benjamin Van Roy. "Epistemic neural networks." arXiv preprint arXiv:2107.08924 (2021).
> >
> > [7] Janz, David, Jiri Hron, Przemysław Mazur, Katja Hofmann, José Miguel Hernández-Lobato, and Sebastian Tschiatschek. "Successor uncertainties: exploration and uncertainty in temporal difference learning." Advances in Neural Information Processing Systems 32 (2019).
> >
> > [8] Flennerhag, Sebastian, Jane X. Wang, Pablo Sprechmann, Francesco Visin, Alexandre Galashov, Steven Kapturowski, Diana L. Borsa, Nicolas Heess, Andre Barreto, and Razvan Pascanu. "Temporal difference uncertainties as a signal for exploration." arXiv preprint arXiv:2010.02255 (2020).
> >
> > [9] Rakelly, Kate, Aurick Zhou, Chelsea Finn, Sergey Levine, and Deirdre Quillen. "Efficient off-policy meta-reinforcement learning via probabilistic context variables." In International conference on machine learning, pp. 5331-5340. PMLR, 2019.
> >
> > [10] Zintgraf, Luisa, Kyriacos Shiarlis, Maximilian Igl, Sebastian Schulze, Yarin Gal, Katja Hofmann, and Shimon Whiteson. "VariBAD: A Very Good Method for Bayes-Adaptive Deep RL via Meta-Learning." In International Conference on Learning Representations. 2019.
> >
> > [11]  Liu, Evan Z., Aditi Raghunathan, Percy Liang, and Chelsea Finn. "Decoupling exploration and exploitation for meta-reinforcement learning without sacrifices." In International conference on machine learning, pp. 6925-6935. PMLR, 2021.
> >
> > [12] Tishby, Naftali, Fernando C. Pereira, and William Bialek. "The information bottleneck method." arXiv preprint physics/0004057 (2000).
> >
> > [13]  Alemi, Alexander, Ben Poole, Ian Fischer, Joshua Dillon, Rif A. Saurous, and Kevin Murphy. "Fixing a broken ELBO." In International Conference on Machine Learning, pp. 159-168. PMLR, 2018.

---

### Official Review · Reviewer_UCCG · 2022-07-15

**Rating:** 5
**Confidence:** 1
**Soundness:** 2 fair
**Presentation:** 3 good
**Contribution:** 3 good

**Summary:**

This paper studies value equivalence and considers the scenario where the agent limitations may entirely preclude identifying an exact value equivalent model, which immediately gives rise to a trade-off between identifying a model that is simple enough to learn while only incurring bounded suboptimality. To address this problem, the paper proposes an algorithm that iteratively computes an approximately-value-equivalent, lossy compression of the environment which an agent may feasibly target in lieu of the true model. It also proves an information-theoretic, Bayesian regret bound for the proposed algorithm that holds for any finite-horizon, episodic sequential decision-making problem.


**Questions:**

1. On the significance of the theoretical results in terms of practicality in the deep reinforcement learning setting: Even though the theoretical results of this paper are interesting by themselves, I think that providing a brief section on how these results can at least help developing better deep RL algorithms (algorithms that work with high dimensional inputs) can significantly benefit the paper. Is it possible to provide a discussion on this?
2. On the significance of the illustrative example multi-resolution MDP: Is it possible to provide a discussion on why the MDP structure in example 1 is an important one? When does this structure become important? As far as I know, most of the MDPs of interest in the literature do not have this kind of a structure.


**Limitations:**

Yes the authors mention the limitations of their work.


**Strengths And Weaknesses:**

**Originality:** The paper seems to be original in the sense that it introduces an algorithm that implicitly identifies, in a purely data-driven and automated fashion, the resource limitations of the agent to facilitate tractable, near-optimal learning in what may otherwise be an intractable problem. Though I would like to note that I am not particularly familiar with this area of research.

**Quality:** Overall, there seems to be no serious quality issues with the paper. Though, I would like to indicate that I haven’t checked the theoretical results in a very detailed manner and just went over them at a high-level.

**Clarity:** I found no problems with the clarity of the paper. The paper is very well-written. I have nothing to suggest.

**Significance:** Even though the derived theoretical results seem to be interesting in their own sense, I would like to indicate that I found the paper to have problems in the significance of the results. More specifically, I am not sure if the paper provides enough discussion on how the derived results and proposed algorithm can be beneficial in the deep reinforcement learning setting. More on this can be found in the Questions section below.

---

> ### Author Response · Authors · 2022-08-01
> **Response to Reviewer UCCG**
>
> We appreciate the reviewer’s comments on our paper despite the submission being out of their primary area.
>
> Admittedly, we were a bit surprised by the reviewer’s low scores on soundness and presentation, despite the main criticisms of their review focusing on issues of significance. We will address the latter in our response and hope that the reviewer will provide clarification on any issues with the former axes as needed.
>
> > I think that providing a brief section on how these results can at least help developing better deep RL algorithms (algorithms that work with high dimensional inputs) can significantly benefit the paper. Is it possible to provide a discussion on this?
>
> Overall, this first question by the reviewer touches upon a point raised in common by all reviewers of our paper; namely, that the contributions of this paper are motivated by real-world reinforcement learning and yet we do not offer any computational experiments to this effect. We wish to emphasize that the lack of an empirical contribution is deliberate and our paper is, by design, focused on reinforcement learning theory. Nevertheless, we appreciate the reviewers’ broader point that theory, without an eye towards practice, is perhaps less impactful as a whole. If the reviewer has not already done so, we kindly invite them to consult Section C.3 of the appendix, which is aimed at providing a direct response to the reviewer’s first question. However, based on comments by RiP5r, this section possibly leaves something to be desired.
>
> To remedy this, we plan on expanding Section C.3 to include discussion of more scalable versions of VSRL through randomized value function (RVF) approaches, which are equivalent to PSRL [1]. These methods represent epistemic uncertainty over the optimal value function $Q^\star$, rather than the underlying model, making them more amenable to implementation via deep neural networks [2-6].
>
> > On the significance of the illustrative example multi-resolution MDP: Is it possible to provide a discussion on why the MDP structure in example 1 is an important one? When does this structure become important? As far as I know, most of the MDPs of interest in the literature do not have this kind of a structure.
>
> This example represents a sequential decision-making process that is hierarchical in nature, emblematic of human cognition, for instance, where high-level task execution and reasoning occurs at an abstract level while simultaneously being translated into movements at the lower level of individual body parts, which are themselves governed by individual muscle contractions orchestrated at the lowest level of granularity by electric impulses through the brain and nervous system.  One might aspire to have a reinforcement-learning agent competently control such a  complex system, but requiring this agent to identify every environmental detail in order to do so seems like a rather unlikely path for success.
>
>
> [1] Osband, Ian. "Deep Exploration via Randomized Value Functions." PhD dissertation, Stanford University, 2016.
>
> [2] Osband, Ian, Charles Blundell, Alexander Pritzel, and Benjamin Van Roy. "Deep exploration via bootstrapped DQN." Advances in neural information processing systems 29 (2016).
>
> [3] Osband, Ian, John Aslanides, and Albin Cassirer. "Randomized prior functions for deep reinforcement learning." Advances in Neural Information Processing Systems 31 (2018).
>
> [4] O’Donoghue, Brendan, Ian Osband, Remi Munos, and Volodymyr Mnih. "The uncertainty Bellman equation and exploration." In International Conference on Machine Learning, pp. 3836-3845. 2018.
>
> [5] Dwaracherla, Vikranth, Xiuyuan Lu, Morteza Ibrahimi, Ian Osband, Zheng Wen, and Benjamin Van Roy. "Hypermodels for Exploration." In International Conference on Learning Representations. 2019.
>
> [6] Osband, Ian, Zheng Wen, Mohammad Asghari, Morteza Ibrahimi, Xiyuan Lu, and Benjamin Van Roy. "Epistemic neural networks." arXiv preprint arXiv:2107.08924 (2021).

---

> > ### Comment · Reviewer_UCCG · 2022-08-07
> > **Response to the Rebuttal**
> >
> > I would like to thank the authors for clarifying my concerns and answering my questions. I think that expanding Section would definitely improve the paper's quality in terms of practically (even though its primary focus is not this). However, as the paper is out of my primary area of research, to keep things neutral, I am leaning on keeping my score and confidence the same.

---

### Meta-Review · Area_Chair_CJh5 · 2022-08-26

**Recommendation:** Accept
**Confidence:** Less certain

**Metareview:**

This paper studies a model-based reinforcement learning approach that is able to identify a model that is simple enough to be learned by a limited agent while incurring only bounded sub-optimality.
After reading each other's reviews and the authors' feedback, most of the reviewers' concerns have been solved and the reviewers agree that this paper deserves publication.
While preparing the final version of their paper, the authors need to consider the reviewers' concerns about the significance of the proposed approach and how the theoretical findings presented in the paper can inspire practical algorithms.

**Award:**

No

---

### Decision · Program_Chairs · 2022-09-14

Accept